# Deep neural networks divide and conquer dihedral multiplication

## Abstract

We find multilayer perceptrons and transformers both learn an instantiation of the same divide-and-conquer algorithm and solve dihedral multiplication with logarithmic feature efficiency. Applying principal component analyses to clusters of neurons where neurons have similar activation behavior reveals remarkably clear structure: *the neural representations correspond to Cayley graphs.* It's noteworthy that these Cayley graphs are the low-dimensional manifolds networks are predicted to learn under the manifold hypothesis, and thus each neural representation learned by networks corresponds to a manifold. To our knowledge, our novel methodology allows this to be the first work that fully characterizes and describes the neural representations distributed over many neurons on a task.

## 1 Introduction

The universality hypothesis—that *deep neural networks (DNNs) trained on different (but related) datasets will make use of similar algorithmic strategies* (Li et al., 2015; Olah et al., 2020)—lies at the crux of the quest for generalizable interpretability methods. If true, there's hope for researchers to discover methods to automatically detect circuits in large language models (LLMs). Such a feature would be a great advancement towards safe, interpretable artificial intelligence.

Our motivation for this particular case study stems from two recent works. A position paper by Eberle et al. (2025) argues that we must work toward an algorithmic understanding of generative models. They state "What algorithms do LLMs actually learn and use to solve problems? Studies addressing this question are sparse, as research priorities are focused on improving performance through scale, leaving a theoretical and empirical gap in understanding emergent algorithms". Indeed, this gap has just recently started to be filled in. In a follow-up to the position, researchers used modular addition—the *commutative* cyclic group multiplication—to construct the first rigorous proof that neural networks learn a divide-and-conquer algorithm (McCracken et al., 2025). Furthermore, this work conjectured that the universality learned in group multiplications would involve coset circuits being learned and composed together to form divide-and-conquer algorithms.

Our work follows McCracken et al. (2025) and seeks to provide a harder example than modular addition to further understandings of how DNNs learn to fit data. We will study both MLPs and transformers that learned dihedral multiplication (which is non-commutative: $ab \neq ba$), to build on the commutative ($ab = ba$) results in the literature on modular addition. Our focus is *what* the algorithm DNNs universally recover is, *what* the neuron activations are, *what* the neural representations are, and *how* the neural representations interact via superposition to perform the global computation of the algorithm that's being learned. Our contributions are the following:

**A new example of universality.** Satisfyingly, we find neurons in DNNs trained on dihedral group multiplication learn either coset structure or approximate coset structure, aligning with prior work (Stander et al., 2024; McCracken et al., 2025).

**Cayley graphs as a new perspective on how DNNs learn group tasks.** By clustering neurons that learned the same (approximate) coset and studying the entire neuron-cluster simultaneously, our methodology yields a novel perspective on how DNNs *can* solve group multiplications: the neural representations are Cayley graphs. Indeed, we quantifiably analyze the neural representations and algorithmically determine their structure. We also study network preferences for various Cayley graphs by determining their rate of occurrence across seeds and architectures. We find remarkable

feature efficiency in all networks – $\mathcal{O}(\log(n))$ – as shown in Figure 6. Cayley graphs are learned universally by either transformers or MLPs in all dihedral groups tested over many orders of magnitude.

**DNNs divide and conquer dihedral multiplication.** We show that $\mathcal{O}(\log(n))$ neural representations (being Cayley graphs) are learned. Additionally, we show that as the context length of the transformer grows, a recursive pattern emerges in the activations. For each neural representation, DNNs recursively compute the equivalence class of $a$, then $b$, then $c$, and so on, to ultimately compute the equivalence class of the answer $C$ for multiplications $abc \cdots = C$. These answers are then linearly combined at the logits, with each neural representation pushing positive mass onto logits of the equivalence class they understand $C$ to belong to. Thus, the correct logit $C$ is maximized (see 7).

## 2 BACKGROUND AND SETUP

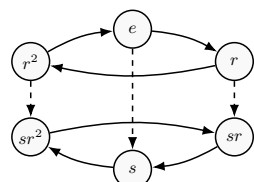

Figure 1: A $D_3$ Cayley Graph. Solid arrows apply left multiplication by $r$; dashed lines apply left multiplication by $s$. Non-commutativity appears $r \cdot s \neq s \cdot r$.

The *dihedral group* $D_n$ is the set of symmetries of a regular $n$-gon, containing $2n$ elements: $n$ rotations $r^k$ for $k \in \{0, \ldots n-1\}$ that rotate the $n$-gon by $2\pi/n$ radians, and $n$ reflections $sr^k$ reflecting about $n$ distinct axes. The rotation $r^0$ is the *identity* element, denoted $e$, for which $ex = xe = x$ for any $x \in D_n$. These operations form a non-commutative group multiplication when $n \geq 3$, meaning the order in which operations are multiplied matters—for instance, $sr \neq rs$. **Group multiplication**, $a \cdot b = C$, $a, b \in D_n$ involves composing two symmetries in sequence (from right to left): $r^a \cdot r^b = r^{(a+b) \bmod n}$ (rotation), $sr^a \cdot r^b = sr^{(a+b) \bmod n}$ (reflection), $r^a \cdot sr^b = sr^{(b-a) \bmod n}$ (reflection), $sr^a \cdot sr^b = r^{(b-a) \bmod n}$ (rotation).

**Cayley graphs** geometrically encode a group's structure. The Cayley graph of $D_n$ may be expressed via a *generating set* $\{r, s\}$, where nodes are group elements and (directed) edges are labeled by $\{r, s\}$. Particularly, an edge labeled $x \in \{r, s\}$ between nodes $a, b$ exists if $b = xa$. A Cayley graph for $D_3$ is in Fig. 1. Note: six $D_3$ Cayley graphs compose $D_{18}$, each one corresponding to a coset.

**Group Fourier Transform (GFT).** Analogously to the classical Fourier transform, which decomposes a time-domain signal into sinusoidal components, the group Fourier transform (GFT) decomposes a complex-valued function on a group $G$ into spectral components that reflect the symmetry structure of $G$. These components are indexed by the (unitary) irreducible representations of $G$ and capture how the function varies along different symmetry modes, such as rotational versus reflective modes in the case of the dihedral group $D_n$; see Appendix A.1.2.

**Cosets / approximate cosets.** Given a subgroup $H \leq G$, the left cosets $gH$ all have the same cardinality and form a partition of $G$. In our setting, this corresponds to the case where a learned "frequency" is well aligned with the group structure, so that the neuron's response is (approximately) constant on each coset block. By contrast, when the frequency does not divide the group size (for example, attempting to "divide" the dihedral group $D_{18}$ by 5), no such exact partition exists: the neuron's activations vary almost uniquely across individual group elements. We refer to this pattern of responses as an *approximate coset*.

**The Chinese remainder theorem** states that when the moduli $m$ and $n$ are coprime, arithmetic modulo $mn$ is equivalent to performing arithmetic modulo $m$ and modulo $n$ in parallel: every integer $x$ modulo $mn$ is uniquely determined by the pair of its remainders $(x \bmod m, x \bmod n)$. In our setting, this provides intuition for when a "frequency" or pattern factors through a product of smaller cyclic components of a group.

**Experimental setup.** We train one- and two-layer perceptrons (MLPs) to 100% test accuracy with 512 ReLU neurons per layer, with two different embedding matrices, one for $a$ and one for $b$, primarily on all pairs $(a, b) \in D_n \times D_n$, $n \in \{18, 19\}$. $D_{18}$ is chosen for primary presentation in the main text because it has more cosets compared to a prime or odd dihedral group, which allows clearer contrast of preferences for exact cosets vs. approximate cosets in Fig. 4. Let elements 0–17 be the rotation class elements $r^k$ and 18–35 the reflection class elements $sr^k$. For readability, neuron

preactivation plots insert a visual break between elements $17$ and $18$ to distinguish the two classes. On the Cayley graph under the sign character, $r^k$ elements lie in the $+1$ region, $sr^k$ in the $-1$.

We also train one- and two-layer transformers by treating the dihedral multiplication as a language, *e.g.*, given a sequence $a, b, \ldots$ predict the final token, which is the correct answer to the multiplication. We sample pairs of group elements uniformly from $D_n$. These transformers parallel language models, with one embedding matrix that all tokens share, and a context length equal to the length of the multiplication. They include positional embeddings and residual connections.

**Two-layer behavior.** For the two-layer networks, we empirically observe that the neural representations in the middle layer are geometrically highly similar to the cluster contributions to the logits in the corresponding one-layer networks. To keep the presentation concise, the main text therefore focuses on figures based on layer 1 activations, while additional quantitative results for the two-layer models are reported in Appendix B.6.1.

## 3 RELATED WORK

Group multiplication tasks have emerged as standard evaluation settings across both the mechanistic interpretability community (Nanda et al., 2023; Chughtai et al., 2023b;a; He et al., 2024; Tao et al., 2025; Doshi et al., 2023; Stander et al., 2024) and the theoretical deep learning literature (Gromov, 2023; Morwani et al., 2024; Mohamadi et al., 2023; McCracken et al., 2025). These tasks have served as a shared foundation for empirically and theoretically oriented researchers to test and validate their hypotheses. In particular, studies on group multiplication have become central to examining the *Universality Hypothesis* (Li et al., 2015; Olah et al., 2020; Chughtai et al., 2023a; Huh et al., 2024), which asserts that neural networks trained on related problems tend to develop internal structures that are similar. The hypothesis further suggests that, independent of factors such as model architecture, initialization, or training setup, deep neural networks will discover comparable representational mechanisms grounded in common underlying principles.

Empirically, the widely discussed phenomenon of grokking was initially discovered during experiments on modular addition—an instance of a group multiplication task (Power et al., 2022). This discovery inspired the work of Nanda et al. (2023), which offered remarkably clear algorithmic explanations for how transformer models exhibit grokking behavior. Building on this, Chughtai et al. (2023a) conducted a broader empirical test of the Universality Hypothesis by extending the analysis to both cyclic and permutation group multiplication tasks. Their study suggested that neural networks consistently developed internal mechanisms equivalent to learning matrix representations of groups and performing matrix multiplications to compute results. However, subsequent research challenged these conclusions. Zhong et al. (2023) demonstrated that distinct transformer architectures, when trained on modular addition, in fact learned two fundamentally different computational circuits. Following this, Stander et al. (2024) failed to reproduce Chughtai et al. (2023a)'s findings, instead identifying coset circuits as the structures underlying models trained on permutation group tasks. Consequently, the initial claims of universality were undermined—modular addition appeared to admit two incompatible circuit interpretations, neither of which matched the coset-based mechanism observed for permutation groups, a closely related setting.

More recently, these inconsistencies were resolved. McCracken et al. (2025) provided a rigorous theoretical demonstration that deep neural networks trained on modular addition universally rely on approximate coset structures. Shortly thereafter, Moisescu-Pareja et al. (2025) were unable to reproduce the findings of Zhong et al. (2023), which had suggested the existence of two distinct circuit types. As a result, the Universality Hypothesis regained traction—empirical and theoretical evidence converged to show that models performing group multiplication tasks employ generalized (or approximate) coset representations, effectively unifying the competing interpretations.

Concurrently, substantial theoretical progress was achieved using cyclic group multiplication as a framework. Gromov (2023) derived an analytic characterization of cross-entropy minimization in networks with quadratic activation functions. Building on this, Lyu et al. (2023) demonstrated that smoothness can act as an inductive bias capable of driving grokking phenomena. Subsequently, Morwani et al. (2024) proved that single-layer networks require $\mathcal{O}(n)$ features to represent these tasks, and proposed that the emergence of sinusoidal frequency components during training results from an implicit objective to maximize the margin between the correct and the next-highest logit—an idea

also independently advanced by Mohamadi et al. (2023). Finally, McCracken et al. (2025) delivered the first formal proof that neural networks are capable of learning divide-and-conquer algorithms, establishing an expected feature efficiency of $\Theta(\log(n))$ and showing through experiments that this theoretical bound aligns closely with empirical observations.

# 4 RESULTS

The main text will focus primarily on $D_{18}$ and focus on two specific neural representations that divide $D_{18}$ into simpler subproblems. The reason for this is to familiarize the reader with the two possible cases of what can be learned. Case 1 occurs when neurons activate on cosets—this means the sinusoidal frequency of the neuron induces equivalence class structure on the group. In this case, many points in the dataset of size $(2n)^2$ become equivalent, and the dataset is easier to study. To represent this case, we choose $D_3$, arising from neurons learning frequency 6, because it breaks the $2n = 36$ group elements of $D_{18}$ down into six hexagrams of size 6. Case 2 occurs when neurons activate on approximate cosets, this means the frequency of the neurons doesn't divide $2n$. Thus, such a neural representation will not break $D_{18}$ down into simpler subproblems. Appendices B.6.2 and B.6.3 follow the paper, showing every possible type of neuron, and the Cayley graph corresponding to their neural representation. Additionally, the main text will focus on models that learn good local minima, but we also study models learning poor local minima in appendix B.9.

We investigate the dihedral group on 36 elements, $D_{18}$. Let elements 0 to 17 be rotations $r^k$ and elements 18 to 35 be reflections $sr^k$; neuron preactivation plots will therefore have a "disconnect" between elements 17 and 18. Note: rotations $r^k$ are located in the sign +1 region of the Cayley graph and reflections $sr^k$ are located in the sign -1 region of the Cayley graph.

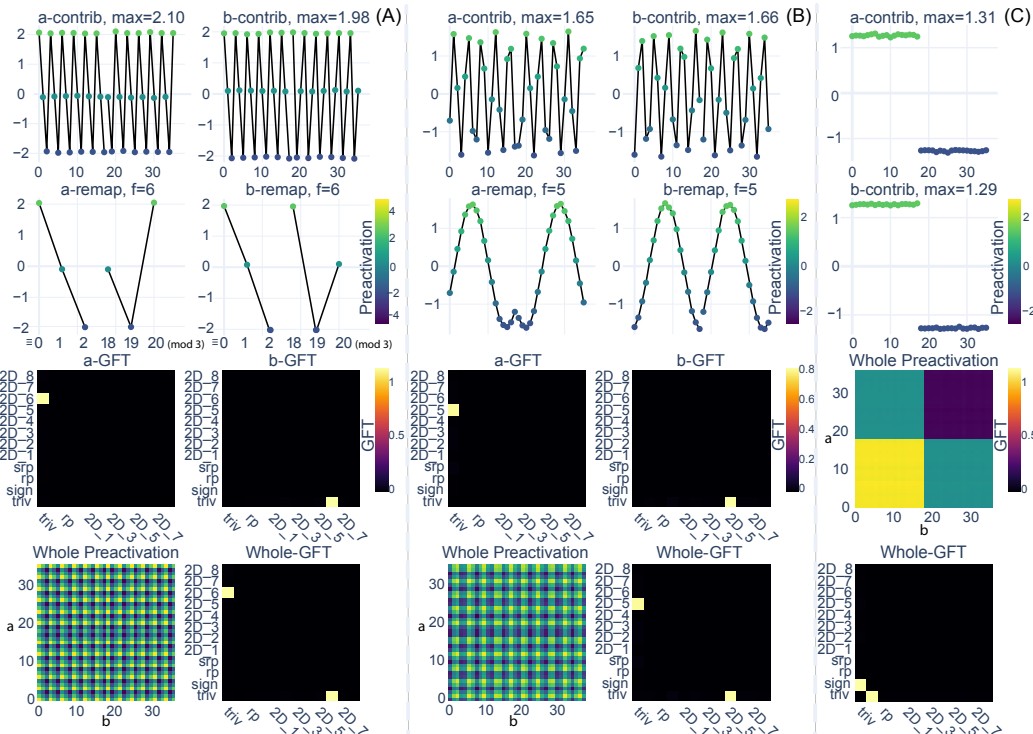

Figure 2: (A) a neuron learning frequency 6, corresponding to learning precise cosets. Remapping collapses the points in row 1 onto six cosets of $D_{18}$ in row 2; three are in rotation class ($x < 18$): $x \equiv 0, 1, 2 \pmod 3$, and three are in reflection class ($x \geq 18$): $x \equiv 18, 19, 20 \pmod 3$, $x \in \{a, b\}$. (B) The same plots are shown for a neuron learning frequency 5, corresponding to learning approximate cosets—there are no precise equivalence classes. (C) Plots for a neuron learning the sign +1 coset are shown. This neuron effectively acts as an indicator for whether the answer $C$ is in the sign +1 part of the graph.

With this in mind, see Figure 2, showing the preactivations of neurons and their group Fourier transforms (GFT). The preactivations are split into the contribution to the neurons' preactivation

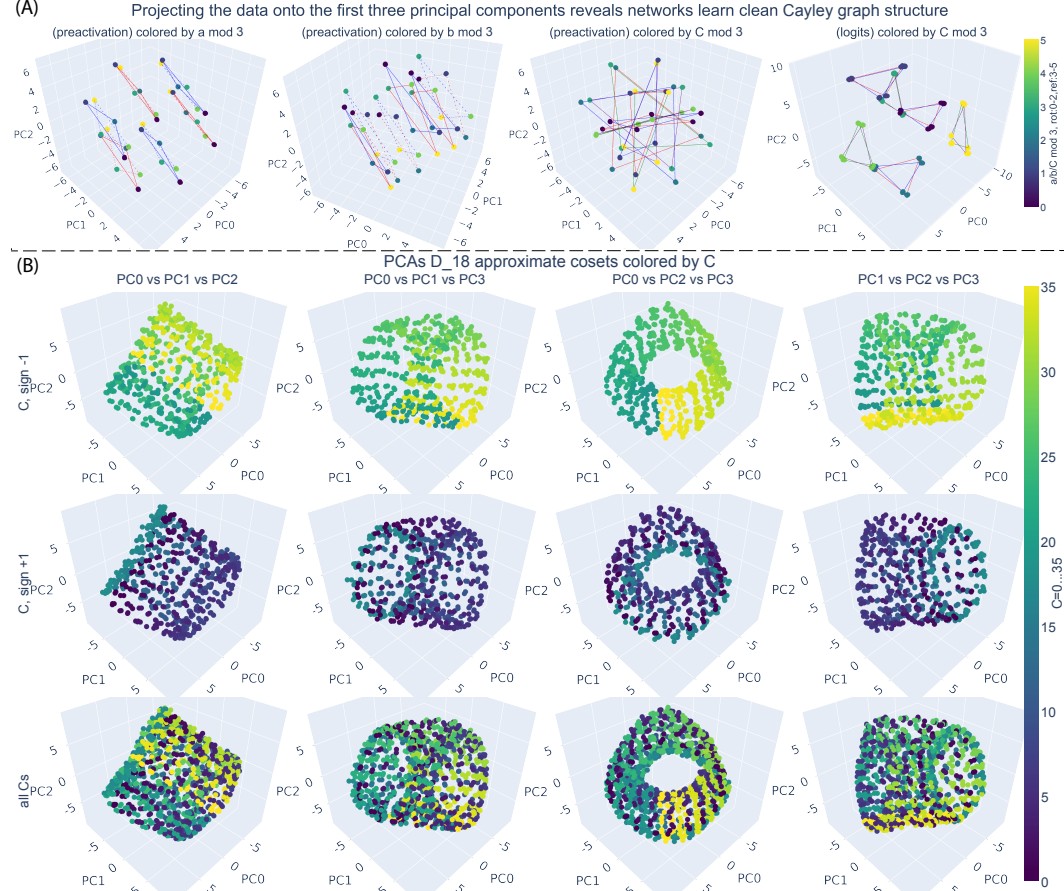

Figure 3: (A) PCA of the preactivations of the frequency-6 neuron cluster shows that the network has cleanly organized the six $D_3$ cosets. Coloring by the coset of $a$ or $b$ produces six "hexagrams", while coloring by the coset of $C$ only becomes clearly separated once we apply PCA to the cluster's contribution to the logits rather than to its preactivations. In this logits space we recover six well-separated cosets–three corresponding to answers with sign $+1$ and three with sign $-1$. (B) PCAs of an approximate coset cluster's contributions to the logits, with frequency 1, reveal that the network stores information in sign $+1$ ((B) row 1) orthogonally to information in sign -1 ((B) row 2); look at points of constant color in row 1 vs. row 2. The underlying structure is effectively four-dimensional, with the cumulative explained variance reaching 0.9987 by the fourth principal component. Color and line styles encode rotation vs. reflection cosets for $a$, $b$ and $C$; see Appendix B.1 for a precise description of the coloring and line rules and the construction of the logits-contribution PCA.

coming from just $a$ (a-contrib), and just $b$ (b-contrib). This split is done to emphasize that neurons activate on the cosets of $a$ and $b$. The GFTs tell us which Fourier basis the preactivations concentrate on. See *e.g.* the neuron learning frequency 6 in Fig. 2 (A), concentrating on the $2D_6$ Fourier basis. For $k \in \{0, 1, 2\}$, let coset $R_k = \{r^{3t+k}\}_{t=0}^5$, $S_k = \{sr^{3t+k}\}_{t=0}^5$. Here, for $x \in \{a, b\}$, the neuron is constant on $R_k \cup S_{\sigma_x(k)}$ with value $A_k \in \{2, 0, -2\}$, where $\sigma_a(k) = k - 1$, $\sigma_b(k) = -k$ (mod 3).

After remapping to frequency 1, each coset $R_k$ and $S_k$ collapses to a single representative. E.g., $k = 1$, $x = a \Rightarrow \sigma_a(1) = 0$: the neuron is constant on $R_1 \cup S_0 = \{r^1, r^4, \ldots, r^{16}\} \cup \{s, sr^3, \ldots, sr^{15}\}$; after remapping, $R_1$ and $S_0$ each collapses to a single representative. Compare this to the neuron learning frequency 5 in (B): when normalized to frequency 1 it has no collapse. This occurs because the neuron's frequency has $\gcd(5, 18) = 1$ and thus, the neuron can not divide $D_{18}$ evenly into substructures. This is the definition of an approximate coset, which occur when frequency $f$ has $\gcd(f, n) = 1$. Thus, it could be hypothesized that such a neuron has learned an Cayley graph representation for $D_{18}$ involving 36 unique elements. This comes from the fact that cosets would collapse the Cayley graph to a substructure. The final column of Fig. 2 shows a neuron that learned the sign $+1$ coset—this neuron only activates when the sign of $a$ and $b$ is 1, *i.e.* $a$ and $b$ are both in

the rotations part of the Cayley graph. This acts as an indicator for the half of the Cayley graph the answer $C$ is in.

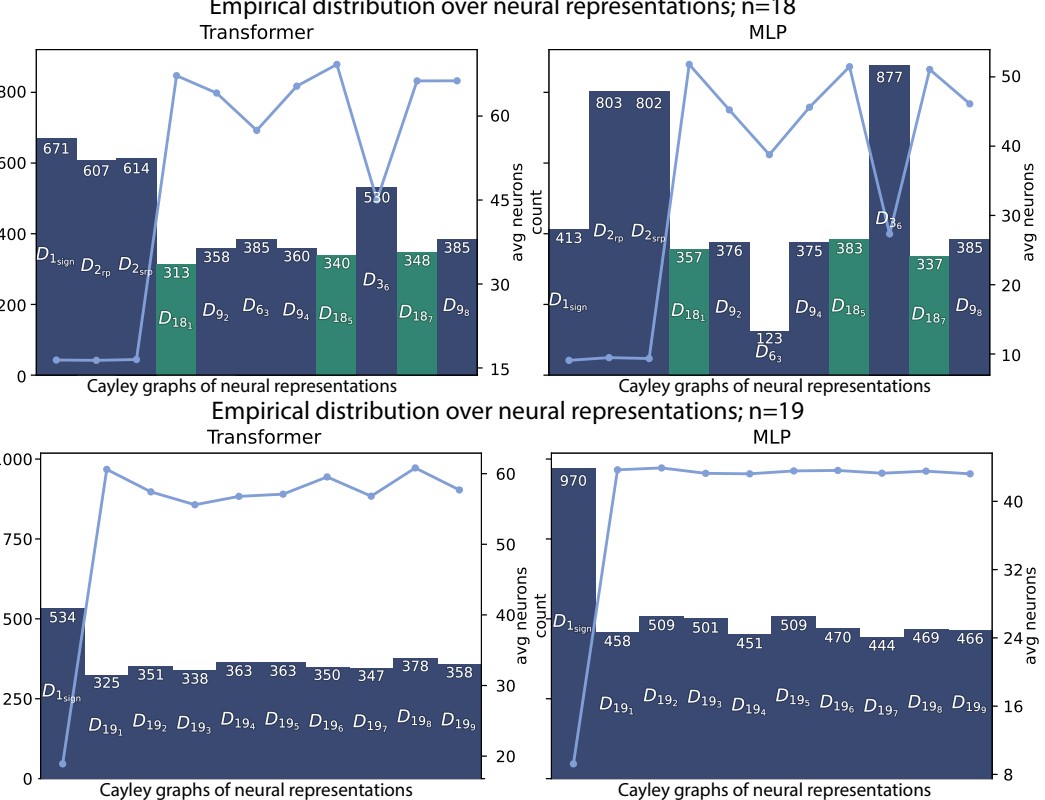

Figure 4: We study both transformers and MLPs over 1000 random seeds on $D_{18}$ (full of coset structure) and $D_{19}$ (no non-trivial coset structure). **Barplot.** We find that transformers learn fewer neural representations on average (this is shown over many orders of magnitude of $D_n$ later in Figure 6). Additionally, transformers are closer to a uniform distribution over the learned Cayley graph representations, while MLP models seem to avoid $D_{6_3}$. **Line plot.** On average transformers use slightly more neurons than MLPs to learn the same Cayley graph representations.

We cluster neurons by identifying all units that activate on the same Fourier basis with the GFT; for each neuron in the cluster we build a $2n \times 2n$ matrix whose entry $(a, b)$ is the neuron's preactivation on datum $(a, b)$, flatten each matrix, and stack the resulting vectors to form a $|\text{cluster} f| \times (2n)^2$ "cluster of preactivations" matrix. We then perform principal component analysis (PCA) on this matrix of neuron preactivations and project all $(2n)^2$ data $(a, b)$ onto the principal components (PCs). When true cosets are learned, data in the same joint equivalence class—*e.g.*, all points with $(a \equiv 0 \pmod 3, \ b \equiv 0 \pmod 3)$ collapse to the same coordinate. For example, in $D_{18}$ there are $36^2 = 1296$ points, but for neurons learning frequency 6 (since $\gcd(6, 18) = 6 \neq 1$) only 36 points are plotted (Figure 3 (A)). These 36 points correspond to the 36 joint equivalence classes. For each fixed $a$, there are 6 points determined by $b$: three with $b < 18$ (rotations) and $b \equiv 0, 1, 2 \pmod 3$ and three with $b \geq 18$ (reflections) and $b \equiv 18, 19, 20 \pmod 3$; *e.g.*, for $a \equiv 0 \pmod 3$ one may take $b \in \{0, 1, 2\}$ and $b \in \{18, 19, 20\}$ as representatives. By contrast, when frequency 1 is learned (Figure 3 (B)), $\gcd(1, 18) = 1$ and each of the 1296 data points projects to its own coordinate. Figure 3 only shows cluster contributions to the logits for frequency 1 to emphasize that networks store the answer $C$ orthogonally, depending on the sign of the location of the answer.

To quantify what's shown in the PCAs of Fig. 3, we train 1000 MLPs and transformers on $D_{18}$ and $D_{19}$, and record the rate of occurrence of the different Cayley graphs (Fig. 4). On $D_{18}$, we see MLPs have a strong preference for learning cosets, corresponding to neural representations that factor $D_{18}$ into a smaller problem, for example MLPs learn $D_{3_6}$ in 87.7% of the runs. To contrast,

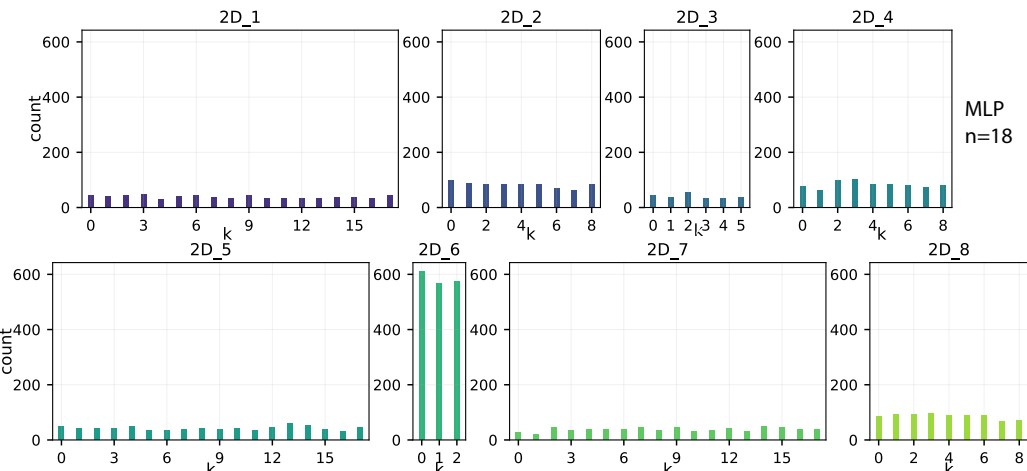

Figure 5: Histograms of the number of times each generator ($sr^k$) was learned for each type of Cayley graph across 1000 seeds. The distributions are all approximately uniform. This matches our intuitions from group theory that there are no benefits to learning specific generators for the same Cayley graph. Networks seem to just want to learn the Cayley graph, and don't prefer particular generators to be used.

transformers seem to learn fewer Cayley graphs, while dedicating more total neurons to each Cayley graph representation. Additionally, we inspect whether some Cayley graphs are more preferred than others, and find that all Cayley graphs are approximately uniformly likely to be learned (Fig. 5). The plot for the transformer's generator preferences is also uniform, but is in the appendix. We also explore how many Cayley graphs need to be learned as a function of the size $n$ of the dihedral group. We find that both transformers and MLPs learn a logarithmic number of Cayley graphs (Fig. 6).

## 5 THE ALGORITHM DNNS LEARN

The cyclic group study showed that neural networks implement an approximate CRT by decomposing modular addition into $\mathcal{O}(\log n)$ subsystems, each corresponding to an approximate coset structure, and then intersecting these cosets to isolate the correct output (McCracken et al., 2025). In the dihedral group $D_n$, an analogous phenomenon occurs: networks instantiate an abstract divide-and-conquer algorithm using Cayley-graph neural representations. Individual neurons indeed learn to approximate coset structure, but the structure distributed over a cluster of neurons activating on the same cosets is a Cayley graph.

**Remark 1.** *The sinusoidal neuron based Cayley-graph decomposition. The dihedral group $D_n = \langle r, s \mid r^n = e, s^2 = e, srs = r^{-1} \rangle$ has two types of generators: rotations $r$ and reflections $s$. Multiplication splits into four quadrants depending on the pair of input types (rotation or reflection). Empirically, neurons in the each layer specialize to sinusoidal features whose preactivations cluster along Cayley-graph structure for either one or two generators. Principal component analysis of these clusters reveals interpretable graphs in which the generators $\{r\}$ or $\{r, s\}$ are directly visible. Thus, each cluster of neurons corresponds to a distinct Cayley graph, implementing one simpler "subproblem" of the global group operation.*

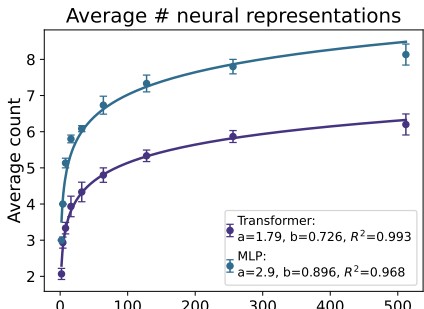

Figure 6: To measure the feature efficiency of different architectures, we train transformers and MLPs over many orders of magnitude (from $D_2$ to $D_{512}$. We observe that transformers learn the dihedral group with significantly fewer neural representations being found in the trained models. Additionally, both architectures are well fit by logarithmic curves, providing empirical support that a highly representation-efficient divide-and-conquer algorithm is learned.

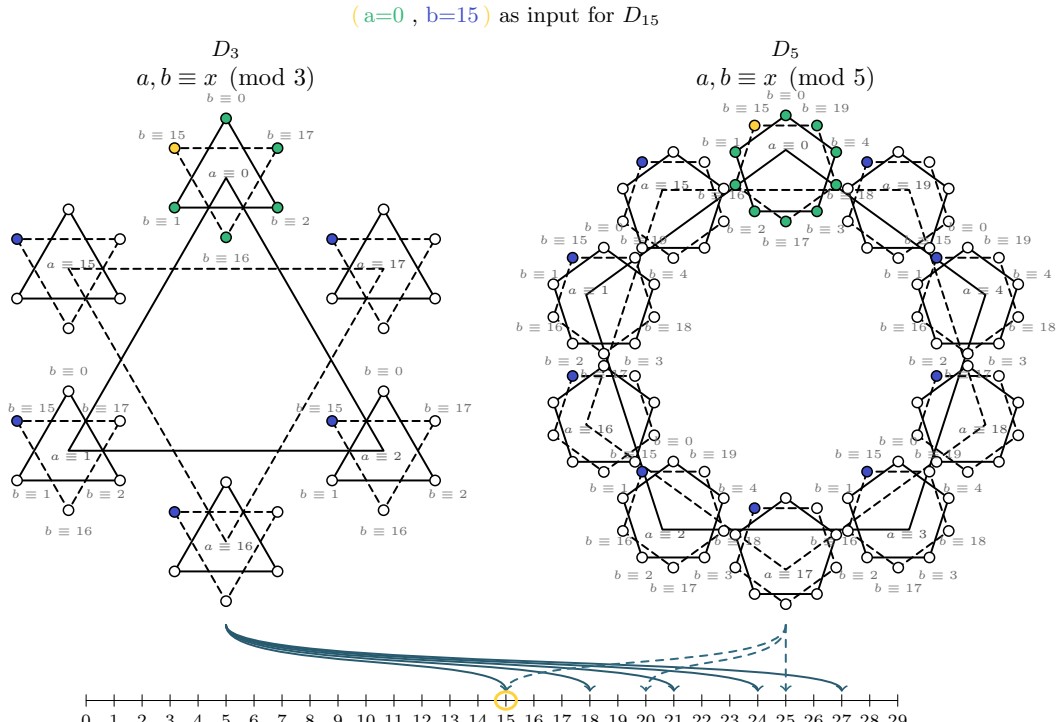

Figure 7: *The algorithm learned by DNNs trained on dihedral group multiplication.* This example shows how a DNN learned to break down $D_{15}$ into simpler subproblems. Two neural representations were learned, one being six $D_3$ subgroups, and the other being ten $D_5$ subgroups. The visualization shows how the correct answer (logit 15) is maximized due to the linear combination from the last layer to the output logits from the neural representations. The neural representation for $D_3$ subgroups pushes mass onto all logits where $a \equiv 0 \pmod 3$ and $b \equiv 15 \pmod 3$. The structure of the neural representation is such that the network stores information about the cosets of $a$ and the cosets of $b$ independently, narrowing down the cosets containing $c$. The $D_3$ neural representation outputs maximum mass onto reflection elements $15, 18, 21, 24, 27$. In other words, it has narrowed down the membership of $c$ to these four elements, all being in the reflected halve of the $D_{15}$. The $D_5$ neural representation outputs maximum mass onto reflection elements $15, 20, 25$. The intersection of these sets is $15$, and thus $15$ has a higher logit value than any other logit, and the correct answer is selected by argmax. Naturally, this is a simplification, and usually there are more Cayley graphs due to the network learning $\mathcal{O}(\log(n))$ neural representations instead of just 2. **Note:** if the multiplication is extended from $ab$ to $abc$, the small vertices corresponding to the equivalence classes of $b$ would contain yet another recursive structure. Thus, the computation can be thought of conceptually like a fractal: every neural representation computes the equivalence class of $a, b, c, \dots$. Inside each vertex of the $D_n$ representation would be the Cayley graph with vertices labeled by the equivalence classes of the next input token.

The analogy with the cyclic case is as follows. In the Chinese Remainder Theorem viewpoint, $\mathcal{O}(\log n)$ frequencies suffice to decompose $(a + b) \bmod n$ into manageable subproblems, each enforcing approximate coset constraints. In the dihedral case, $\mathcal{O}(\log n)$ Cayley-graph representations suffice: each representation handles an equivalence class of inputs $(a, b)$ with respect to one generator set, and outputs strongly on the corresponding class of candidate answers. Some representations rely only on rotation ($\{r\}$), which induces a front/back ambiguity, while others rely on both rotation and reflection ($\{r, s\}$), which resolves the ambiguity and corresponds to "better" local minima since such a representation can narrow down the answer $C$ to less possible group elements.

Thus, the dihedral case instantiates the same abstract strategy as the cyclic case: neural networks universally decompose the group operation into $\mathcal{O}(\log n)$ structured subproblems. The difference is that the structure is now governed by Cayley graphs rather than cosets alone. Our empirical evidence (Section 4) shows that across seeds and architectures, neuron clusters reliably align to these Cayley

*Informal algorithm.* Take $\mathcal{O}(\log n)$ neuron clusters, each realizing a Cayley graph $\mathcal{C}_k$ generated by either $\{r\}$ or $\{r, s\}$. For each cluster, use sinusoidal phases to separate input equivalence classes according to $\mathcal{C}_k$. Then sum across clusters: the correct output logit lies in the intersection of all equivalence classes and is maximized as each neural representation contributes positive mass to members of the correct logits equivalence class, while incorrect logits receive negative mass.

---

**Algorithm 1** Implicit Template: DNNs Divide-and-Conquer via Cayley-Graph Representations

---

**Require:** Trained network $\mathcal{N}$; task domain with size $n$; dihedral group $D_n$ with generators $r$ (rotation), $sr$ (reflection) so $|D_n| = 2n$.
**Ensure:** A set of neural representations $\mathcal{R}$ with $|\mathcal{R}| = \Theta(\log n)$, realized inside $\mathcal{N}$.
1: $\mathcal{R} \leftarrow \emptyset$; $\quad K \leftarrow \lceil \log n \rceil$ $\qquad\qquad$ ▷ number of distinct representations realized on average
2: **for** $k = 1$ to $K$ **do**
3: $\qquad$ **Representation** $k$**:** $\mathcal{N}$ realizes a Cayley-graph–structured code

$$\mathcal{C}_k = \text{CAYLEYGRAPH}(D_n, \mathcal{G}_k), \quad \mathcal{G}_k \in \big\{\{r\}, \{r, s\}\big\}.$$

4: $\qquad$ **Generator pattern:**
$\qquad\quad$ Either a single rotational generator ($\mathcal{G}_k = \{r\}$), *or* a rotational+reflective pair ($\mathcal{G}_k = \{r, s\}$).
5: $\qquad$ Record $(\mathcal{C}_k, \mathcal{G}_k)$ in $\mathcal{R}$.
6: **return** $\mathcal{R}$ containing each neural representation, which when intersected, uniquely identify $c$.

---

graphs, providing strong evidence that Algorithm 1 is learned universally across seeds, and in both MLPs and transformers.

We claim our evidence supports that all neural networks universally learn different implementations of the following abstract algorithm. Indeed, over all experiments we trained, the networks were found to use the following algorithm.

# 6 CONCLUSION

Our results extend understanding of the universality hypothesis and its relationship to broader hypotheses about what deep networks learn by serving as the second group multiplication to be reverse engineered at multiple levels of abstraction, and the first work to fully characterize and determine the structures corresponding to the neural representations learned universally across architectures learning dihedral multiplication. Specifically, we find Cayley graphs embedded on the surface of manifolds correspond to each neural representation learned by trained models. We find: 1) neuron preactivations are on coset or approximate coset structure 2) neural representations act to divide the data manifold into different subproblems, being Cayley graphs, where the answer is computed. 3) the global algorithm merges the computational results of each neural representation with the correct output logit being maximized via an error correcting code. Because different representations all output their maximum values on the equivalence class of the correct logit, the sum over representations results in the correct logit being maximized and incorrect logits having their values pushed down exponentially after softmax.

This picture suggests a candidate connection between three influential hypotheses about deep learning. (i) The first is the universality hypothesis, that DNNs trained on similar data will make use of similar principles (Li et al., 2015; Olah et al., 2020). McCracken et al. (2025)'s conjecture only relates to this hypothesis. Our work however, sheds light into universality and the nature of two more: (ii) the manifold hypothesis, that DNNs efficiently learn (despite the curse of dimensionality typically applying to such high dimensional objects) by finding low dimensional manifolds that the dataset lives on (Goodfellow et al., 2016) and (iii) the platonic representation hypothesis, which posits architectures trained on different learning objectives at large scales, learn a platonic representation, meaning distances between datapoints in different classes of data "converge" between models (Huh et al., 2024). One of our novelties is that our methodology can be used to falsifiably study these latter two hypotheses on group multiplications.

**Limitations.** Our methodology relies on the fact we can use the Group Fourier Transform to identify neuron behaviors. After using it, we gain the ability to form the neuron-clusters and study the neural

representations qualitatively and quantitatively with great precision. For the time being however, there is no such method available for use on natural datasets. That said, part of the goal of studying toy models is to hopefully, once studying enough of them, gain enough insight into how to DNNs work to be able to generalize the community's collective insights onto natural datasets.

## 6.1 FUTURE WORK

Our work thus refined and extended the conjecture of McCracken et al. (2025) by proposing explicit structure, being that Cayley graph manifolds correspond to the neural representations being learned. We conjecture: (i) universality will be found as DNNs of different architectures learning divide-and-conquer algorithms to fit group tasks (McCracken et al., 2025). (ii) The structures that correspond to neural representations will universally be subproblems of the group, being Cayley graphs embedded geometrically as manifolds (with simpler subproblems (subgroups) preferred). (iii) Finally, Individual neurons will universally learn coset structure (McCracken et al., 2025).

This gives three independent levels of abstraction where universality may be falsified by studying DNNs learning group multiplications.

1. Globally: what are the solutions learned by DNNs? Can work find DNNs not using divide-and-conquer algorithmic structure?
2. Structurally: what are the neural representations? Can work find they're not always Cayley graphs?
3. Neuron-wise: what are the activations? Can work find they're not always cosets?

Should universality be found to be true at all three of these levels over a diverse breadth of group multiplication tasks, we believe this would give substantial insight into the nature of universality. Furthermore, such a result would provide insight into the manifold hypothesis, as the Cayley graph neural representations are embedded on the surfaces of manifolds. Finally, our study introduces a methodology that can study how the distances between datapoints in cosets on the Cayley graph converge in 11, This computation can be used by future work to quantitatively verify Cayley graph structure is learned.

Thus, our work offers the ability to inspect these three major hypotheses simultaneously. Should this conjecture be verified across a diverse breadth of group tasks, it will show that "shared principles" DNNs learn on similar data are universal at every level of abstraction. This would suggest there are universality classes of particular manifolds that are learned by DNNs trained by stochastic gradient methods.

A natural place for future work to start investigating our conjecture and McCracken et al. (2025)'s conjecture is on groups belonging to a different class of finite simple group. We believe the alternating and permutation groups should be revisited, as well as some simple groups of Lie type and at least one sporadic group. It's likely that investigating even the smallest sporadic group will require significant compute resources. It will be interesting to see if the theoretical interpretability community can get to the point of being able to predict what the weights of neural networks will before training, under the assumption that the neural representations will be the Cayley graph manifolds we propose.

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

| $g\backslash h$ | Rotations | | | Reflections | | |
|---|---|---|---|---|---|---|
| | $e$ | $r$ | $r^2$ | $s$ | $sr$ | $sr^2$ |
| $e$ | $e$ | $r$ | $r^2$ | $s$ | $sr$ | $sr^2$ |
| $r$ | $r$ | $r^2$ | $e$ | $sr^2$ | $s$ | $sr$ |
| $r^2$ | $r^2$ | $e$ | $r$ | $sr$ | $sr^2$ | $s$ |
| $s$ | $s$ | $sr$ | $sr^2$ | $e$ | $r$ | $r^2$ |
| $sr$ | $sr$ | $sr^2$ | $s$ | $r^2$ | $e$ | $r$ |
| $sr^2$ | $sr^2$ | $s$ | $sr$ | $r$ | $r^2$ | $e$ |

Entries are $g \cdot h$ (left multiplication).
$r^i r^j = r^{(i+j) \bmod 3}$,
$r^i (sr^j) = sr^{(j-i) \bmod 3}$,
$(sr^i) r^j = sr^{(i+j) \bmod 3}$,
$(sr^i)(sr^j) = r^{(j-i) \bmod 3}$

Figure 8: Cayley table for $D_3 (= S_3)$.

# A APPENDIX

## A.1 EXTRA BACKGROUND

### A.1.1 DIHEDRAL GROUP MULTIPLICATION TABLE

See Table 8.

### A.1.2 IRREDUCIBLE REPRESENTATIONS AND GROUP FOURIER TRANSFORM

**Representation and irreducible representation.** Let $G$ be a group. A (finite-dimensional) representation of $G$ is a homomorphism

$$\rho : G \to \mathrm{GL}(V),$$

where $V$ is a complex vector space. We say that $\rho$ (or $V$) is *irreducible* if $V$ has no nontrivial $G$-invariant subspaces, i.e., the only subspaces $W \subseteq V$ with $\rho(g)W \subseteq W$ for all $g \in G$ are $W = \{0\}$ and $W = V$. A complete set of pairwise inequivalent irreducible representations is denoted by $\widehat{G}$.

**Group Fourier transform.** Let $G$ be a finite group and let $\widehat{G}$ denote a complete set of unitary irreducible representations (irreps) $\rho : G \to \mathrm{U}(d_\rho)$. For a function $f : G \to \mathbb{C}$, we define the (matrix-valued) *group Fourier transform* by

$$\widehat{f}(\rho) \;=\; \frac{1}{|G|} \sum_{g \in G} f(g)\, \rho(g)^\dagger \;\in\; \mathbb{C}^{d_\rho \times d_\rho}, \tag{1}$$

and the inverse transform by

$$f(g) \;=\; \sum_{\rho \in \widehat{G}} d_\rho \, \mathrm{Tr}\Big( \rho(g)\, \widehat{f}(\rho) \Big). \tag{2}$$

**Schur orthogonality (matrix elements).** For irreducible representations $\rho^\lambda$ and $\rho^\mu$ of dimensions $d_\lambda$ and $d_\mu$,

$$\frac{1}{|G|} \sum_{g \in G} \rho_{ij}^\lambda(g) \, \overline{\rho_{i'j'}^\mu(g)} \;=\; \frac{\delta_{\lambda\mu}\, \delta_{ii'}\, \delta_{jj'}}{d_\lambda}. \tag{3}$$

With the normalization in equation 1, the corresponding Parseval/Plancherel identity is

$$\sum_{g \in G} |f(g)|^2 \;=\; \sum_{\rho \in \widehat{G}} d_\rho \, \|\widehat{f}(\rho)\|_F^2. \tag{4}$$

**Product groups.** For $F : G \times G \to \mathbb{C}$ we use the tensor-product of irreducible representations $\rho \otimes \sigma$ of dimension $d_\rho d_\sigma$. The two-axis Fourier transform and its inverse are

$$\widehat{F}(\rho, \sigma) \;=\; \frac{1}{|G|^2} \sum_{(g_1,g_2) \in G \times G} F(g_1, g_2) \left( \rho(g_1)^\dagger \otimes \sigma(g_2)^\dagger \right), \tag{5}$$

$$F(g_1, g_2) \;=\; \sum_{\rho, \sigma} d_\rho d_\sigma \, \mathrm{Tr}\!\left( \left( \rho(g_1) \otimes \sigma(g_2) \right) \widehat{F}(\rho, \sigma) \right). \tag{6}$$

For this convention, the Parseval identity takes the form

$$\sum_{g_1, g_2 \in G} |F(g_1, g_2)|^2 \;=\; \sum_{\rho, \sigma} d_\rho d_\sigma \, \|\widehat{F}(\rho, \sigma)\|_F^2. \tag{7}$$

### A.1.3 Irreducible Representations of the Dihedral Group $D_n$

We use the presentation

$$D_n \;=\; \langle r, s \mid r^n = e, \; s^2 = e, \; srs = r^{-1} \rangle.$$

**One-dimensional irreducible representations.** For all $n$, there are the trivial and sign characters:

$$\chi_{\mathrm{triv}}(r^m) = 1, \quad \chi_{\mathrm{triv}}(sr^m) = 1; \qquad \chi_{\mathrm{sign}}(r^m) = 1, \quad \chi_{\mathrm{sign}}(sr^m) = -1.$$

If $n$ is *even*, there are two additional one-dimensional irreps obtained by sending $r \mapsto -1$ and $s \mapsto \pm 1$. Explicitly, for $m \in \{0, \ldots, n-1\}$,

$$\chi_{rp}(r^m) = (-1)^m, \quad \chi_{rp}(sr^m) = (-1)^m,$$
$$\chi_{srp}(r^m) = (-1)^m, \quad \chi_{srp}(sr^m) = -(-1)^m.$$

**Two-dimensional irreducible representations ($2D_k$).** For $k = 1, \ldots, \lfloor (n-1)/2 \rfloor$, define

$$R_k(r^m) = \begin{pmatrix} \cos \frac{2\pi km}{n} & -\sin \frac{2\pi km}{n} \\ \sin \frac{2\pi km}{n} & \cos \frac{2\pi km}{n} \end{pmatrix}, \qquad R_k(sr^m) = \begin{pmatrix} 1 & 0 \\ 0 & -1 \end{pmatrix} R_k(r^m). \tag{8}$$

These are real orthogonal (hence unitary) representations and satisfy the defining relation $srs = r^{-1}$. Substituting these matrices into the general formulas equation 1–equation 6 yields the transforms used in our implementation.

### A.1.4 Approximate cosets

Approximate cosets are intuitively, the generalization of cosets to "almost a coset". They arise when neurons in a network learn to divide a group using a structure that doesn't actually divide the group. For example, $D_{18}$ has 36 elements, 18 of which are arranged in a circle in the front, and 18 are "in the back" as a mirror reflection. Thus, it's the case that we could choose to divide the 36 elements by 6, giving us 6 sets, and since there are 18 elements in the front and 18 in the mirror reflection, we also divide 18 by 6 = 3. This tells us that our division of the group into 6 sets, will give us smaller sets, where 3 elements are in the front and three are in the back.

Suppose alternatively that we were trying to divide $D_{18}$ by 5. Since the $\gcd(5, 18) = 1$, 5 doesn't factorize 18 into anything smaller. Thus, there are no cosets, and resultantly, a neuron learning frequency 5 has learned the full group structure of $D_{18}$. Such a neuron has 5 peaks (maximum values), and if a peak is located at $a$, the next peaks are located $a \pm \frac{18}{5} = 3.6$. Naturally, because the problem is discrete, this results in every point having a different activation value and can be seen in Figure 2 by contrasting (A) the coset case with (B) the approximate coset case. In (A), the neuron activates with 3 strengths depending on a and 3 strengths depending on $b$, whereas in (B) the neuron activates with 18 different strengths for $a$ and 18 different strengths for $b$.

We offer the reader the following intuition: approximate cosets are simply when a neuron has learned something that doesn't allow it to cleanly divide the Cayley graph into smaller pieces. A natural response is to think "perhaps neurons would prefer to learn things that cleanly divide the group?" and indeed, this is observed later in Figure 4. For the mathematical definition, please see Section 3 and 4 in McCracken et al. (2025).

## B    EXPERIMENTAL SETTINGS

All experiments trained for 5000 epochs using Adam. All experiments were done on CPUs except for the scaling dihedral group size $n$ plot, which required RTX8000 GPUs to make the experiment faster.

If specific hyperparameters aren't listed, MLPs use mlp lr 0.001 wd 0.0001

and transformers learning rate = 0.0005 and L2 weight decay = 0.0000001

### B.1    FIGURE 3

$D_{18}$.

#### B.1.1    COLOR AND LINE CONVENTIONS FOR FIGURE 3

Preactivation PCA and Color Coding by $a$ or $b$

In the preactivation PCA plots of Figure 3 (A), we color each point according to the coset of $a$, $b$, or $C$. For all cases we use a continuous colormap (viridis) split into two halves:

- the lower half of the colormap is used for *rotation* cosets (sign $+1$);

- the upper half is used for *reflection* cosets (sign $-1$).

This makes the rotation vs. reflection structure visible as a change in hue even when plotting a single scalar residue.

Lines When Coloring by $a$

When we color by $a$, we additionally draw line segments to reveal how residues in the same coset are arranged:

- We fix a value of $b$ and order the corresponding points by stepping $a \mapsto a + d \pmod{g}$, where $g = n/\gcd(n, f)$ and $d$ is defined in 2, starting from the smallest residue present in that stripe.

- For a given fixed $b$, we obtain up to four "stripes" depending on whether $a$ and $b$ lie in the rotation block or the reflection block.

- **Blue vs. red** encodes whether $b$ is in the rotation block (blue) or the reflection block (red).

- **Solid vs. dashed** encodes whether $a$ is in the rotation block (solid) or the reflection block (dashed).

- Within each stripe, points are connected in the order given by repeated addition of the step size $d$ modulo $g$, so each polyline traces out a full cycle of residues inside that coset.

Lines When Coloring by $b$

The construction for "color by $b$" is symmetric:

- We fix $a$ and order points by stepping $b \mapsto b + d \pmod{g}$.

- As before, we obtain up to four stripes depending on whether $a$ and $b$ are in rotation block or reflection block, and we connect points in each stripe according to the modular step order.

These color and line conventions are purely for visualization: they do not affect the PCA itself, but make the underlying $D_3$ coset structure and the separation between rotation and reflection components visually apparent.

### B.1.2 CONSTRUCTION OF CLUSTER CONTRIBUTIONS TO THE LOGITS

For each frequency cluster, we define its *cluster contribution to the logits* by isolating the post-activations of the neurons in that cluster at the penultimate layer and propagating only those activations to the logits.

Concretely, let $H_{\text{cluster}} \in \mathbb{R}^{n^2 \times m}$ collect the post-activations of all neurons in the cluster across the $n^2$ input points (rows index inputs, columns index neurons in the cluster), and let $W_{\text{cluster}} \in \mathbb{R}^{m \times n}$ be the slice of the final weight matrix restricted to those neurons. The logit contribution matrix for this cluster is

$$L_{\text{cluster}} = H_{\text{cluster}} W_{\text{cluster}} \in \mathbb{R}^{n^2 \times n}, \tag{9}$$

which has the same shape as the full logit matrix.

All PCA plots of the "cluster's contributions to the logits" in Figure 3 are obtained by running PCA on matrix $L_{\text{cluster}}$.

## B.2 FIGURE 4 & FIGURE 5

90% train test split, $D_{18}$, $D_{19}$.

## B.3 FIGURE 6

90% train test split $D_{2^k}$, with $k \in [1, 9]$

| $n$ | LR | WD |
|---|---|---|
| 2 | 1e-3 | 1e-5 |
| 4 | 5e-4 | 1e-4 |
| 8 | 5e-4 | 1e-4 |
| 16 | 2e-3 | 1e-4 |
| 32 | 2e-4 | 1e-6 |
| 64 | 2e-4 | 1e-7 |
| 128 | 2e-5 | 1e-7 |
| 256 | 2e-5 | 1e-7 |
| 512 | 2e-5 | 1e-7 |

Table 1: Transformer scaling experiments: learning rate (LR) and weight decay (WD) values per $n$.

| $n$ | LR | WD |
|---|---|---|
| 2 | 5e-5 | 1e-4 |
| 4 | 5e-5 | 1e-6 |
| 8 | 1e-3 | 1e-4 |
| 16 | 2e-3 | 1e-4 |
| 32 | 2e-3 | 1e-4 |
| 64 | 5e-4 | 1e-4 |
| 128 | 2e-3 | 1e-5, |
| 256 | 1e-4 | 1e-6 |
| 512 | 1e-5 | 1e-7 |

Table 2: MLP scaling experiments: learning rate (LR) and weight decay (WD) selections per width $n$.

## B.4 HYPERPARAMETER TUNING RESULTS OVER DEPTH, WIDTH, AND ACTIVATION FUNCTIONS

See fig.12, fig.13, fig.14 and fig.15

## B.5 RANDOM MULTIPLICATION VS. DIHEDRAL GROUP MULTIPLICATION

In this section we replace the true dihedral group multiplication by a random multiplication table. We keep the same one-layer architecture as in the main text and train the network either on the true

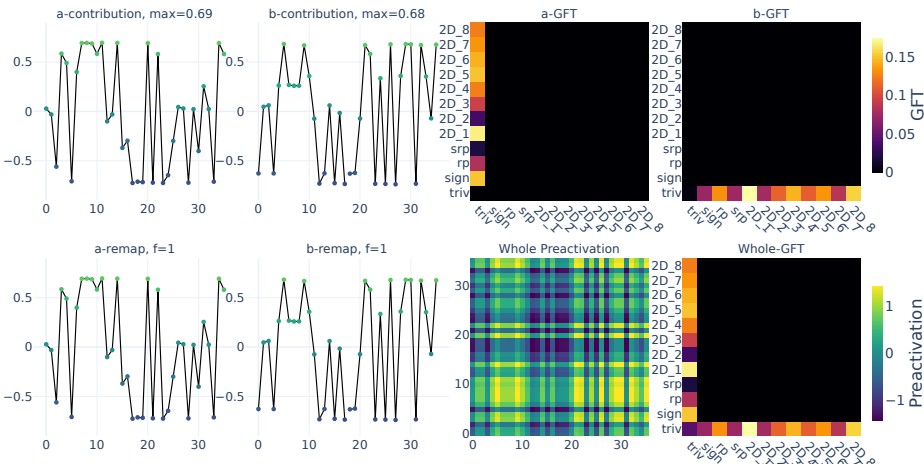

Figure 9: Preactivations of a single neuron under the random multiplication task. Unlike in the dihedral setting, the preactivation does not exhibit structured sinusoidal behavior or concentration on a small number of Fourier modes.

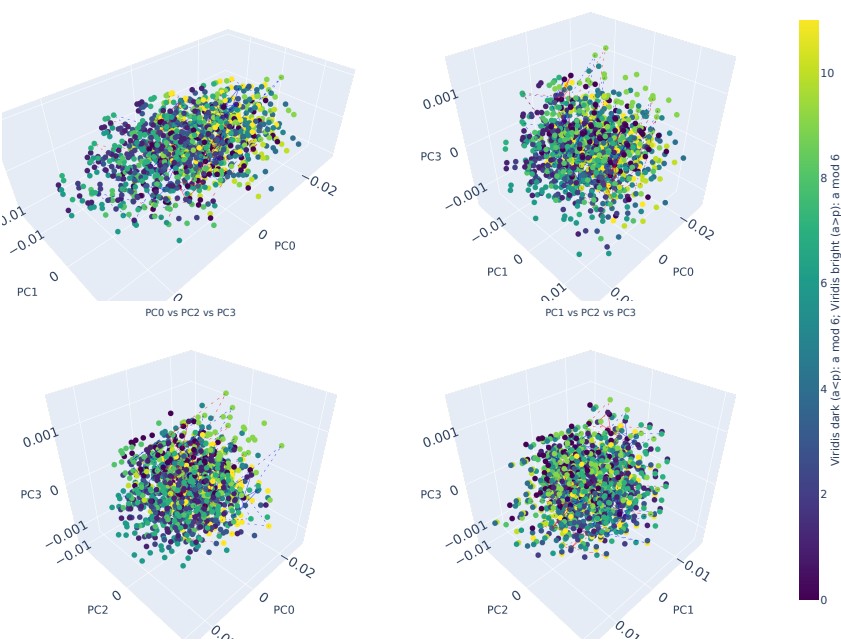

Figure 10: PCA visualization of a representative cluster in the random multiplication task, using the same pipeline as in the main paper. Unlike in the dihedral setting, the points do not form geometrically structured clusters in PCA space, indicating the absence of coherent group-induced feature organization.

dihedral multiplication or on the random multiplication, using the same training hyperparameters and number of seeds. This provides a negative-control setting in which no coherent group structure is expected.

For each neuron, we fit first-order sinusoids to its preactivations (as a function of the group element) and report the resulting average coefficient of determination $R^2$. Higher values indicate that the neuron is well approximated by a small number of Fourier modes.

For the coset-level geometry, we work in the same low-dimensional representation space used in the main paper and compute:

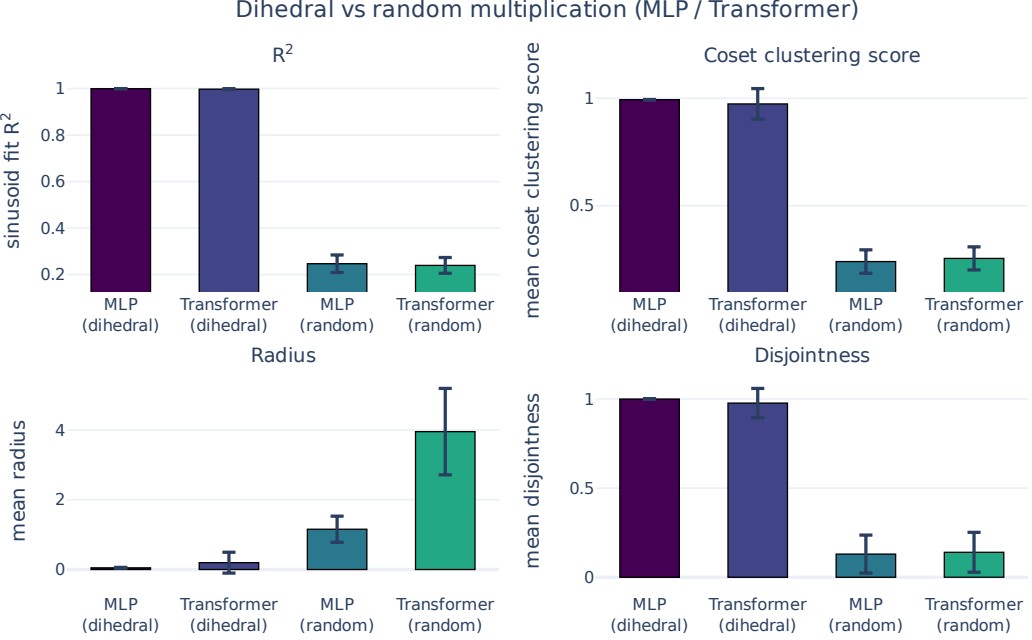

Figure 11: We train one-layer MLPs and one-layer transformers either on the true dihedral multiplication law or on a random multiplication table, using the same training protocol and number of random seeds for each setting. We aggregate the sinusoidal-fit $R^2$ and coset-level geometric statistics over seeds.

1. **Coset clustering score** $s_{\text{cluster}}$: we compute the mean distance between points within the same coset and the mean distance between points in different cosets. We define

$$s_{\text{cluster}} = 1 - \frac{\mathbb{E}[\text{distance between points within the same coset}]}{\mathbb{E}[\text{distance between points across different cosets}]},$$

   so that larger values indicate tighter and better-separated coset clusters.

2. **Coset radius.** For each coset, we compute its minimal enclosing ball in PCA space and record the radius; we report the mean radius across cosets.

3. **Coset disjointness.** For each coset, we compute the fraction of other cosets whose enclosing balls are strictly disjoint from it; we report the mean fraction across cosets.

As shown in Figure 11, for both architectures the true dihedral law yields substantially higher sinusoidal-fit $R^2$ and coset clustering scores, as well as smaller coset radii and larger coset disjointness, than the random-multiplication baseline. In other words, when we destroy the group structure while keeping the architecture, training protocol, and analysis pipeline fixed, both the simple sinusoidal behavior at the neuron preactivation level and the coset-level geometry largely disappear. This negative-control experiment therefore supports the interpretation that the structures we observe in the main paper reflect the learned dihedral multiplication law itself, rather than generic artifacts of the PCA or group Fourier analysis.

## B.6 MORE FIGURES

### B.6.1 ADDITIONAL RESULTS OVER 1000 SEEDS

Here, we show that 2-layer networks also learn Cayley graphs (3 and 16).

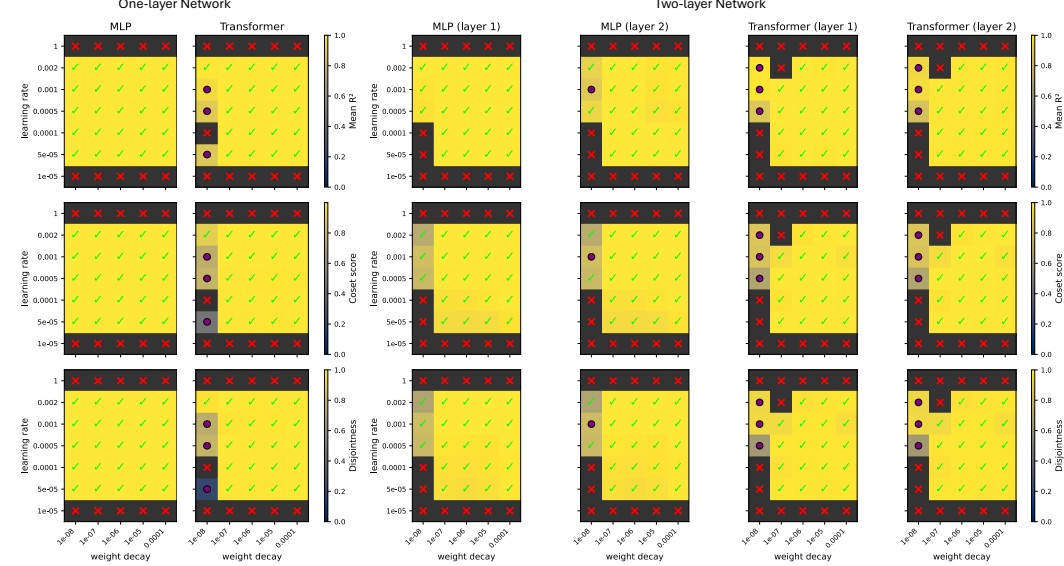

Figure 12: Hyperparameter tuning results for one- and two-layer MLPs and transformers with ReLU as activation function. Each heatmap shows a sweep over learning rate (y-axis) and weight decay (x-axis) for MLPs and transformers. Rows correspond to different metrics: the top row reports the mean coefficient of determination $R^2$ between network preactivations and the sinusoidal fits, the middle row reports the coset clustering score, and the bottom row reports the disjointness metric (formally defined in Section B.5). Colors indicate the metric value averaged over random seeds, we only fill a cell when all seeds reach $100\%$ train and test accuracy during training; runs where any seed fails to generalize to $100\%$ accuracy are marked with a red "×". Among successful runs, a purple "∘" denotes that replacing the neuron preactivations by the sinusoidal fits results in test accuracy $< 99.9999\%$, while a green "✓" denotes replace neuron's activation by the sinusoidal fit results in test accuracy $> 99.9999\%$.

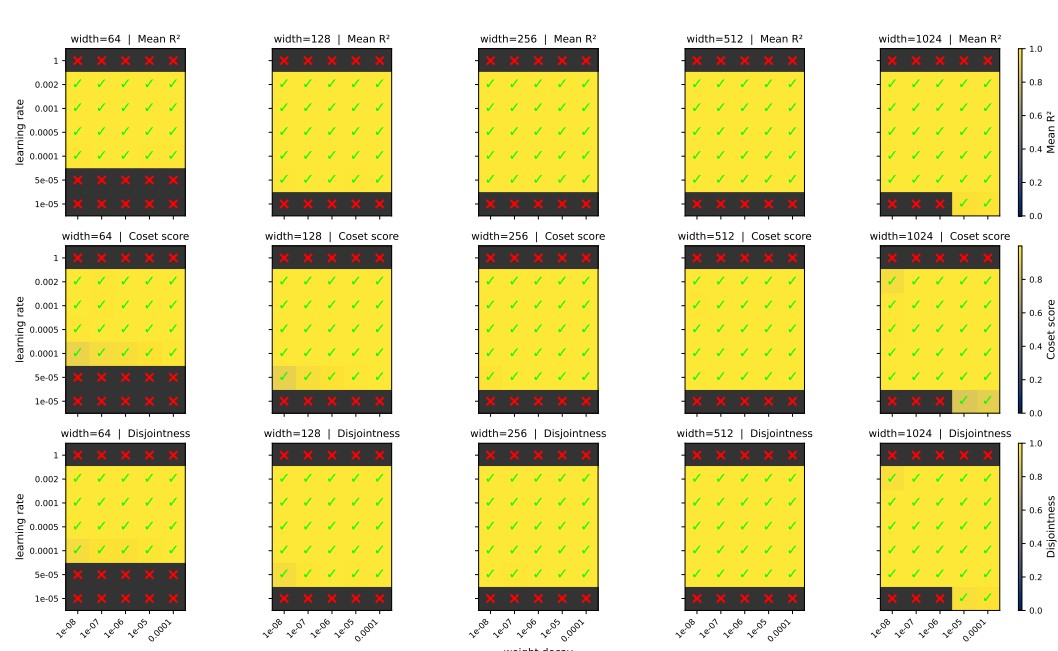

Figure 13: Hyperparameter tuning results over different widths of one-layer MLPs. As in Figure 12, each heatmap shows a sweep over learning rate (y-axis) and weight decay (x-axis), and the color scale and markers (red "×", purple "∘", and green "✓") follow the same conventions.

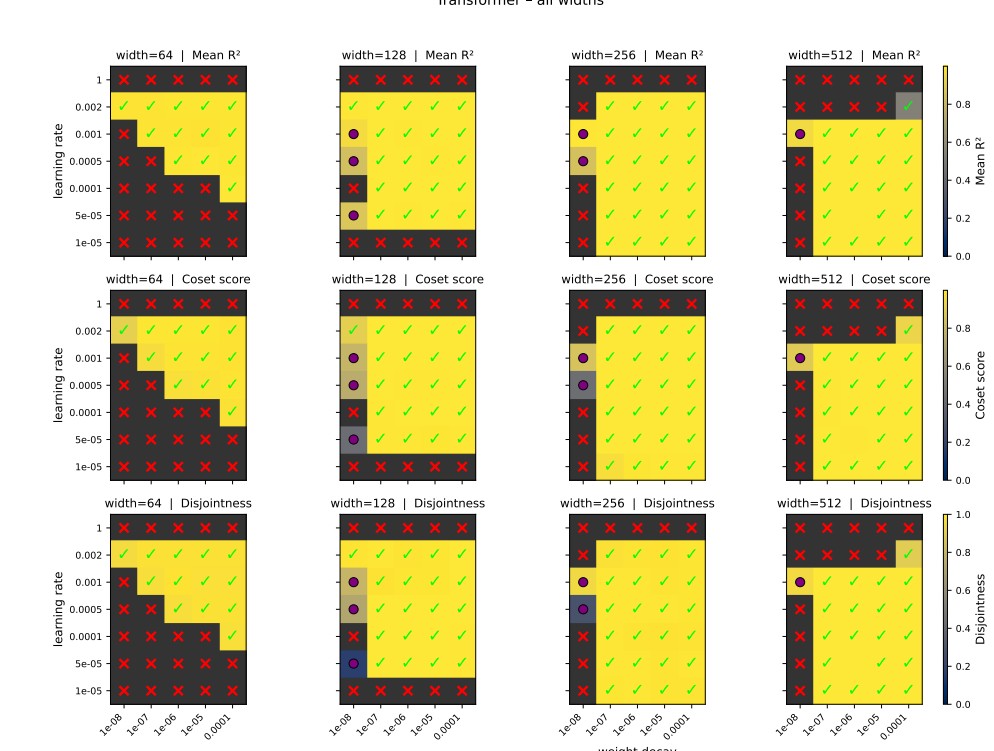

Figure 14: Hyperparameter tuning results over different widths (dmodel) of one-layer transformers. As in Figure 12, each heatmap shows a sweep over learning rate (y-axis) and weight decay (x-axis), and the color scale and markers (red "×", purple "○", and green "✓") follow the same conventions.

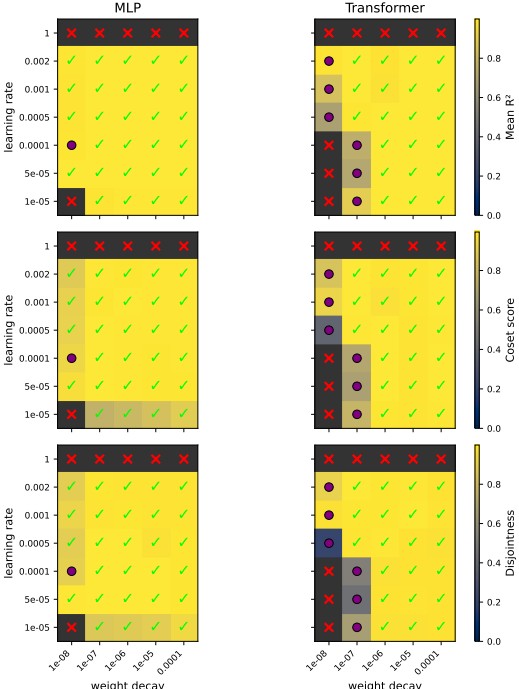

Figure 15: Hyperparameter tuning results for one-layer MLPs and transformers with pure quadratic activation function. As in Figure 12, each heatmap shows a sweep over learning rate (y-axis) and weight decay (x-axis), and the color scale and markers (red "×", purple "○", and green "✓") follow the same conventions.

| Fourier basis | MLPs | | Transformers with lr schedule | | Transformers without lr schedule | |
|---|---|---|---|---|---|---|
| | Layer 1 | Layer 2 | Layer 1 | Layer 2 | Layer 1 | Layer 2 |
| rp | 0.9999 | 0.9988 | 0.9981 | 0.9994 | 0.9849 | 0.9096 |
| srp | 0.9999 | 0.9988 | 0.9986 | 0.9993 | 0.9812 | 0.9084 |
| 2D_1 | 0.9968 | 0.9879 | 0.9971 | 0.9735 | 0.9719 | 0.8642 |
| 2D_2 | 0.9990 | 0.9902 | 0.9970 | 0.9727 | 0.9673 | 0.8611 |
| 2D_3 | 0.9831 | 0.9769 | 0.9999 | 0.9906 | 0.9662 | 0.8638 |
| 2D_4 | 0.9987 | 0.9892 | 0.9952 | 0.9745 | 0.9673 | 0.8696 |
| 2D_5 | 0.9958 | 0.9863 | 0.9972 | 0.9740 | 0.9736 | 0.8707 |
| 2D_6 | 0.9997 | 0.9995 | 0.9946 | 0.9998 | 0.9814 | 0.9203 |
| 2D_7 | 0.9967 | 0.9881 | 0.9975 | 0.9745 | 0.9735 | 0.8677 |
| 2D_8 | 0.9989 | 0.9900 | 0.9941 | 0.9722 | 0.9705 | 0.8699 |

Table 3: Mean of cluster-average $R^2$ per Fourier basis for MLPs and transformers (layer 1 / layer 2 preactivations). In Layer 1 we fit only first-order sinusoids, whereas in Layer 2 we fit both first- and second-order sinusoids. For the MLP and for the transformer with a learning rate schedule, replacing each neuron's preactivation (layerwise) by its fitted sinusoid(s) leaves test accuracy at $100\%$ in $100\%$ of runs.

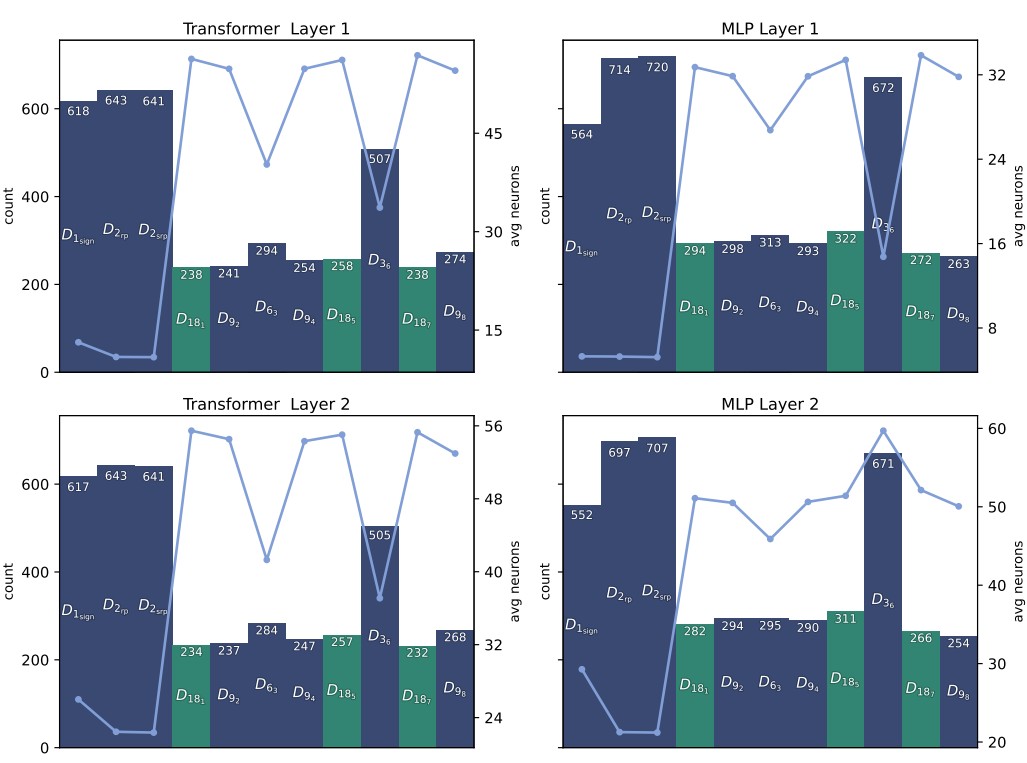

Figure 16: Across-seed average Cayley graph-frequency spectra learned by two-layer networks on $D_{18}$ and the average number of neurons per cluster (1000 random seeds). The two panels overlay results from transformers with a learning rate schedule and MLPs trained under the same protocol. Top: layer 1; bottom: layer 2.

### B.6.2 $D_{18}$

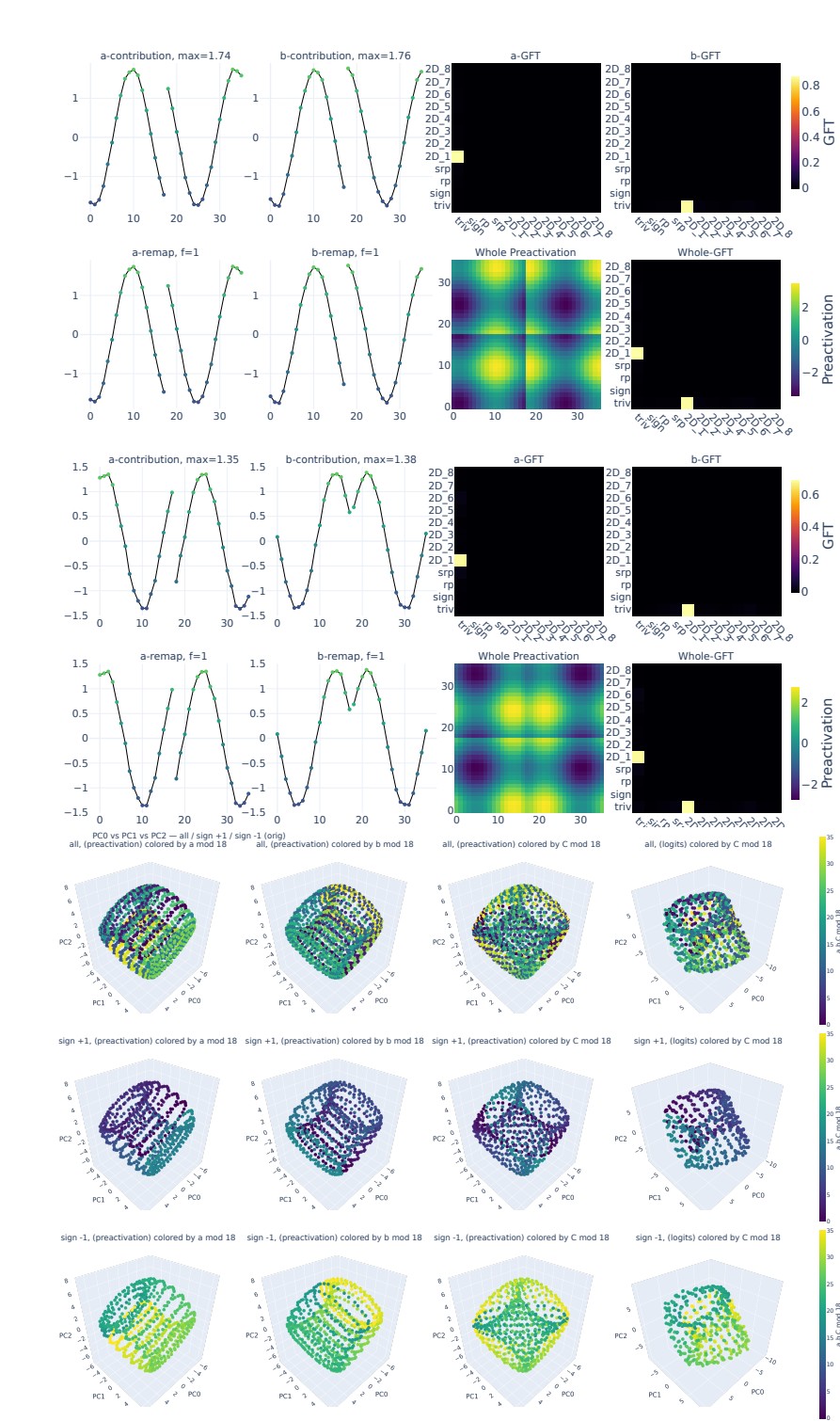

Figure 17: Visualization for $n = 18$, Fourier basis $2D_1$. Top: two neurons with highest preactivation in $2D_1$. Bottom: PCA of preactivations and cluster contributions to logits, colored by $a$, $b$, and $C \bmod 18$ (rotation classes 0–17, reflection classes 18–35).

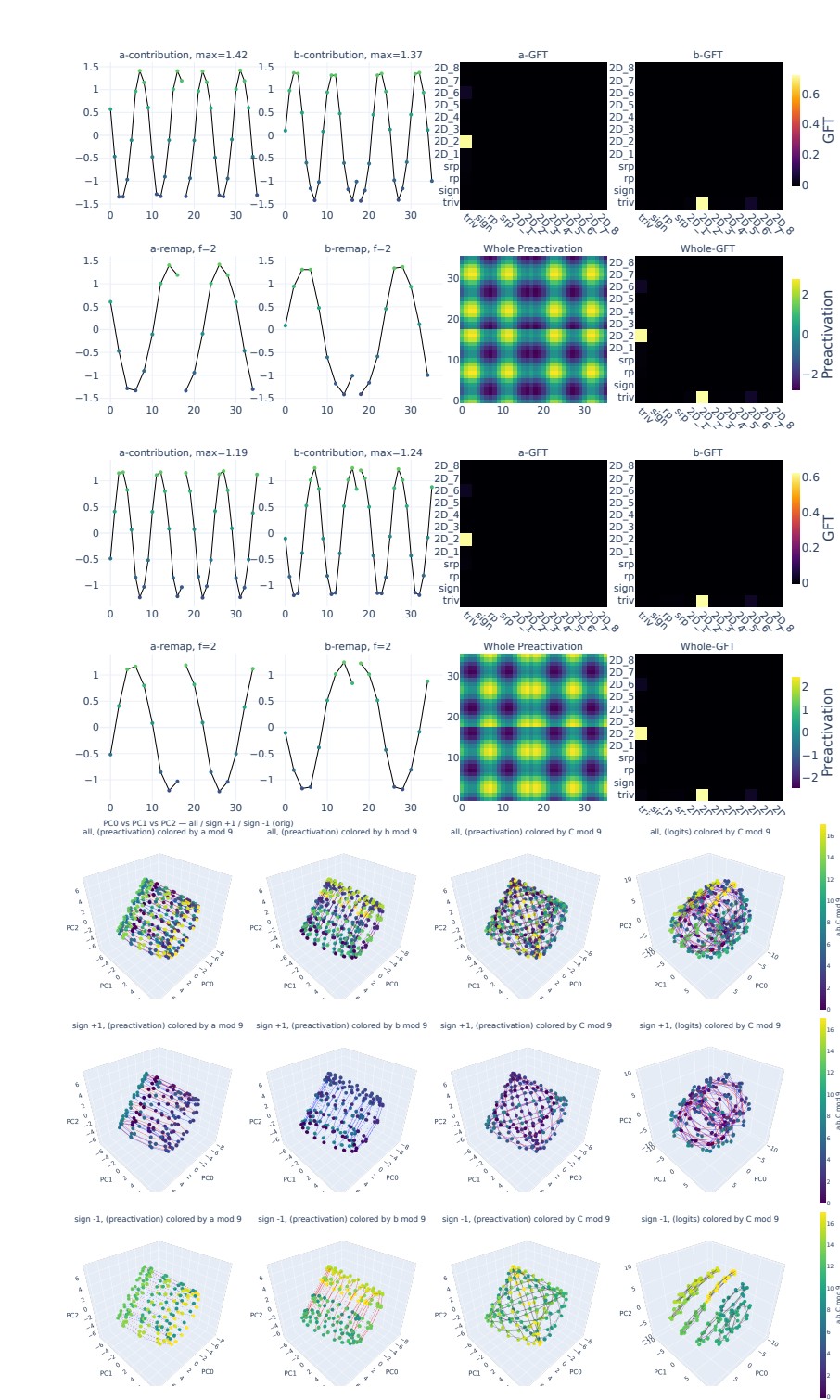

Figure 18: Visualization for $n = 18$, Fourier basis $2D_2$. Top: two neurons with highest preactivation in $2D_2$. Bottom: PCA of preactivations and cluster contributions to logits, colored by $a$, $b$, and $C \bmod 18$ (rotation classes 0–17, reflection classes 18–35).

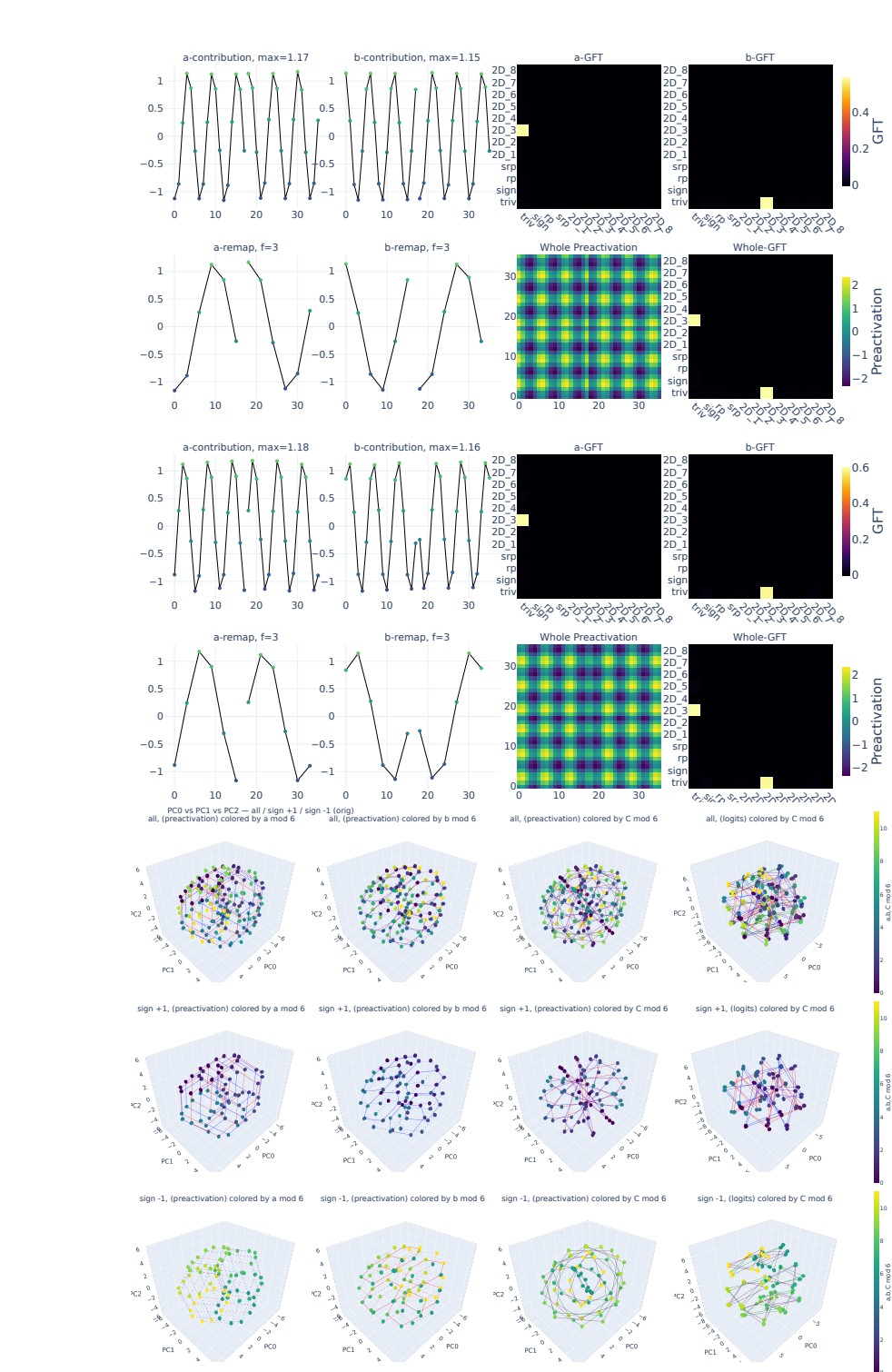

Figure 19: Visualization for $n = 18$, Fourier basis $2D_3$. Top: two neurons with highest preactivation in $2D_3$. Bottom: PCA of preactivations and cluster contributions to logits, colored by $a$, $b$, and $C \bmod 18$ (rotation classes 0–17, reflection classes 18–35).

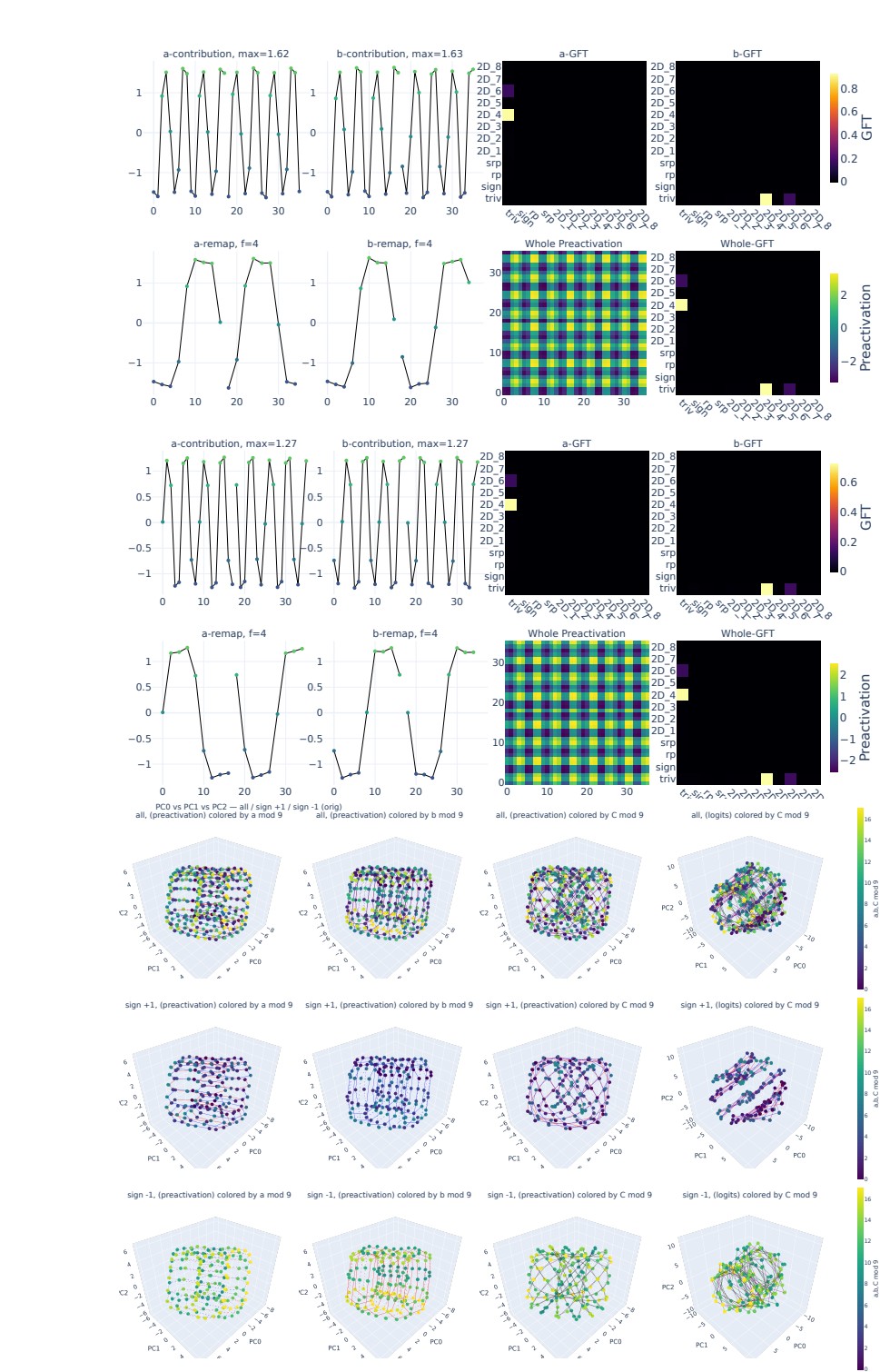

Figure 20: Visualization for $n = 18$, Fourier basis $2D_4$. Top: two neurons with highest preactivation in $2D_4$. Bottom: PCA of preactivations and cluster contributions to logits, colored by $a$, $b$, and $C \bmod 18$ (rotation classes 0–17, reflection classes 18–35).

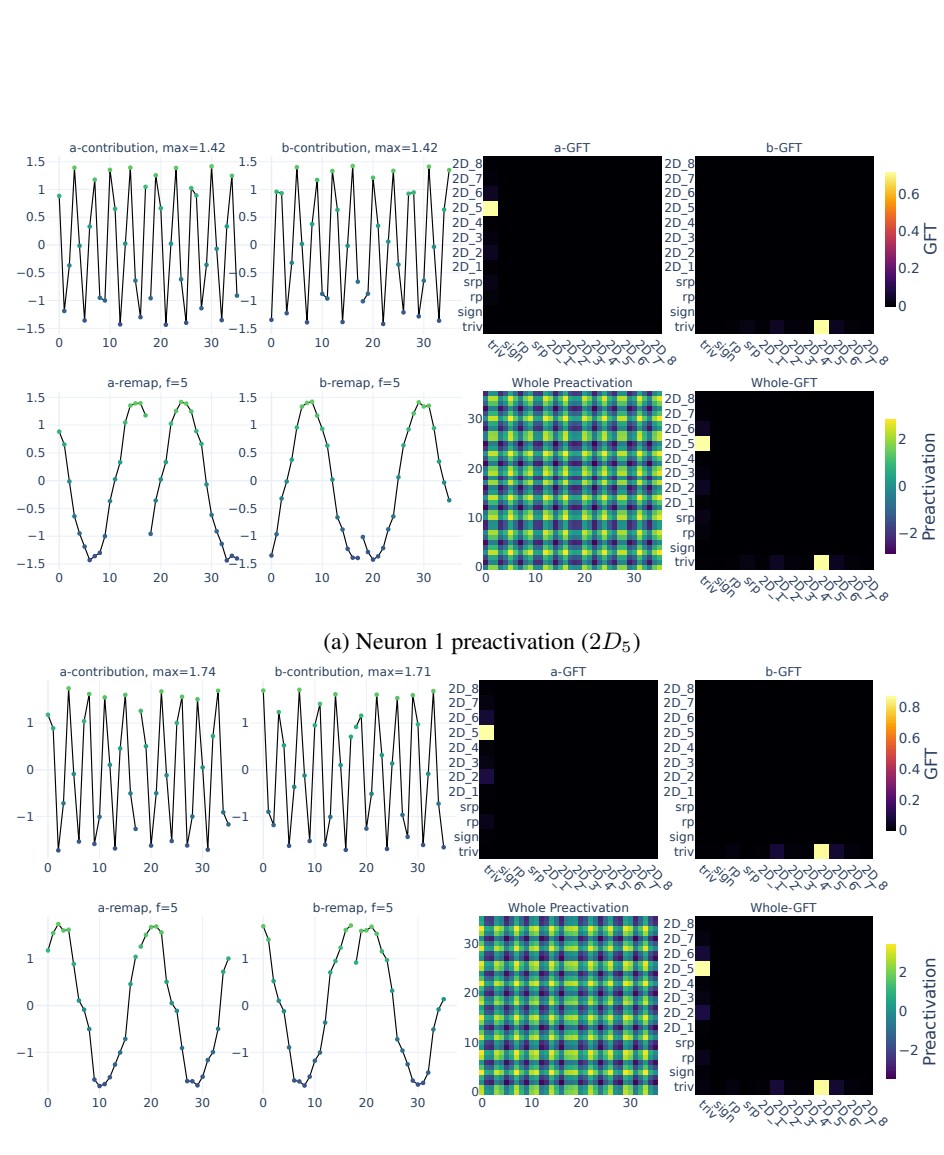

(a) Neuron 1 preactivation ($2D_5$)

(b) Neuron 2 preactivation ($2D_5$)

Figure 21: Visualization for $n = 18$, Fourier basis $2D_5$ (part 1/2).

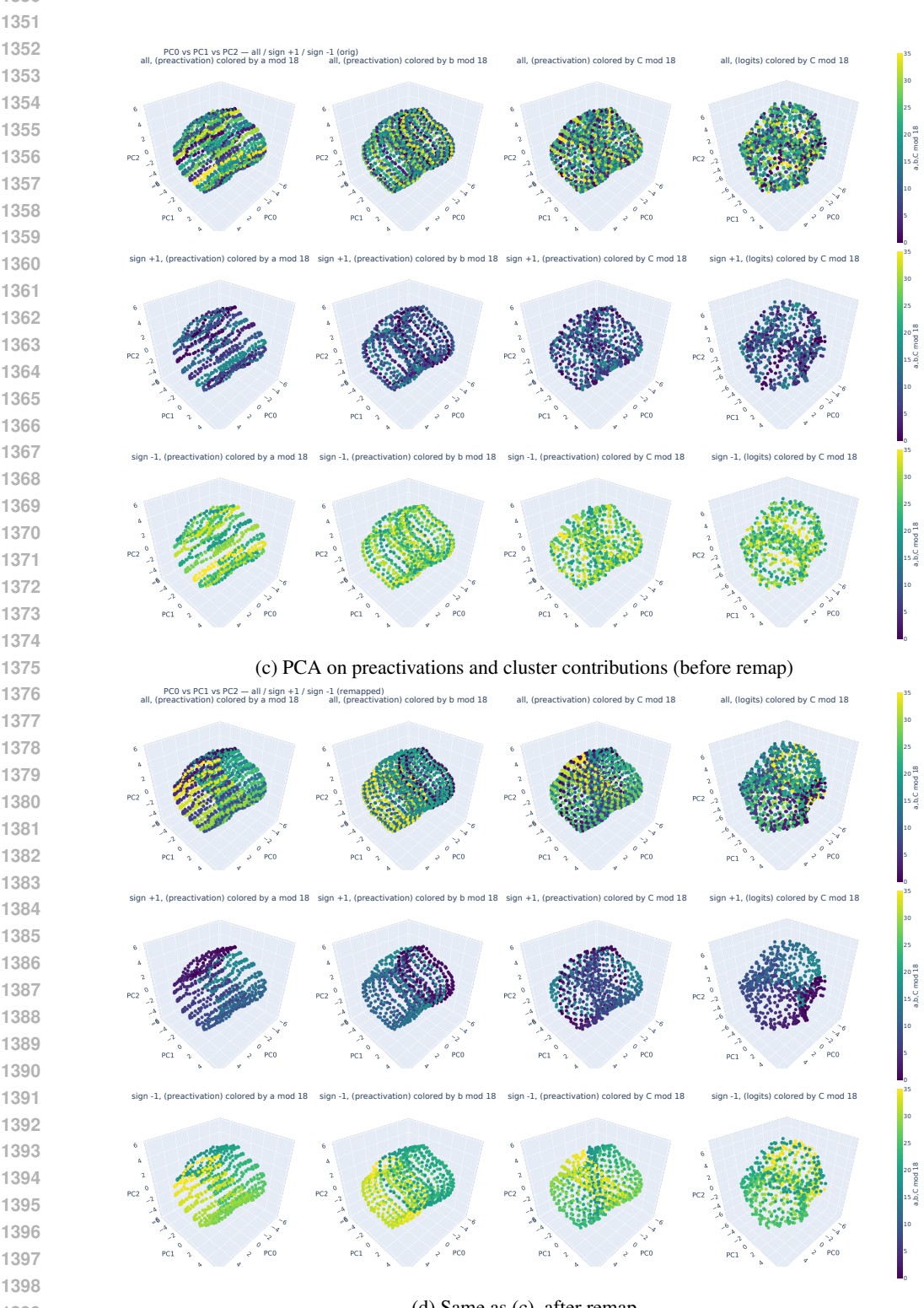

(c) PCA on preactivations and cluster contributions (before remap)

(d) Same as (c), after remap

Figure 21: Visualization for $n = 18$, Fourier basis $2D_5$ (part 2/2). Colored by $a$, $b$, and $C \bmod 18$, before (c) and after (d) remapping.

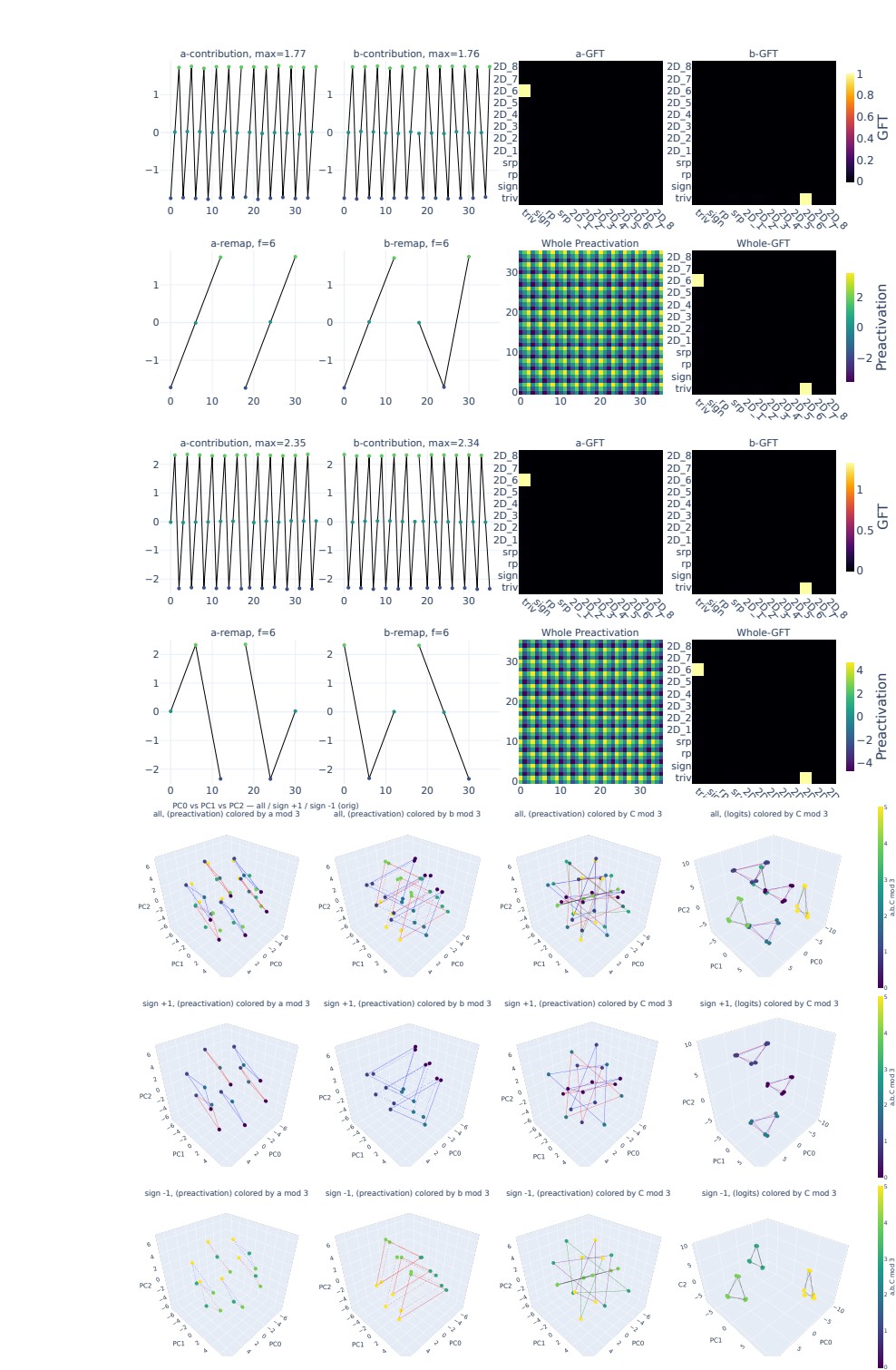

Figure 22: Visualization for $n = 18$, Fourier basis $2D_6$. Top: two neurons with highest preactivation in $2D_6$. Bottom: PCA of preactivations and cluster contributions to logits, colored by $a$, $b$, and $C \bmod 18$ (rotation classes 0–17, reflection classes 18–35).

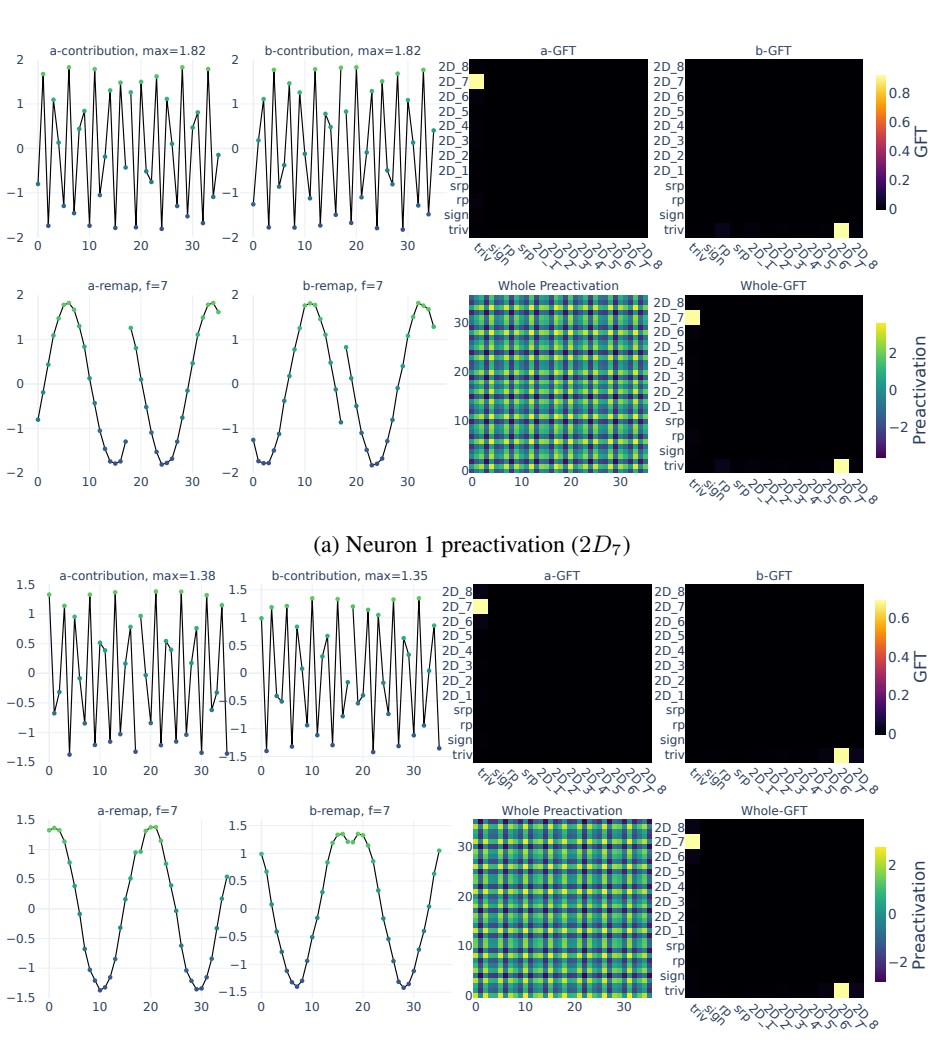

(a) Neuron 1 preactivation ($2D_7$)

(b) Neuron 2 preactivation ($2D_7$)

Figure 23: Visualization for $n = 18$, Fourier basis $2D_7$ (part 1/2). Two neurons with highest preactivation in $2D_7$.

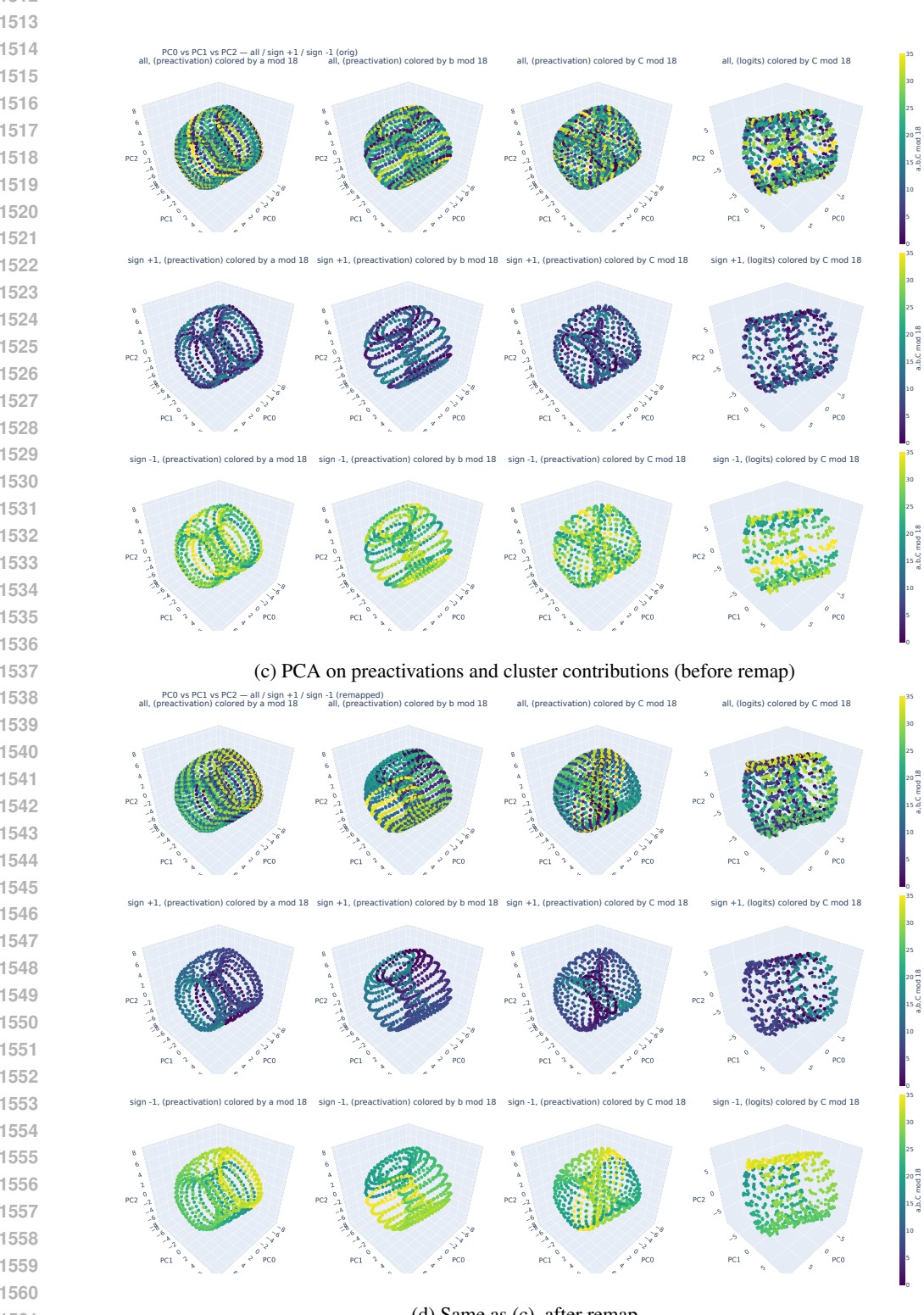

(c) PCA on preactivations and cluster contributions (before remap)

(d) Same as (c), after remap

Figure 23: Visualization for $n = 18$, Fourier basis $2D_7$ (part 2/2). PCA of preactivations and cluster contributions to logits, colored by $a$, $b$, and $C \bmod 18$, shown before and after remapping.

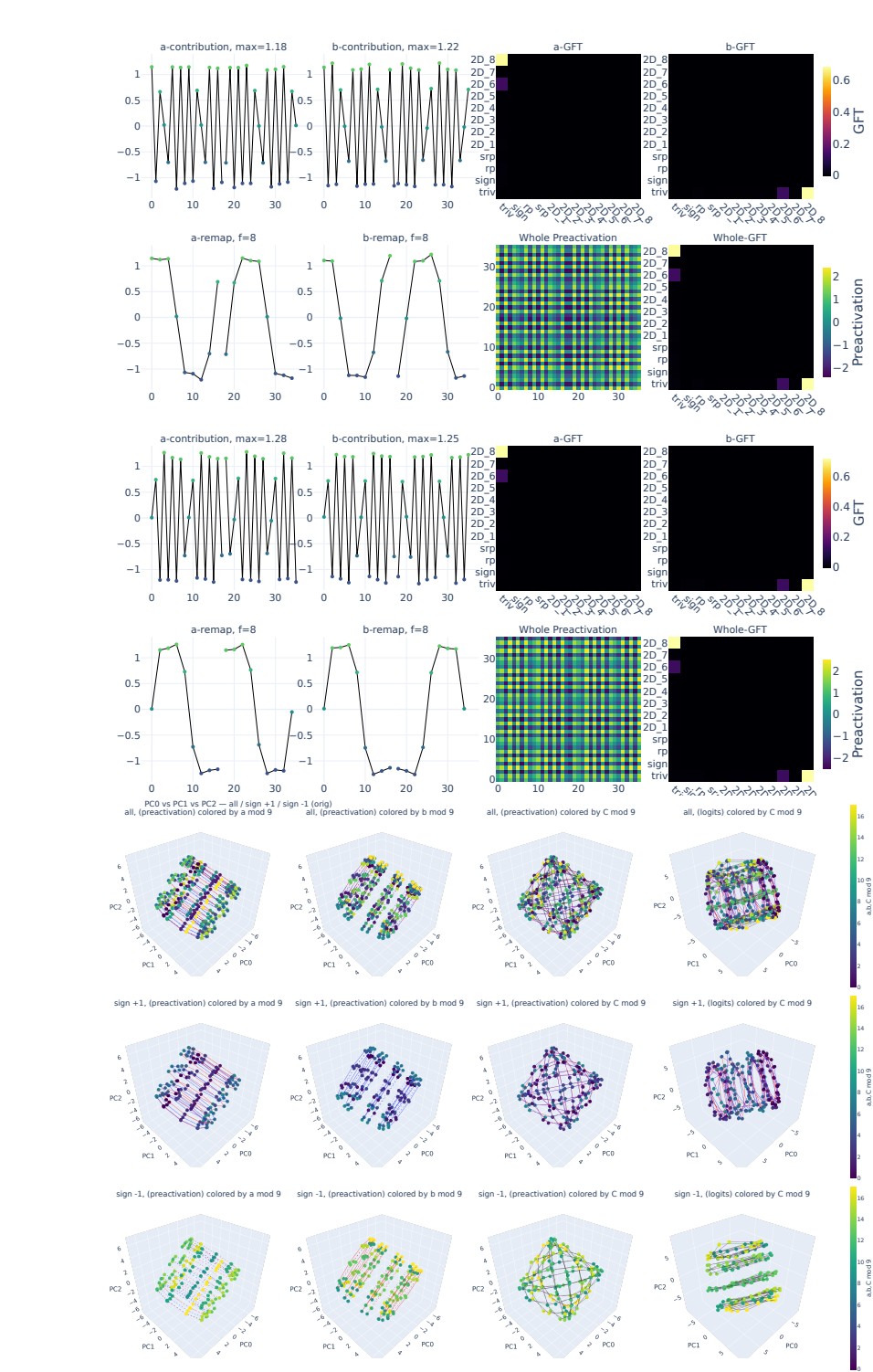

Figure 24: Visualization for $n = 18$, Fourier basis $2D_8$. Top: two neurons with highest preactivation in $2D_8$. Bottom: PCA of preactivations and cluster contributions to logits, colored by $a$, $b$, and $C \bmod 18$ (rotation classes 0–17, reflection classes 18–35).

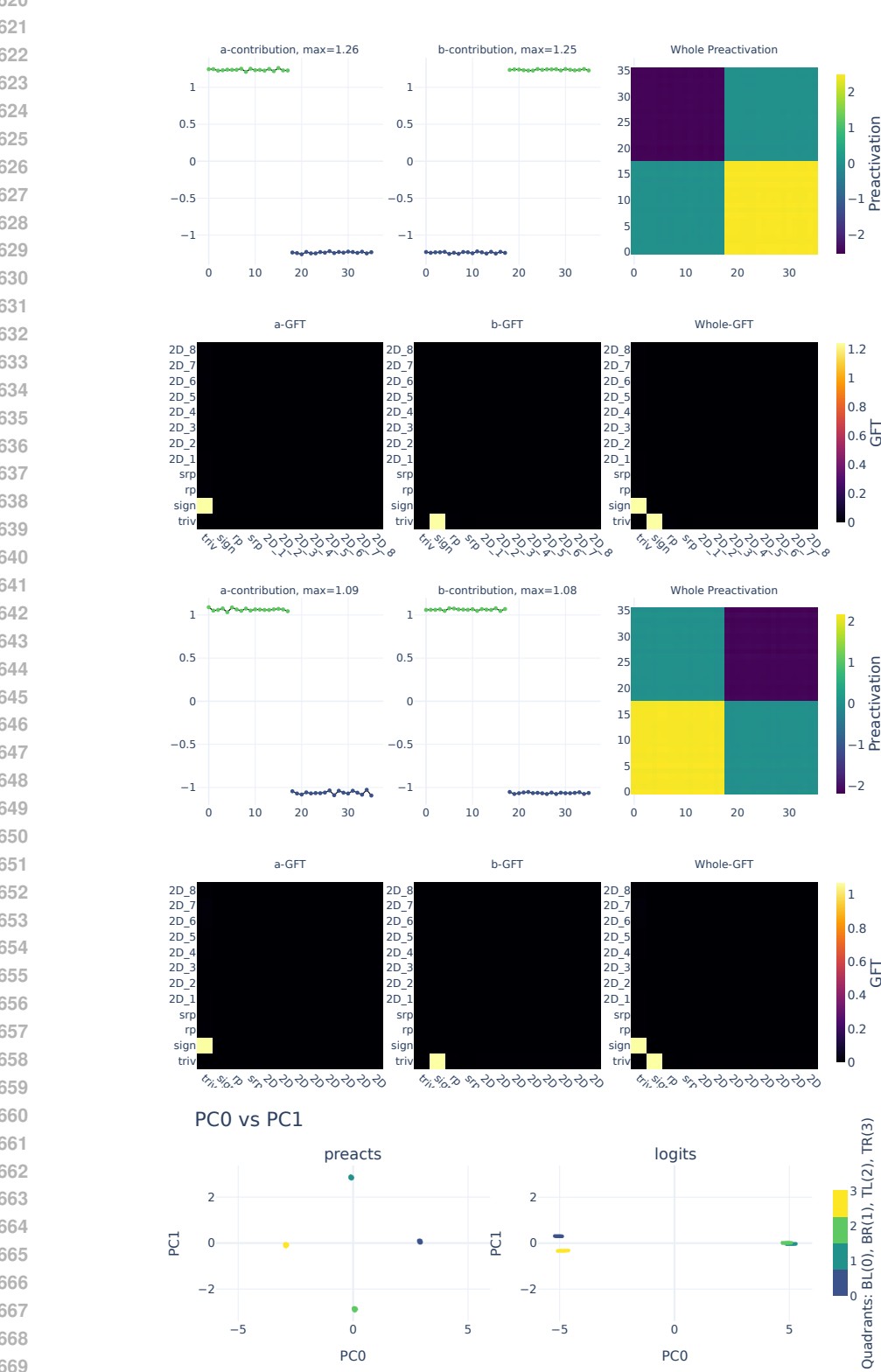

Figure 25: Visualization for $n = 18$, Fourier basis `sign`. Top: two neurons with highest preactivation in `sign`. Bottom: PCA of preactivations and cluster contributions to logits, colored by quadrant (BL, BR, TL, TR) of $(a, b)$ in the $2p \times 2p$ grid.

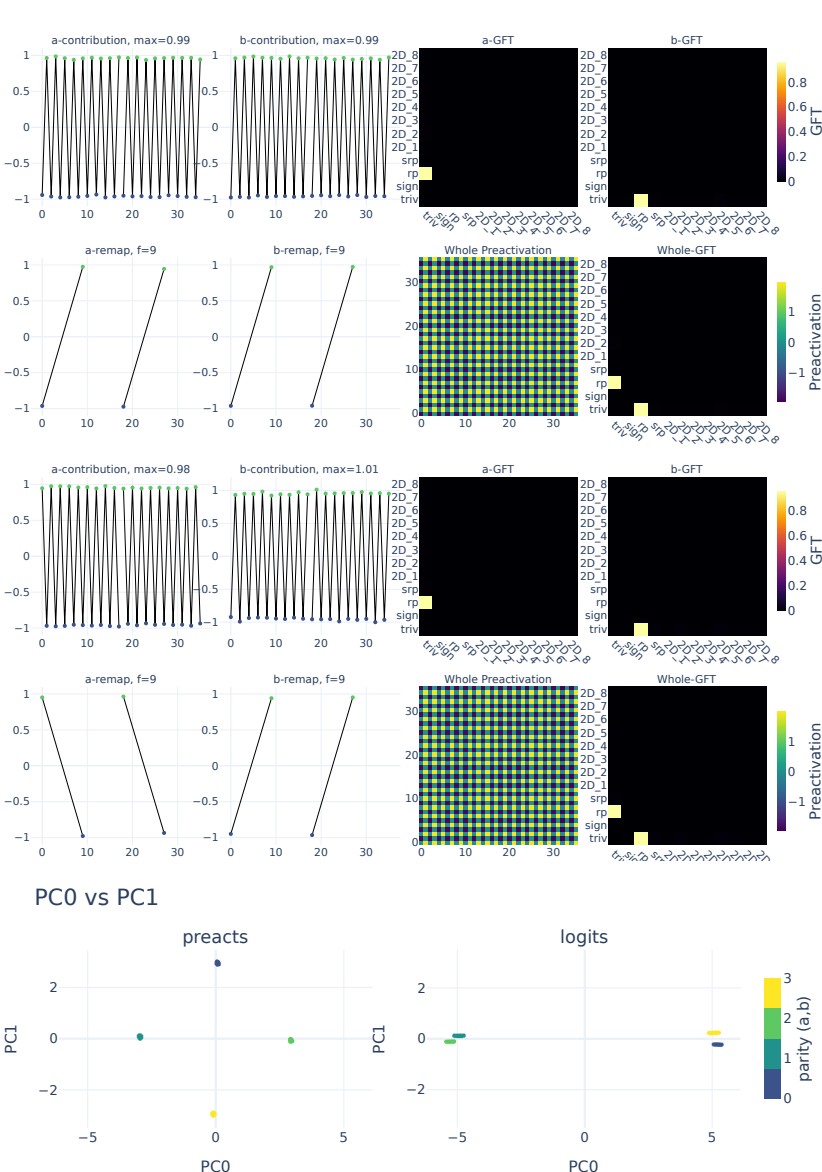

Figure 26: Visualization for $n = 18$, Fourier basis rp. Top: two neurons with highest preactivation in rp. Bottom: PCA of preactivations and cluster contributions to logits, colored by **parity** of $(a, b)$: $\text{code} = 2 \cdot (a \bmod 2) + (b \bmod 2)$.

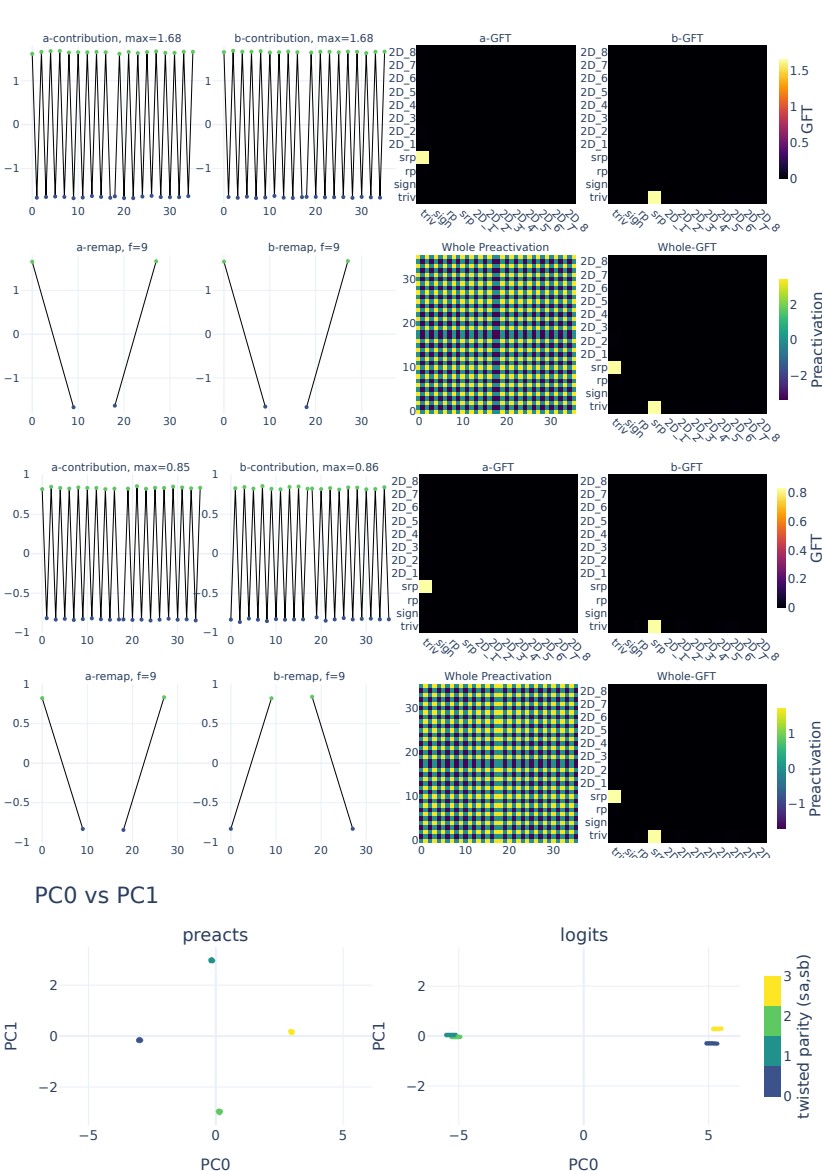

Figure 27: Visualization for $n = 18$, Fourier basis `srp`. Top: two neurons with highest preactivation in `srp`. Bottom: PCA of preactivations and cluster contributions to logits, colored by **twisted parity**: $s_a = (a \bmod 2) \oplus \mathbf{1}[a \geq n]$, $s_b = (b \bmod 2) \oplus \mathbf{1}[b \geq n]$, where $x \oplus y = (x + y) \bmod 2$, and $\text{code} = 2s_a + s_b$.

### B.6.3   $D_{19}$

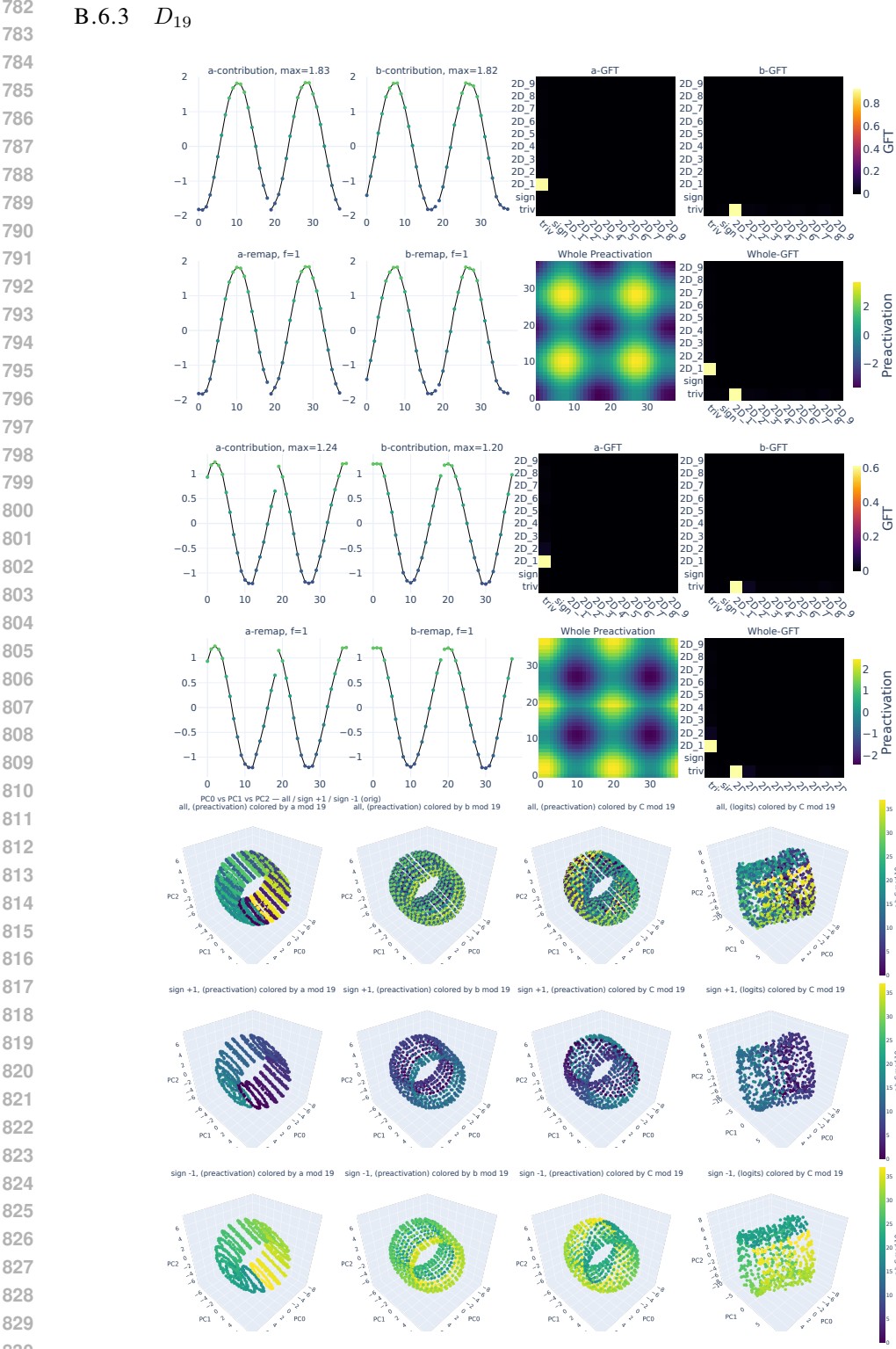

Figure 28: Visualization for $n = 19$, Fourier basis $2D_1$. Top: two neurons with highest preactivation in $2D_1$. Bottom: PCA of preactivations and cluster contributions to logits, colored by $a$, $b$, and $C \bmod 19$ (rotation classes 0–18, reflection classes 19–37).

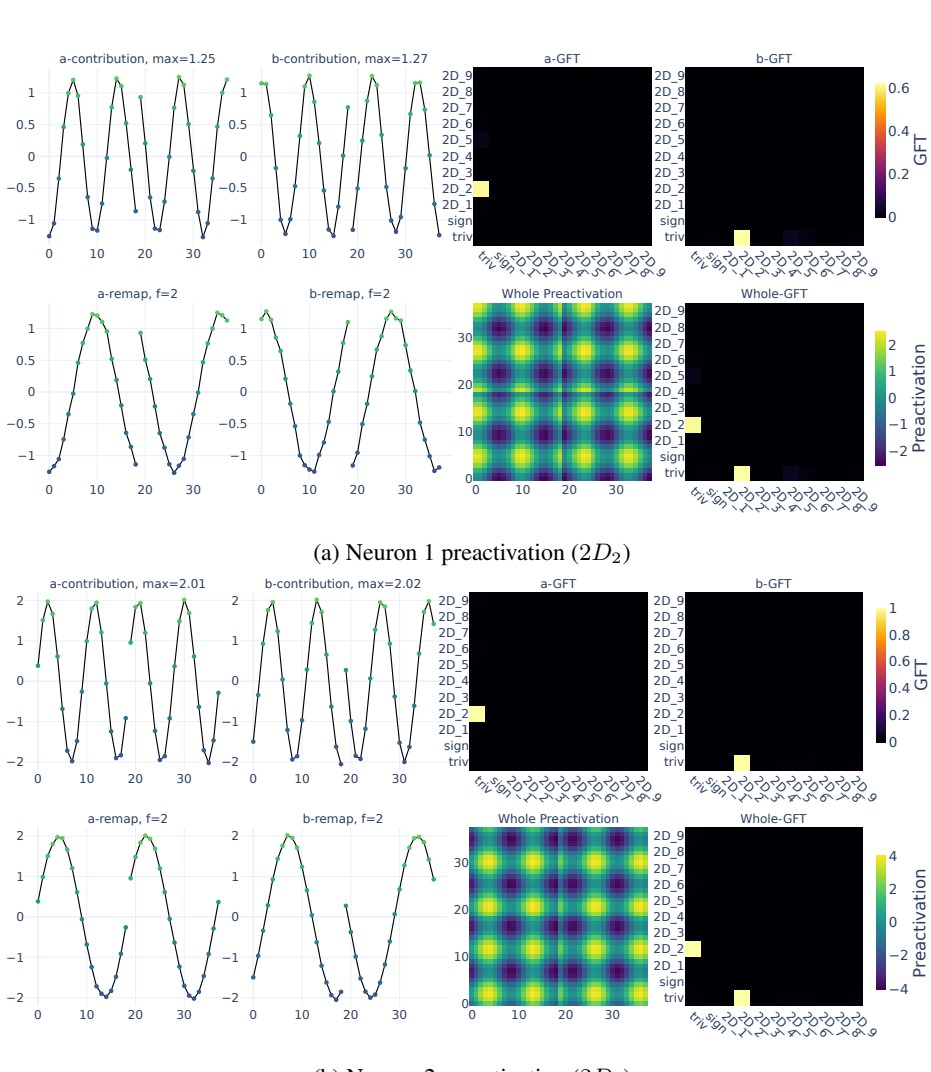

(a) Neuron 1 preactivation ($2D_2$)

(b) Neuron 2 preactivation ($2D_2$)

Figure 29: Visualization for $n = 19$, Fourier basis $2D_2$ (part 1/2). Two neurons with highest preactivation in $2D_2$.

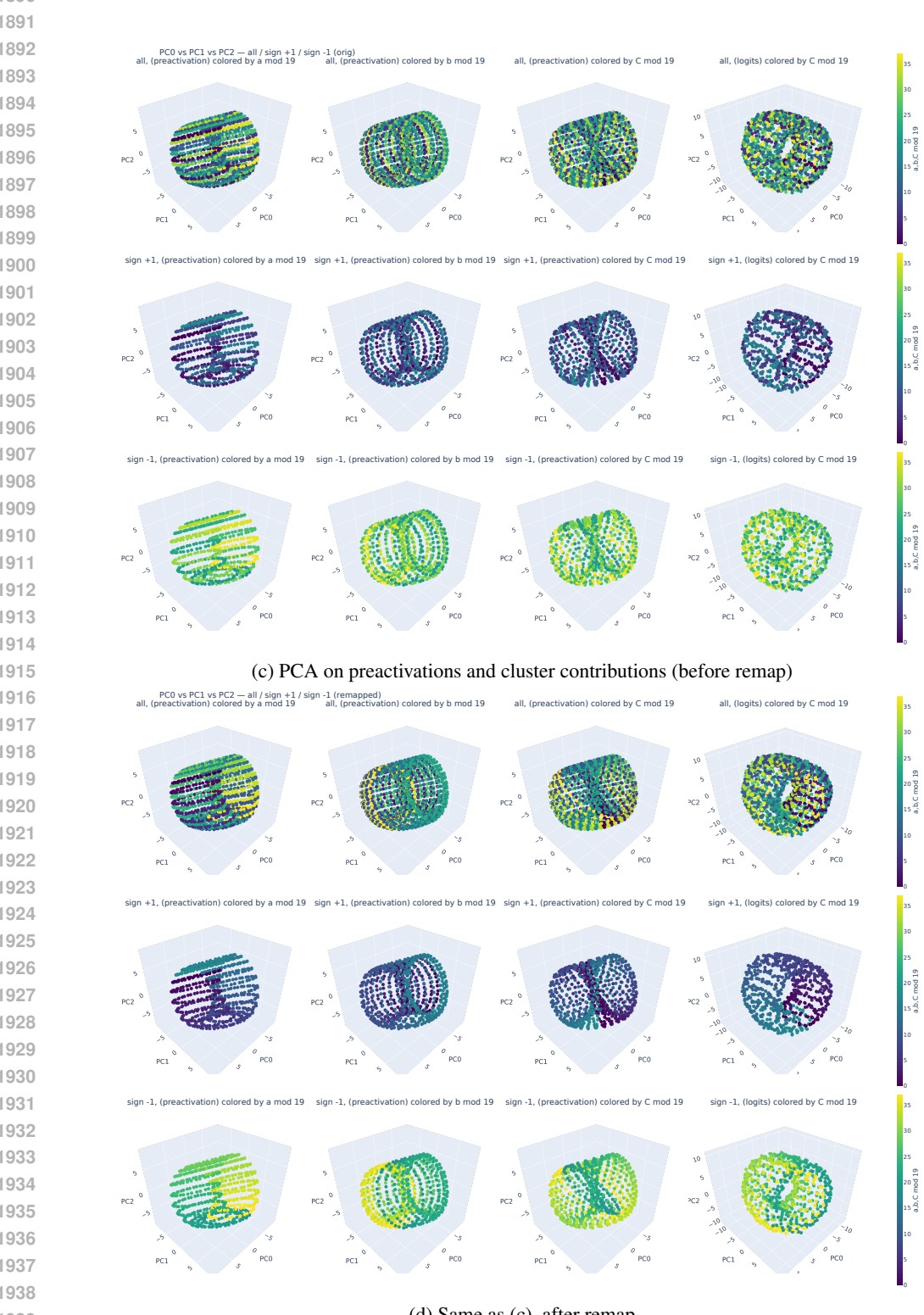

(c) PCA on preactivations and cluster contributions (before remap)

(d) Same as (c), after remap

Figure 29: Visualization for $n = 19$, Fourier basis $2D_2$ (part 2/2). PCA of preactivations and cluster contributions to logits, colored by $a$, $b$, and $C \bmod 19$, shown before and after remapping.

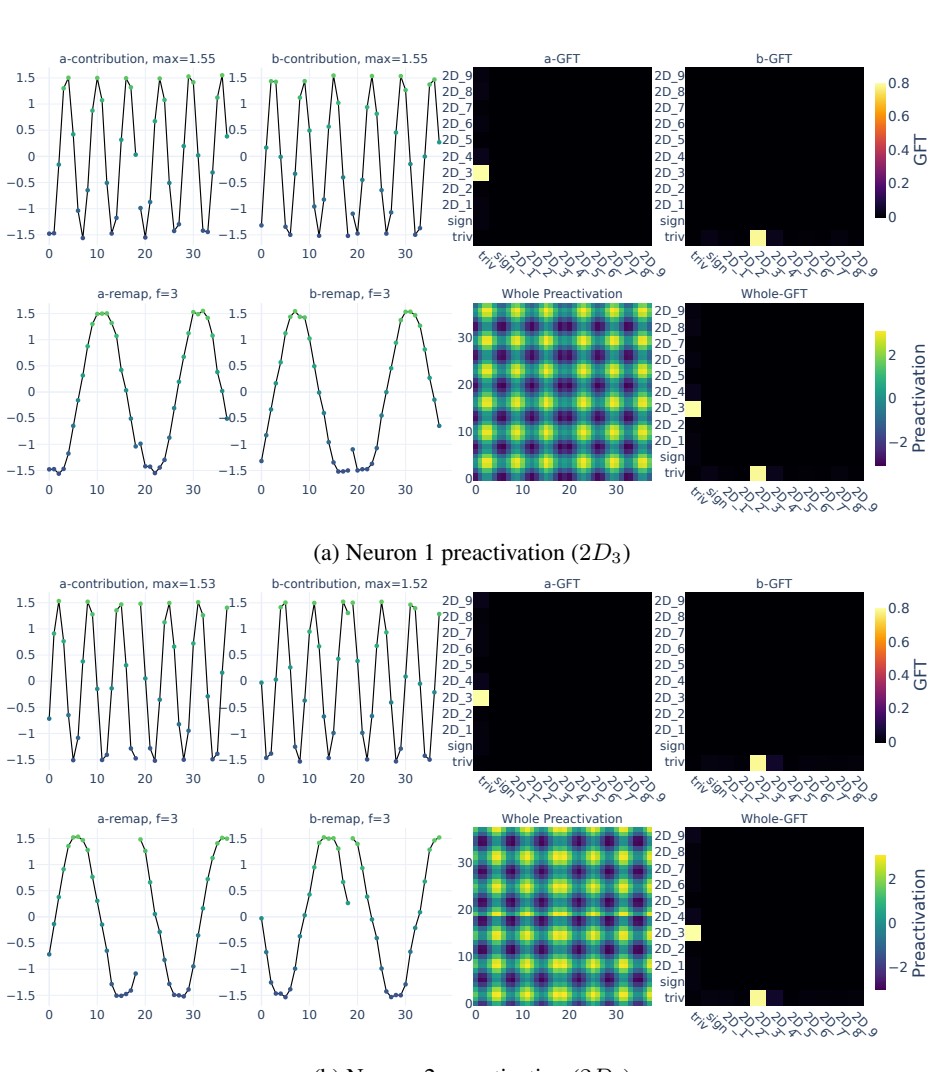

(a) Neuron 1 preactivation ($2D_3$)

(b) Neuron 2 preactivation ($2D_3$)

Figure 30: Visualization for $n = 19$, Fourier basis $2D_3$ (part 1/2). Two neurons with highest preactivation in $2D_3$.

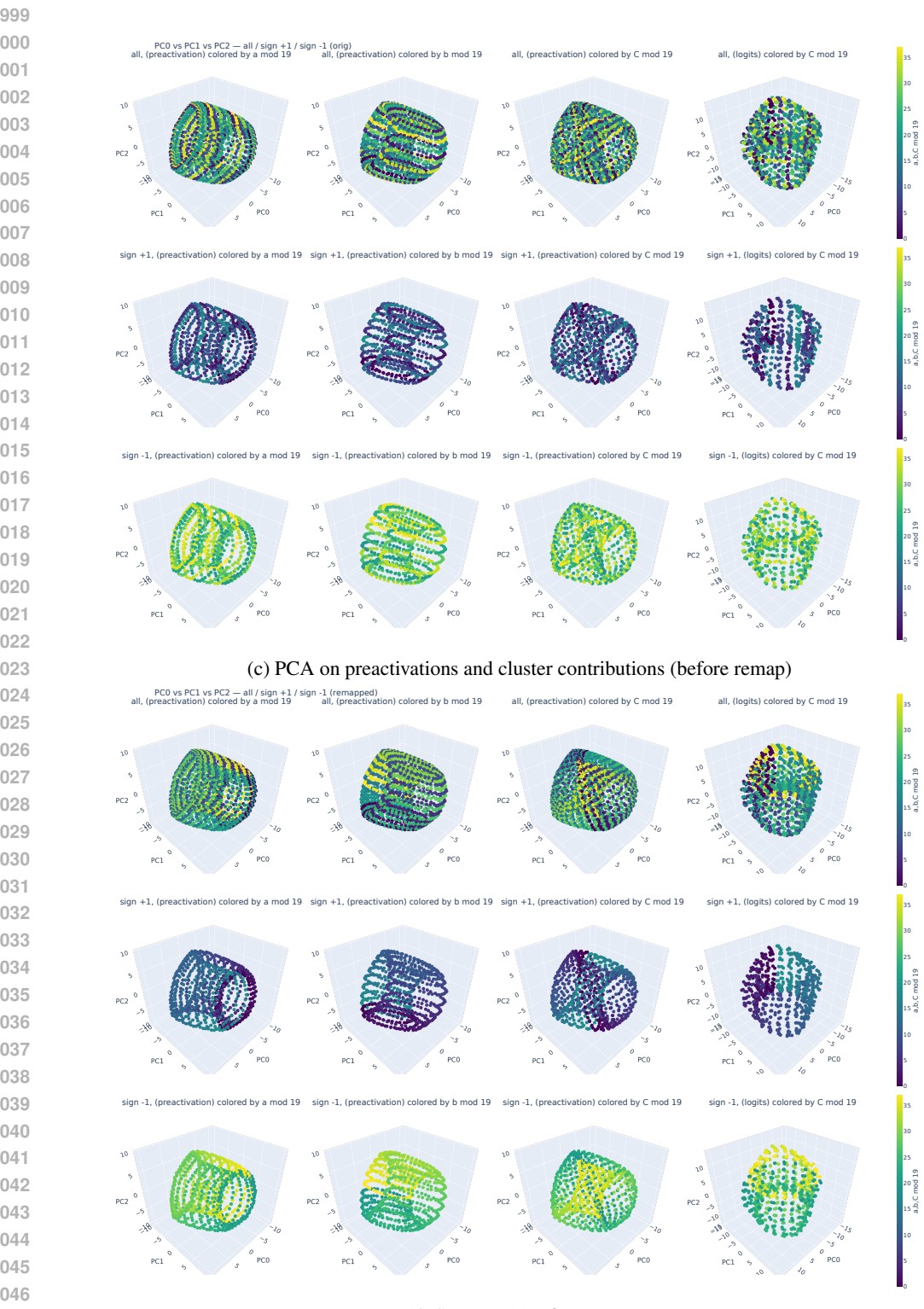

(c) PCA on preactivations and cluster contributions (before remap)

(d) Same as (c), after remap

Figure 30: Visualization for $n = 19$, Fourier basis $2D_3$ (part 2/2). PCA of preactivations and cluster contributions to logits, colored by $a$, $b$, and $C \bmod 19$, shown before and after remapping.

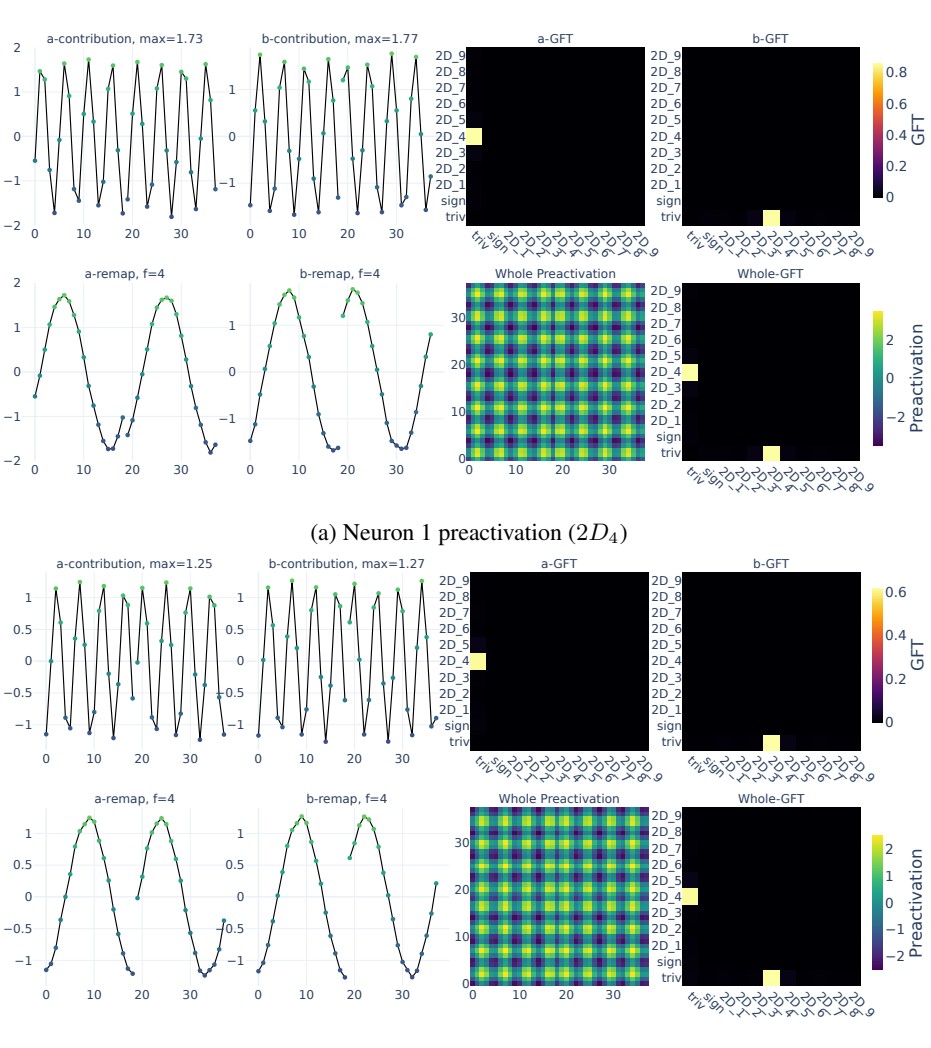

(a) Neuron 1 preactivation ($2D_4$)

(b) Neuron 2 preactivation ($2D_4$)

Figure 31: Visualization for $n = 19$, Fourier basis $2D_4$ (part 1/2). Two neurons with highest preactivation in $2D_4$.

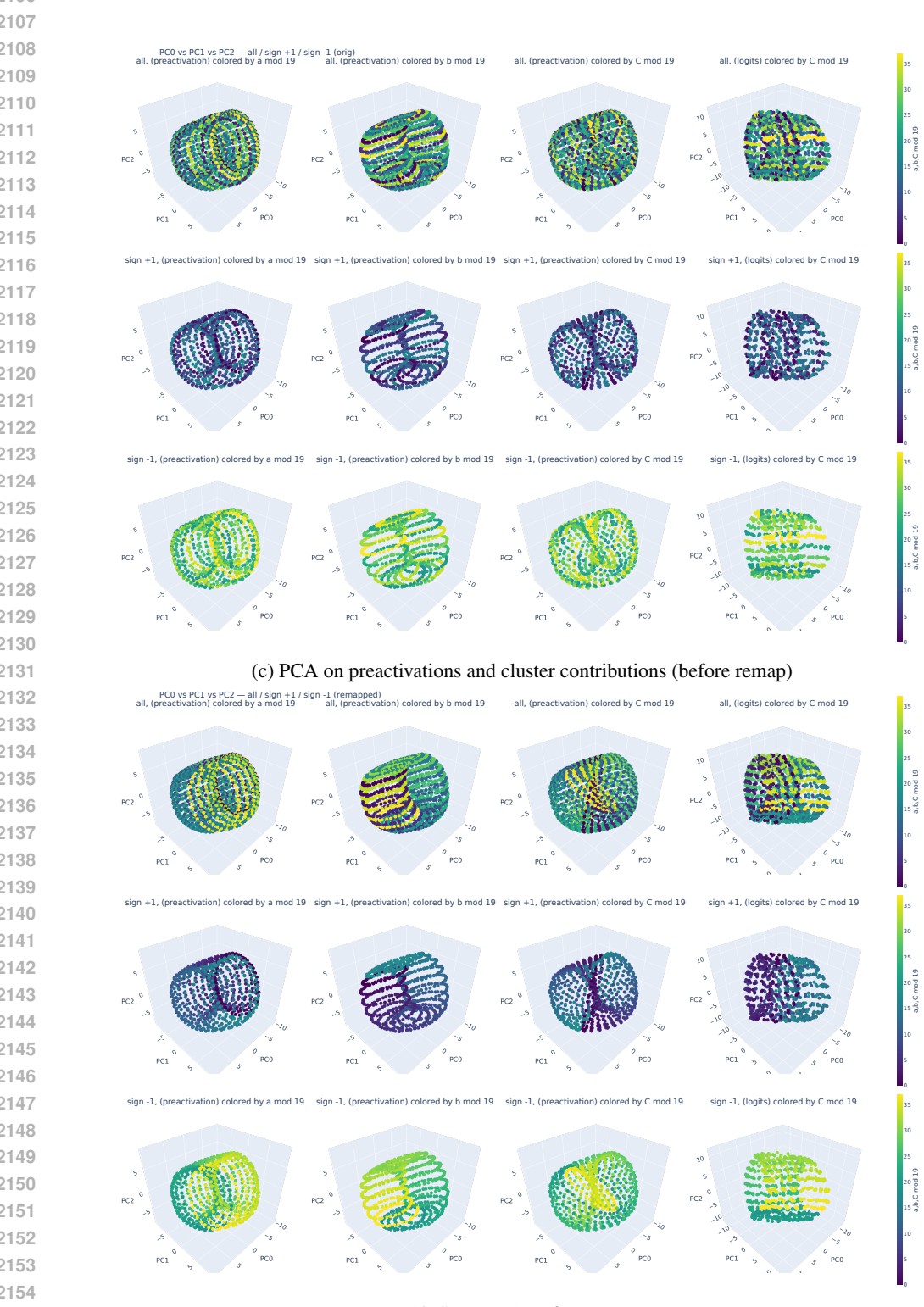

(c) PCA on preactivations and cluster contributions (before remap)

(d) Same as (c), after remap

Figure 31: Visualization for $n = 19$, Fourier basis $2D_4$ (part 2/2). PCA of preactivations and cluster contributions to logits, colored by $a$, $b$, and $C \mod 19$, shown before and after remapping.

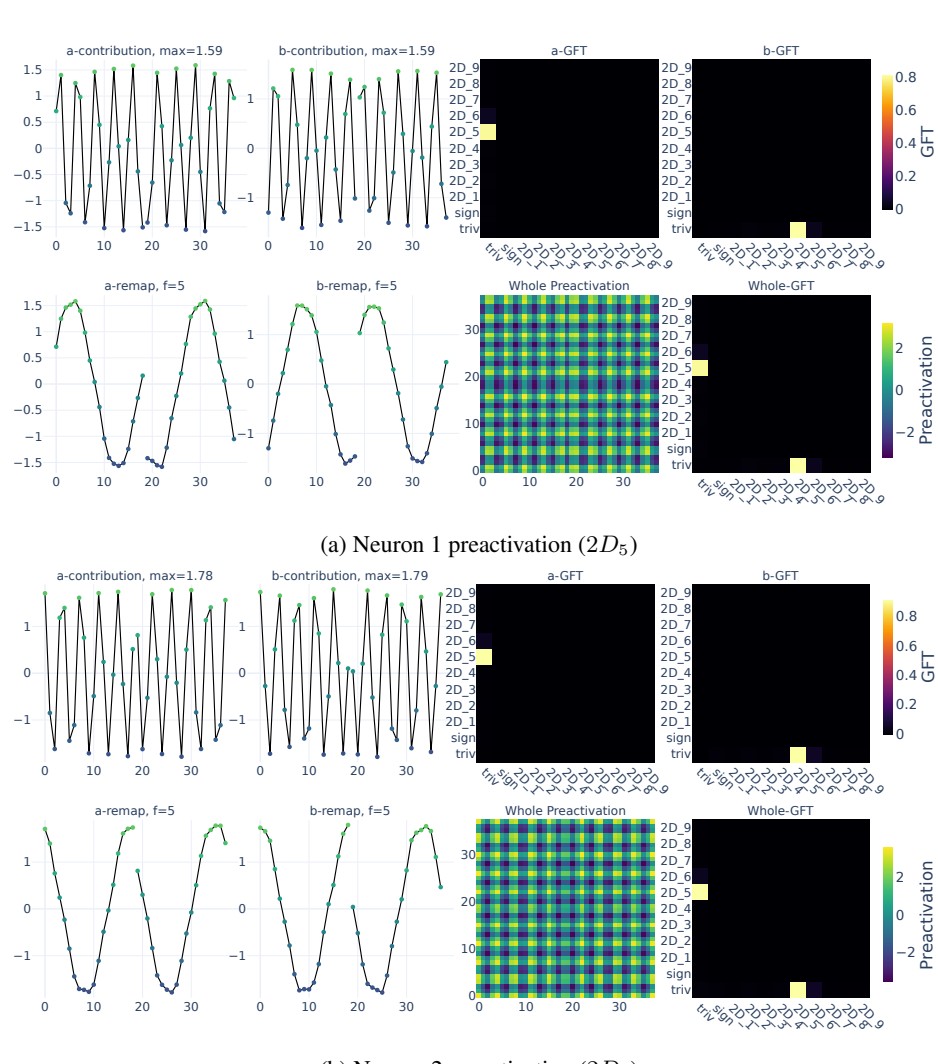

(a) Neuron 1 preactivation ($2D_5$)

(b) Neuron 2 preactivation ($2D_5$)

Figure 32: Visualization for $n = 19$, Fourier basis $2D_5$ (part 1/2). Two neurons with highest preactivation in $2D_5$.

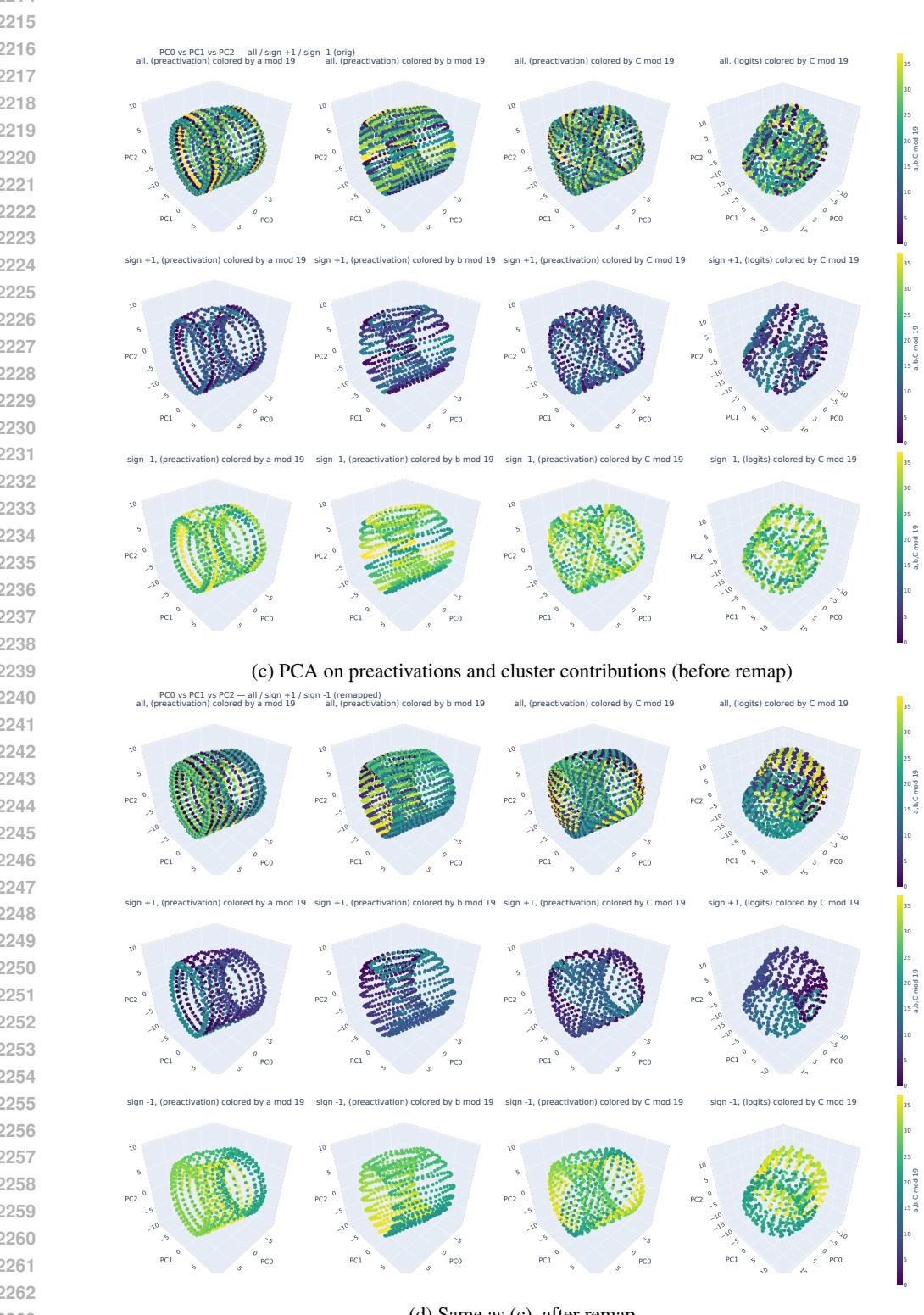

(c) PCA on preactivations and cluster contributions (before remap)

(d) Same as (c), after remap

Figure 32: Visualization for $n = 19$, Fourier basis $2D_5$ (part 2/2). PCA of preactivations and cluster contributions to logits, colored by $a$, $b$, and $C \bmod 19$, shown before and after remapping.

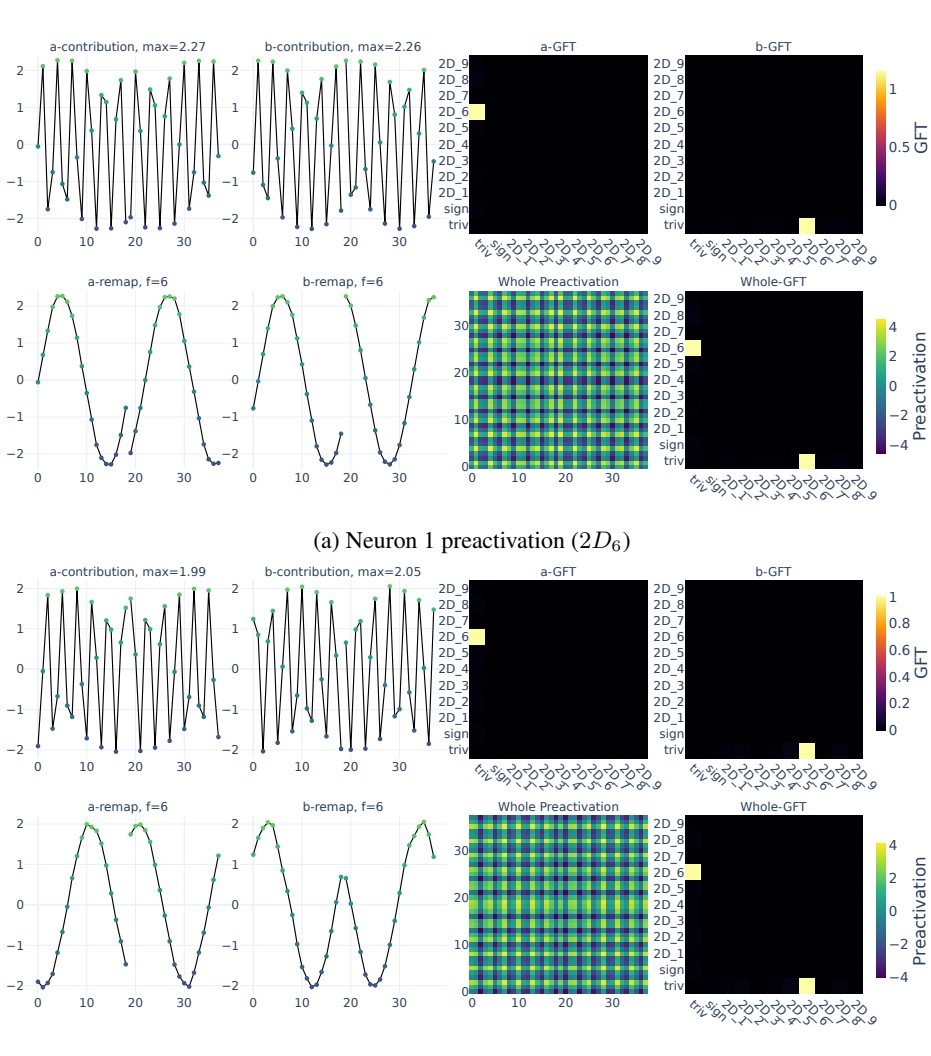

(a) Neuron 1 preactivation ($2D_6$)

(b) Neuron 2 preactivation ($2D_6$)

Figure 33: Visualization for $n = 19$, Fourier basis $2D_6$ (part 1/2). Two neurons with highest preactivation in $2D_6$.

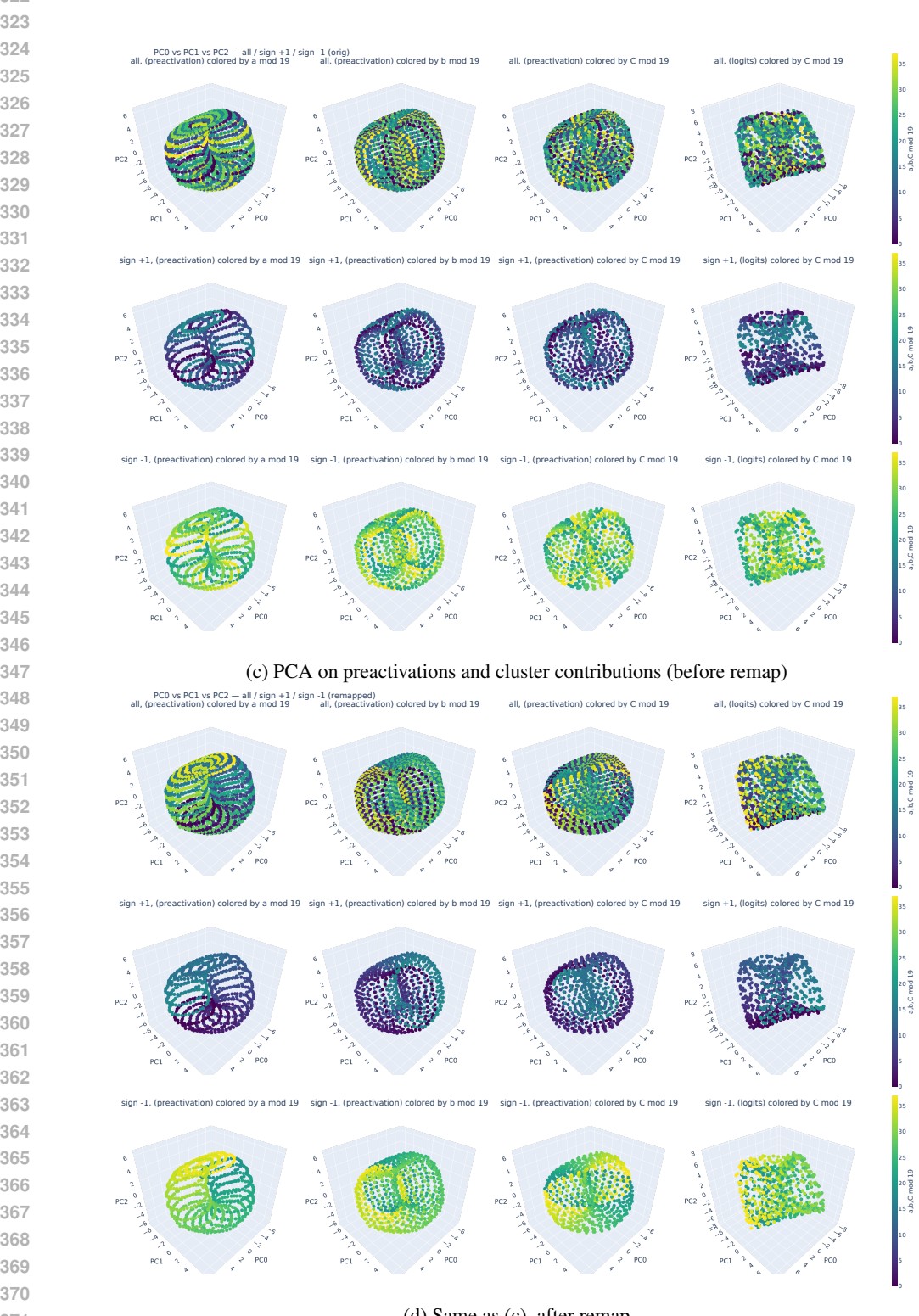

(c) PCA on preactivations and cluster contributions (before remap)

(d) Same as (c), after remap

Figure 33: Visualization for $n = 19$, Fourier basis $2D_6$ (part 2/2). PCA of preactivations and cluster contributions to logits, colored by $a$, $b$, and $C \bmod 19$, shown before and after remapping.

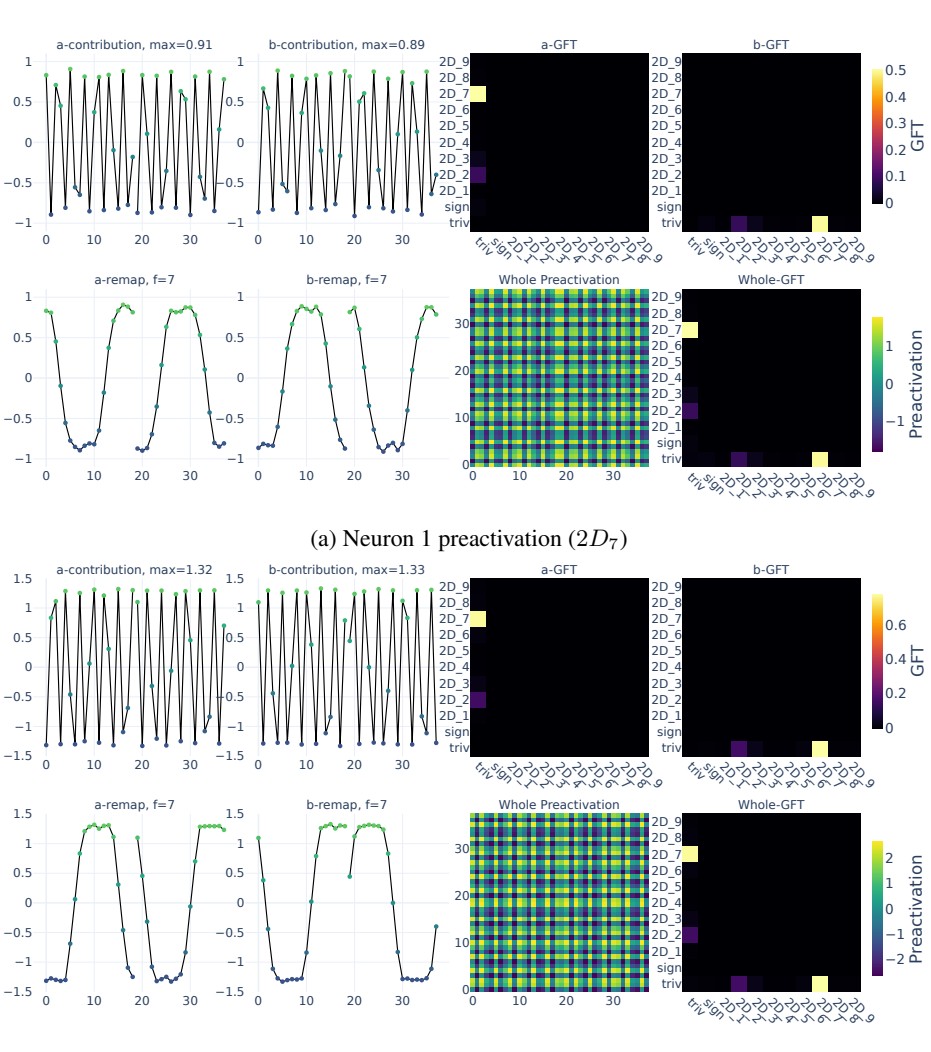

(a) Neuron 1 preactivation ($2D_7$)

(b) Neuron 2 preactivation ($2D_7$)

Figure 34: Visualization for $n = 19$, Fourier basis $2D_7$ (part 1/2). Two neurons with highest preactivation in $2D_7$.

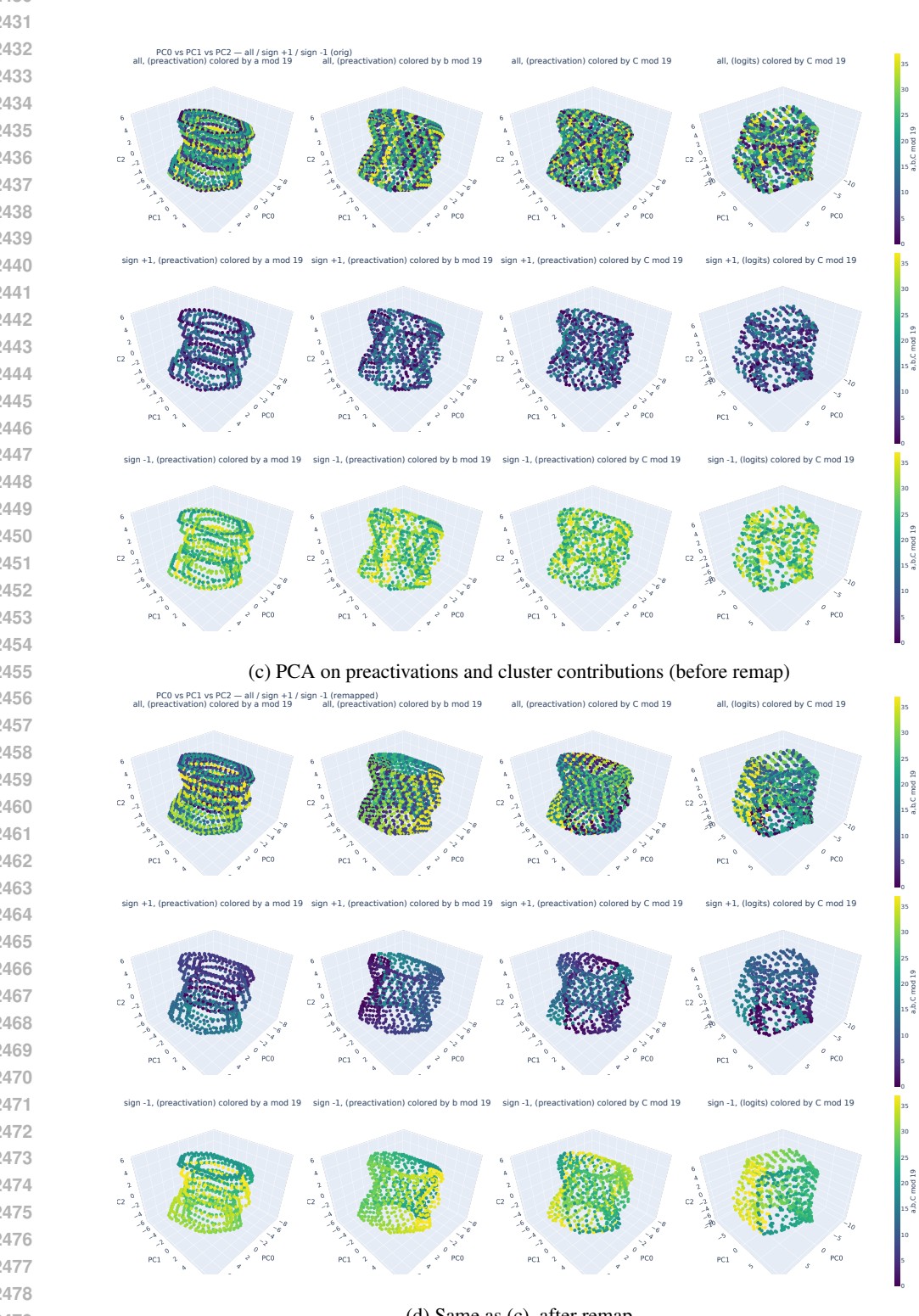

(c) PCA on preactivations and cluster contributions (before remap)

(d) Same as (c), after remap

Figure 34: Visualization for $n = 19$, Fourier basis $2D_7$ (part 2/2). PCA of preactivations and cluster contributions to logits, colored by $a$, $b$, and $C \bmod 19$, shown before and after remapping.

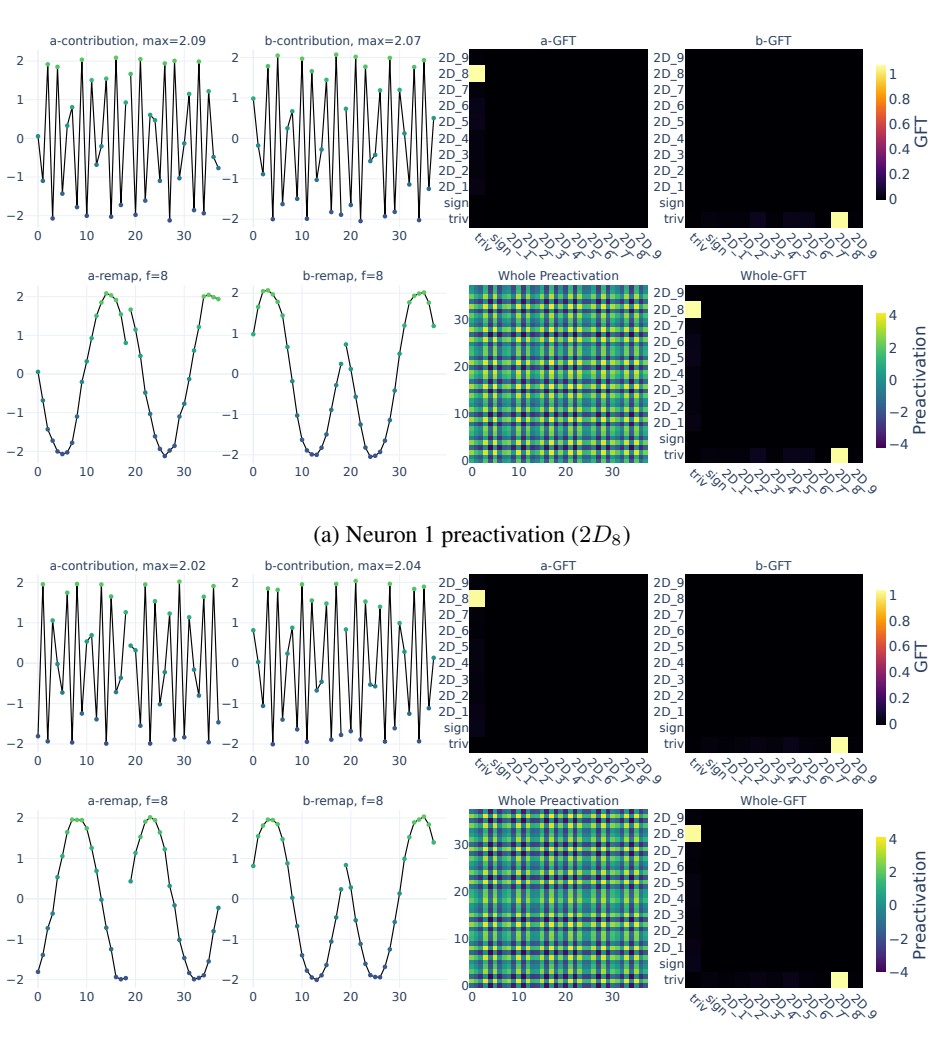

(a) Neuron 1 preactivation ($2D_8$)

(b) Neuron 2 preactivation ($2D_8$)

Figure 35: Visualization for $n = 19$, Fourier basis $2D_8$ (part 1/2). Two neurons with highest preactivation in $2D_8$.

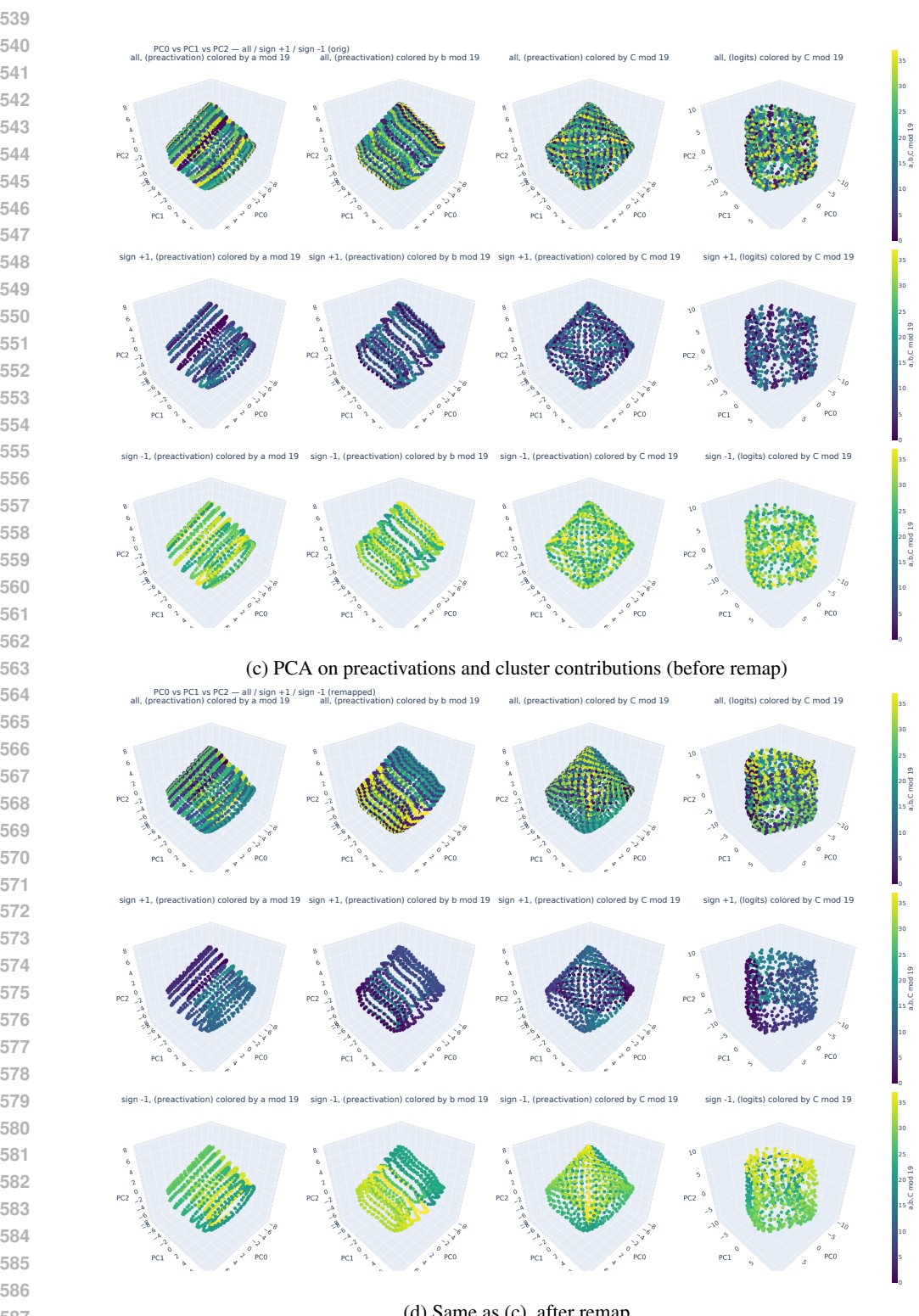

(c) PCA on preactivations and cluster contributions (before remap)

(d) Same as (c), after remap

Figure 35: Visualization for $n = 19$, Fourier basis $2D_8$ (part 2/2). PCA of preactivations and cluster contributions to logits, colored by $a$, $b$, and $C \bmod 19$, shown before and after remapping.

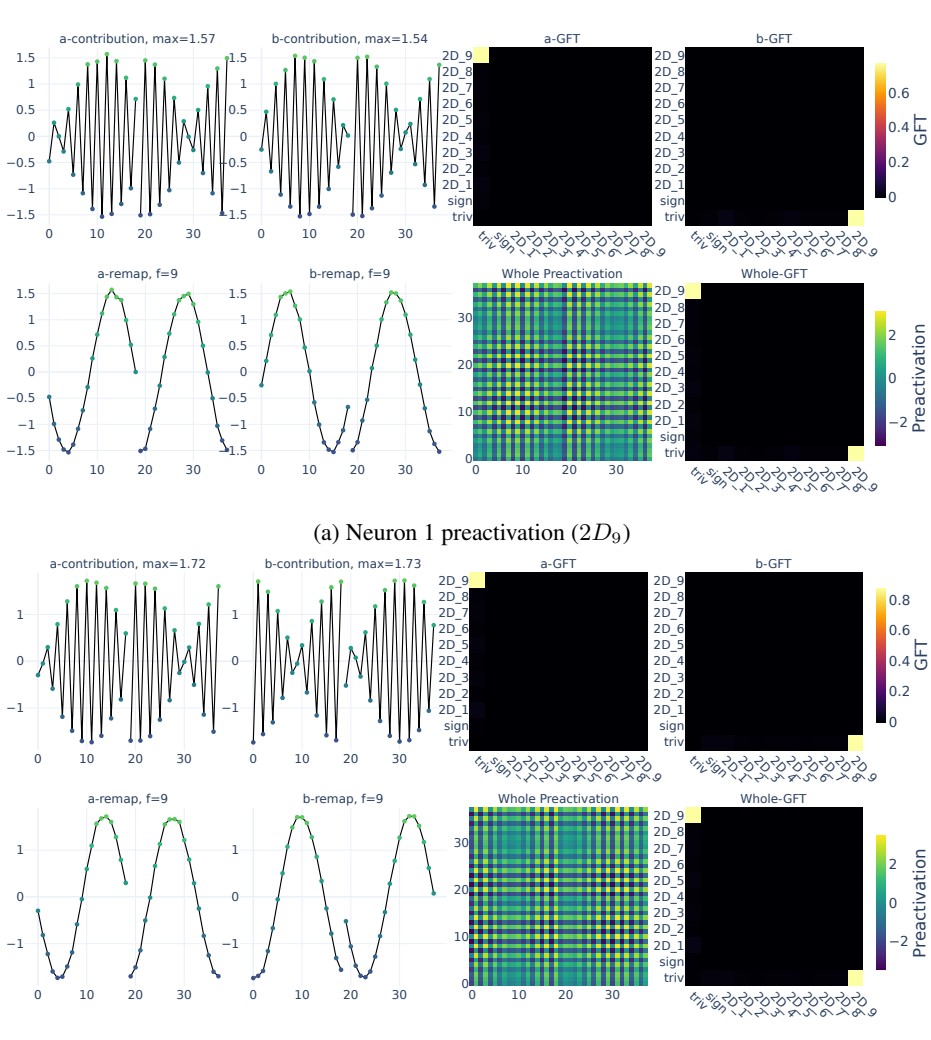

(a) Neuron 1 preactivation ($2D_9$)

(b) Neuron 2 preactivation ($2D_9$)

Figure 36: Visualization for $n = 19$, Fourier basis $2D_9$ (part 1/2). Two neurons with highest preactivation in $2D_9$.

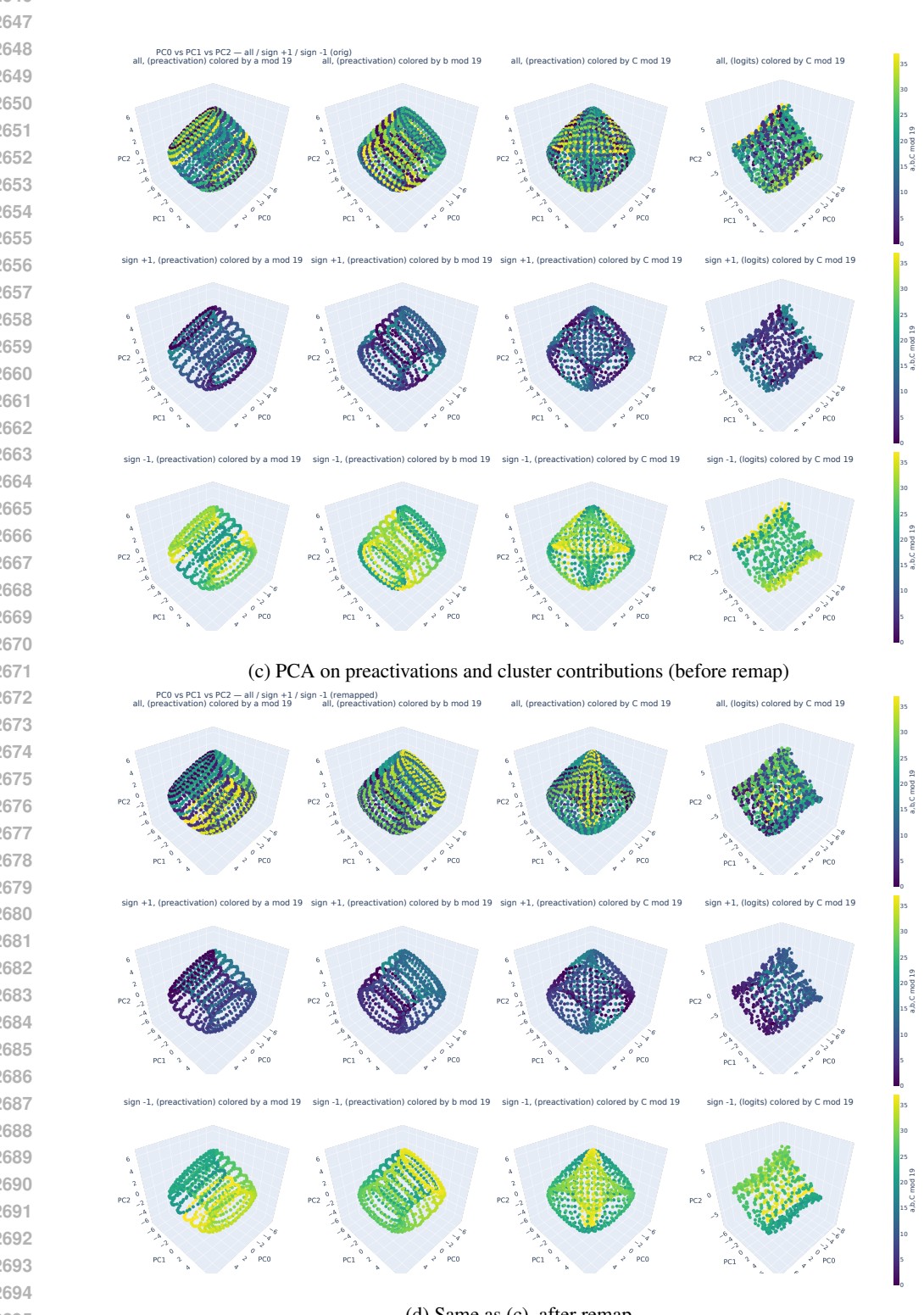

(c) PCA on preactivations and cluster contributions (before remap)

(d) Same as (c), after remap

Figure 36: Visualization for $n = 19$, Fourier basis $2D_9$ (part 2/2). PCA of preactivations and cluster contributions to logits, colored by $a$, $b$, and $C \bmod 19$, shown before and after remapping.

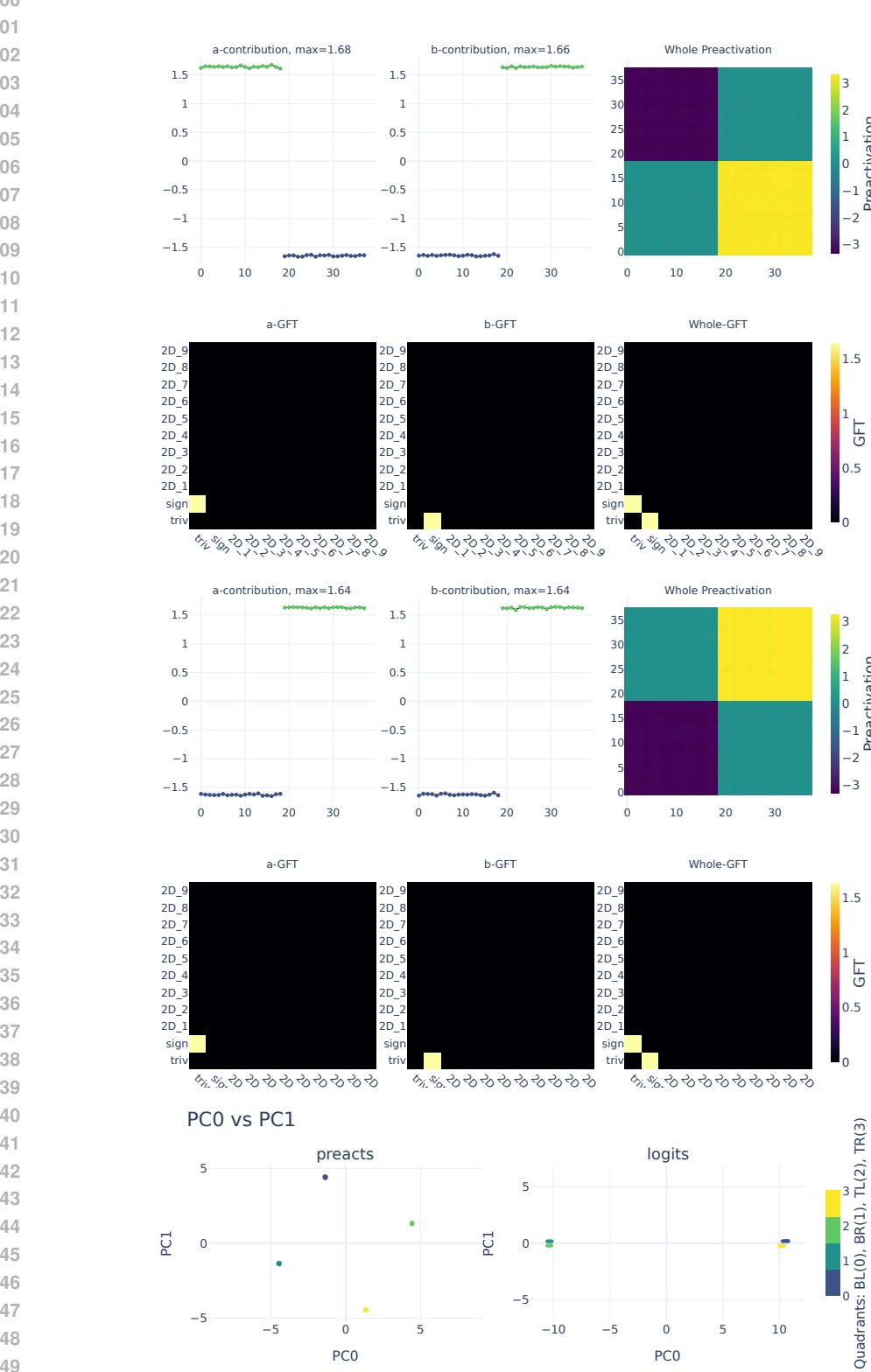

Figure 37: Visualization for $n = 19$. Top: two neurons with highest preactivation in Fourier basis `sign`. Bottom: PCA of preactivations and cluster contributions to logits, colored by quadrant (BL, BR, TL, TR) of $(a, b)$ in the $2p \times 2p$ grid.

### B.6.4 QUADRANTS AFFECT CAYLEY GRAPH STRUCTURE

Fig. 38 shows how the equivalence class of $C$ changes the angle the points in the same equivalence class are stored in within the network.

### B.6.5 TRANSFORMER CALEY GRAPH GENERATOR DISTRIBUTION

### B.7 HOW WERE THE PLOTS MADE

We use an $80\%$, $20\%$ train, test split, and train with Adam (Kingma & Ba, 2014), cross-entropy loss, with learning rate $0.001$ and batch size $= 2n$ for 5000 epochs.

The remapping definition that is used to normalize sinusoidal functions with frequency $f$ so that they can be plotted with frequency 1 is provided below.

**Definition 2** (Step size). $d := (\frac{f}{\gcd(f,n)})^{-1} (\mathrm{mod}\, \frac{n}{\gcd(f,n)})$, *where the modular inverse is used.*

**Definition 3** (Remapping: frequency normalization). *Consider the function $h(x) = \cos(2\pi f x/n)$ with frequency $f$. We define a new function $g$, allowing us to perform something analogous to a change of variables using the step size $d$: $g(d \cdot x) = h(x) \iff g(x) = h(d^{-1} \cdot x)$.*

### B.8 ALGORITHM TO RECOVER THE CAYLEY GRAPH GENERATORS

Figure 40 shows the algorithm we use to compute the generators for the Cayley graph. The step-size gives the rotational generator, but the reflection generator comes from this algorithm.

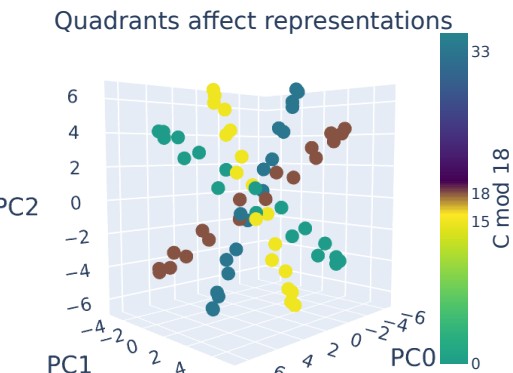

Figure 38: PCAs of the preactivations, filtered to only include data where the answer $C \in \{0, 15, 18, 33\}$. This plot shows that the sign of $C$ results in the embedding for $C$ being reflected. For example, consider $C = 0$ vs $C = 18$, which are the same element, but one is the mirror reflection of the other: the two corresponding circles are reflected across a plane that exists in the middle between them. The same is true for $C = 15$ vs $C = 33$.

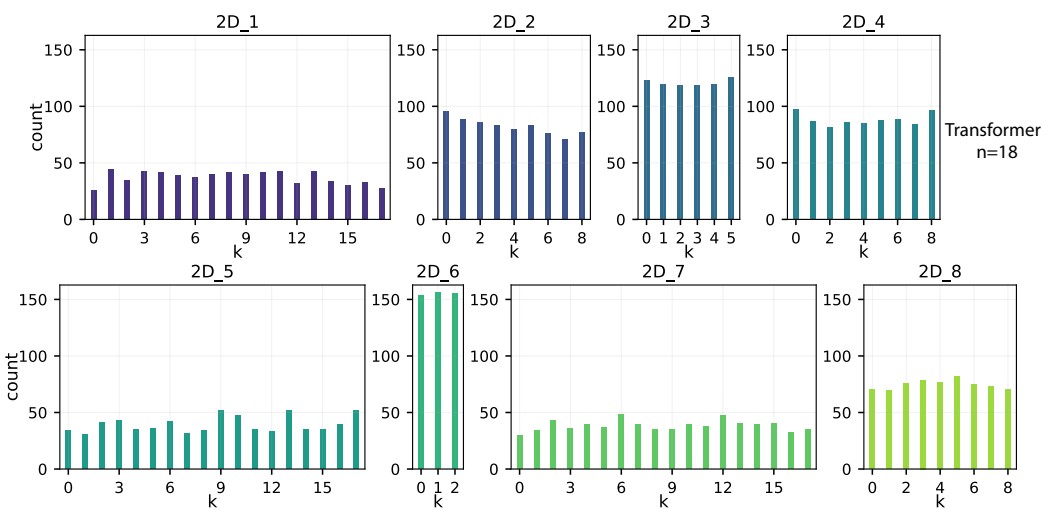

Figure 39: Distributions of how often each generator ($sr^k$) is used for each Cayley-graph type in the transformer. All distributions are approximately uniform, mirroring the behavior observed in the MLP.

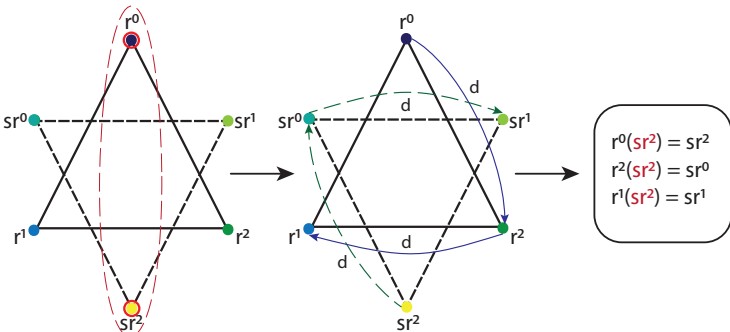

Figure 40: **Clockwise pairing on a stripe** ($g = 3$). Points lie in a 2D embedding (schematic PCA view). Each hexagram is one fixed-$b$ stripe: the blue triangle is the rotation family $\{r^0, r^1, r^2\}$ (indexed by $a$), and the green/yellow triangle is the reflection family $\{sr^0, sr^1, sr^2\}$. *Left:* pick the farthest cross-family pair as a common start (here $r^0$ and $sr^2$). *Middle:* from this start, move *clockwise* through both families with the same modular step $d$, pairing positions encountered at the same time; labels "$d$" indicate the *step size in cyclic index space*, not Euclidean distance. *Right:* in this example, the resulting pairs satisfy a stripe-wise constant-sum rule $i + j \equiv k \pmod{g}$, where $i \in \{0, 1, 2\}$ indexes $r^i$ elements, $j \in \{0, 1, 2\}$ indexes $sr^j$ elements, and the constant $k$ indicates the generator $sr^k$ (here $k = 2$; e.g., $r^0(sr^2) = sr^2$, $r^2(sr^2) = sr^0$, $r^1(sr^2) = sr^1$).

### B.9 POOR LOCAL MINIMA—DEGENERATE PHASE CONFIGURATIONS

We think that one of the cooler things about our methodology is that it even works to detect when poor local minima are learned.

If the phase configurations of the sinusoidal functions learned by neurons aren't sufficiently diverse, the Cayley graph learned by the cluster of neurons will have only one generator. This will be a rotation. Thus, such a neural representation can't tell if the answer to a multiplication is in the front or back (sign +1 or sign -1) part of the dihedral group.

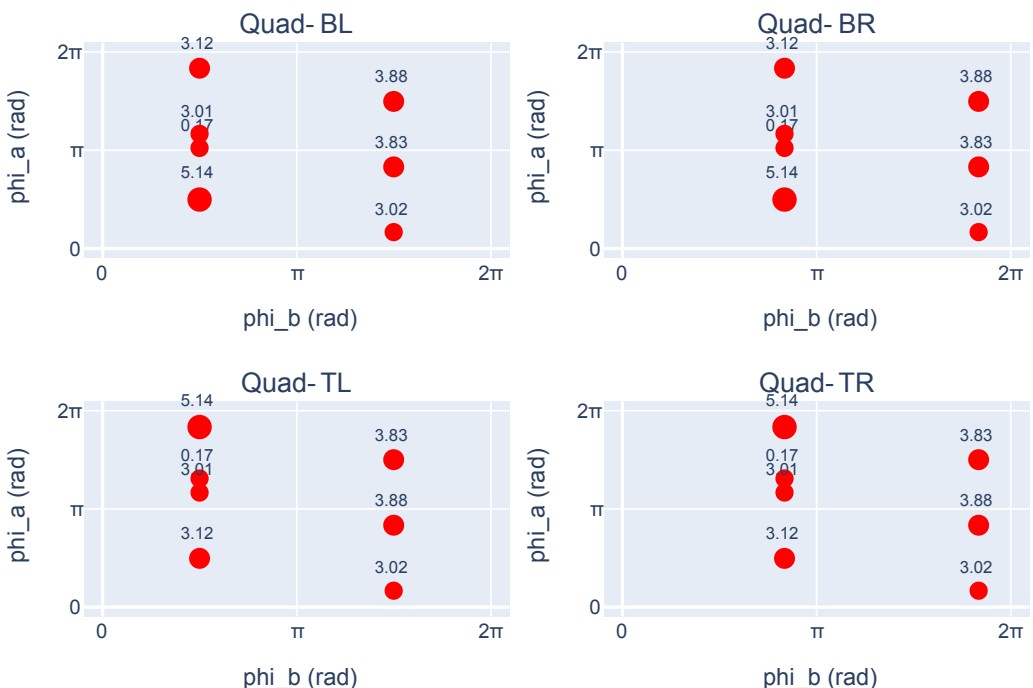

Figure 41: Here we see that the phases for this cluster of neurons in the top left (TL), bottom left (BL), bottom right (BR) and top right (TR) quadrants are aligned on two lines in every quadrant that the neurons activate on. In the case of non-degenerate neuron clusters, the phases appear to have something close to a diverse distribution over the grid.

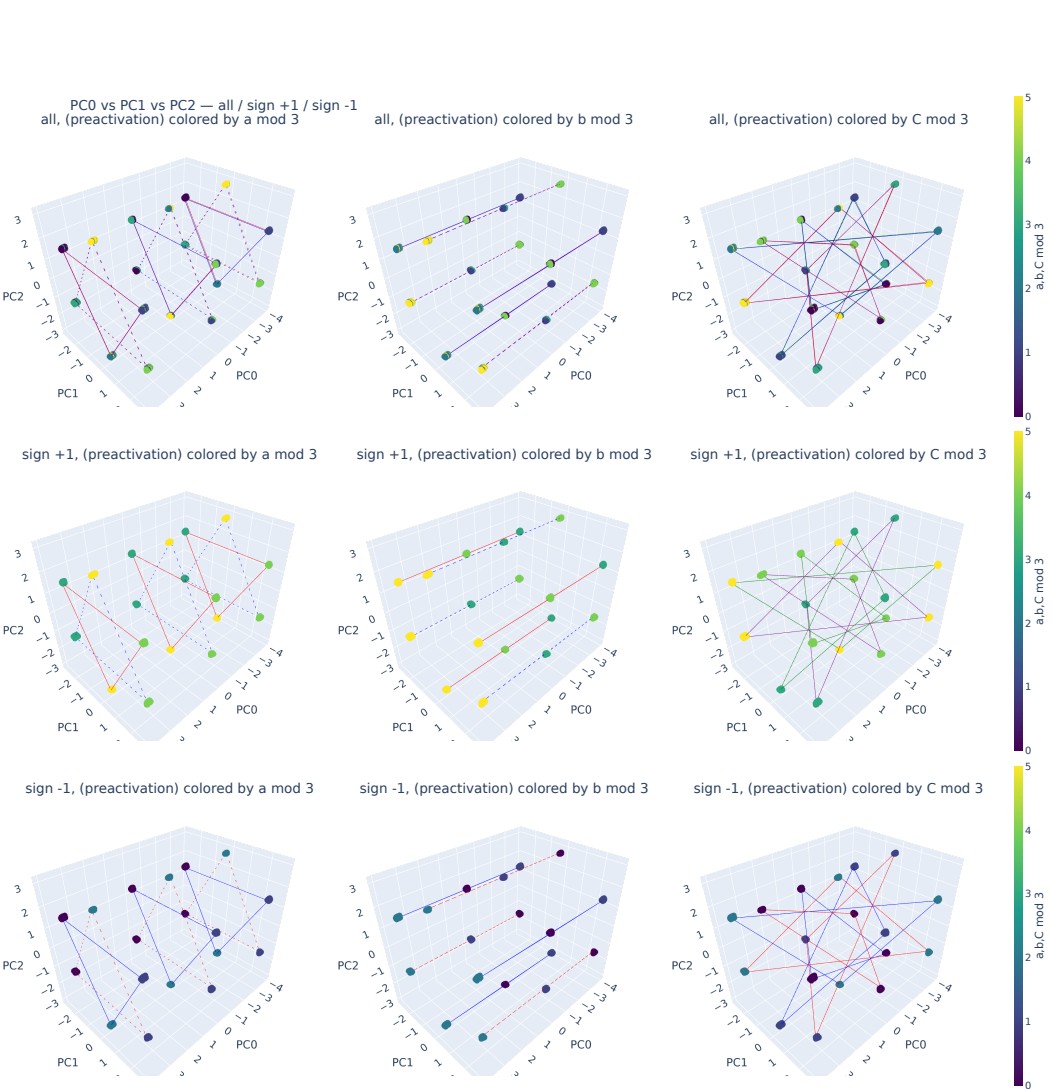

Figure 42: Here we see that there are only three $D_3$ structures, compared to six in the main paper. This is because of the poor distribution of phases seen in fig 41.

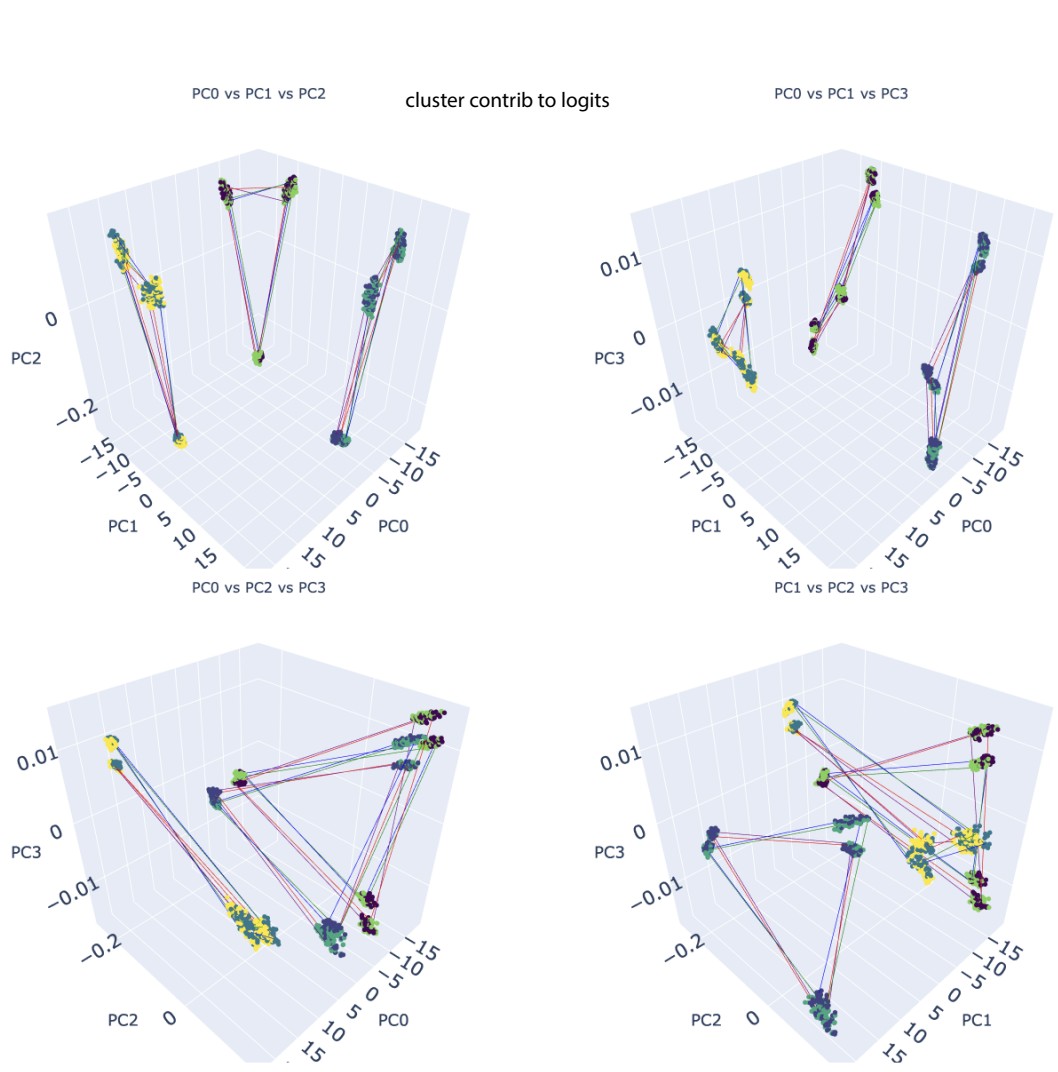

Figure 43: The second layer reveals that cluster of neurons fails to distinguish equivalence classes in the front vs back. Each triangle has two equivalence classes of the correct answer $C$ on the vertices. This is because this cluster is failing to tell the difference between answers in sign +1 and sign -1. This corresponds to a poorer local minima than if it could distinguish between sign +1 and sign -1 (because such a solution would better reduce cross entropy loss).

## B.10 THEOREMS AND LEMMAS PROVING $\mathcal{O}(\log(n))$ CAYLEY GRAPHS ARE ENOUGH

## C ETHICS STATEMENT

This work adheres to the ICLR Code of Ethics. In this study, no human subjects or animal experimentation was involved. All datasets used, including *Dihedral Multiplication*, were sourced in compliance with relevant usage guidelines, ensuring no violation of privacy. We have taken care to avoid any biases or discriminatory outcomes in our research process. No personally identifiable information was used, and no experiments were conducted that could raise privacy or security concerns. We are committed to maintaining transparency and integrity throughout the research process.

## D REPRODUCIBILITY STATEMENT

We have made every effort to ensure that the results presented in this paper are reproducible. All code and datasets are included as anonymized supplementary materials to facilitate replication and verification. The experimental setup, including training steps, model configurations, and hardware details, is described in detail in the paper in section B. We have also provided a full description of training on the dihedral group, to assist others in reproducing our experiments. We believe these measures will enable other researchers to reproduce our work and further advance the field.

