# OpenReview forum: "Deep Neural Networks Divide and Conquer Dihedral Multiplication"
_ICLR.cc/2026/Conference — Submitted to ICLR 2026_

### Official Review · Reviewer_EEmm · 2025-10-25

**Soundness:** 3
**Presentation:** 2
**Contribution:** 2
**Rating:** 4
**Confidence:** 3

**Summary:**

The paper provides evidence for the universality hypothesis by demonstrating that both multi-layer perceptrons (MLPs) and transformers learn the same abstract divide-and-conquer algorithm to solve dihedral group multiplication. The authors employ a methodology using the Group Fourier Transform (GFT) to cluster neurons with similar activation patterns. Subsequent Principal Component Analysis (PCA) on these clusters reveals that the network's distributed neural representations form distinct, geometric structures identified as Cayley graphs. The global algorithm then works by combining the outputs of these different Cayley graph representations—each solving a simpler subproblem—to maximise the logit for the correct answer.

**Strengths:**

1. Novel Perspective on Neural Representations: The shift from analysing single neurons to the emergent structures formed by a neuron cluster is more appropriate under the superposition hypothesis. This goes beyond prior work that modelled individual neuron activations with sinusoidal functions to provide evidence for discrete distributed representation, showing how sinusoidal components collectively build a higher-level geometric object (a Cayley graph).
2. Strong Methodology: The task of dihedral multiplication is rigorously analysed from a theoretical perspective to arrive at an appropriate hypothesis for the behaviour of these deep neural networks. The subsequent implementation of GFT and PCA provides compelling evidence that the proposed algorithm is being implemented. Furthermore, the replication of results across different architectures, random initialisations, and numerous (i.e., different values of n) problem instances strongly supports the claim.

**Weaknesses:**

1. Claim of Universality: Although the authors consider multi-layer perceptrons and transformers, the consideration of more fine-grained architectural choices (i.e., depth, width, activation function) is not considered. To provide comprehensive evidence for the universality hypothesis, the effects of these architectural choices, as well as optimiser, training hyperparameters, etc, would need to be considered.
2. Theoretical Justification: There is little discussion as to why deep neural networks learn this particular algorithm compared to other algorithms for performing dihedral multiplication.
3. Presentation: A significant portion of the paper is devoted to introducing the problem, reviewing related work, and presenting figures. It would perhaps be more appropriate to provide the results of various ablation studies to support the claim of universality.

**Questions:**

1. Does the emergence of this algorithm appear at a particular point in training, say as a grokking effect?
2. How do other architectural factors – such as activation function, depth and width – influence the emergence of the divide and conquer algorithm?

---

> ### Author Response · Authors · 2025-11-27
>
> We thank the reviewer for their thoughtful and constructive feedback. We appreciate the recognition of our shift from neuron-by-neuron analysis to the direct inspection of the geometric structure of neural representations, as well as the positive assessment of our GFT/PCA methodology and its robustness across architectures, seeds, and problem instances.
>
> > Weakness 1. Claim of Universality: Although the authors consider multi-layer perceptrons and transformers, the consideration of more fine-grained architectural choices (i.e., depth, width, activation function) is not considered. To provide comprehensive evidence for the universality hypothesis, the effects of these architectural choices, as well as optimiser, training hyperparameters, etc, would need to be considered.
>
> We sincerely thank you for pointing out that more comprehensive evidence at finer granularities will strengthen the paper by supporting our claim of discovering the universal solution learned by networks learning dihedral multiplication. We fully agree and thus, we’ve added all of the quantitative experiments you suggested and more. We find that both MLPs and transformers universally implement our claimed interpretation across: depths, widths, different activations, and different training parameters (learning rates and weight decays).
> Furthermore, since our paper presents three interpretations, with each at a different scale of abstraction, we introduce two new metrics for these universality experiments, in addition to traditional ablations. The coset clustering score measures whether the neural representations are Cayley graphs, with their vertices containing all points in the same equivalence class clustered tightly together, as we claim. It computes the average distance to all points within the same coset equivalence class and compares this against the average distance to points outside the coset. We define:
>
> $$
> s_{\mathrm{cluster}} = 1 - \frac{\mathbb{E}[\text{distance within the same coset}]}{\mathbb{E}[\text{distance across different cosets}]}.
> $$
>
> Furthermore, we introduce the coset disjointness score, which is the empirical probability that the vertices on the Cayley graphs are “pure”. This is computed by computing the minimum enclosing ball for all points in each equivalence class, and then computing the fraction of balls whose in representation space that are non-overlapping. If the balls aren’t overlapping, then all the points in each coset equivalence class are near to eachother and the coset disjointness score is 1.
>
> Across all hyperparameter sweeps, we report:
>
> (i) **neuron-wise sinusoid fits**, via the mean ($R^2$) across neurons, showing that individual units continue to implement the roles we claimed
>
> (ii) **coset clustering score**, ensuring that points in the same coset are clustered tightly together
>
> (iii) **coset disjointness**, ensuring that points in the same coset are clustered tightly together without any invading points from external cosets.
>
> We’re happy to report high scores on all of these metrics, except at the edge of the hyperparameter region, which corresponds to where networks struggle to learn to generalize to the test set. This makes sense because the solutions at the edge correspond to bad hyperparameters, and it’s well known that hyperparameter quality influences the quality of the learned local minima and ability for trained networks to generalize onto the test set. Furthermore, these plots show a green checkmark on the hyperparameter combinations where deletion of every neuron and replacement every neuron with its best fit resulted in the network maintaining 100% test accuracy. We mark a cell with a **red X** if the network fails to learn, *i.e.* reach 100% test accuracy. If it does generalize, but replacing all neurons with their best fitted sinusoid functions does *not* maintain 100% accuracy on the full dataset (train + test), we mark the cell with a **purple dot**. Note, purple dots are our way of showing how robust our interpretation is, because if a single datum becomes misclassified (MLP) or mispredicted (transformer) a purple dot will appear.
>
> Please see the anonymous link https://anonymous.4open.science/r/plots-D7C4/ for the full set of plots, which we’ve also added to the revised paper.
>
> 1. **Depth** (one- and two- layer MLPs/Transformers with ReLU). (Appendix B.4, fig.12)
> We sweep learning rate and weight decay for 1- and 2-layer ReLU MLPs and Transformers. Each cell in the grid corresponds to a single (lr, wd) configuration.
>
> 2. **Width.** (Appendix B.4, fig.13 and fig. 14). We sweep over different hidden widths for both MLPs and Transformers, again over grids of (lr, wd). For each width, we reuse the same neuron-wise sinusoid fits and report mean (R^2), the coset clustering score, and coset disjointness. We observe the same pattern of green checkmarks over a broad region of hyperparameters for all widths tested, indicating that the solution we proposed is stable under width changes.

---

> > ### Author Response · Authors · 2025-11-27
> >
> > 3. **Activation function (pure quadratic vs ReLU).** (Appendix B.4, fig.15)
> >
> >  Finally, to ensure universality across very different forms of nonlinearity, we train (one-layer) MLPs and Transformers with **quadratic** activation functions. We choose quadratic activations due to their great difference from ReLU, and due to the prior work training networks with them on other group multiplications.
> >
> > These new experiments directly address the reviewer’s request for a more systematic exploration of architectural and hyperparameter choices and we believe these results greatly strengthen the paper, and thank you for your suggestion.
> >
> > Our paper now establishes universality across four scales.
> >
> > **Across datasets.** Given that our results establish universality with the studies of McCracken et al., and Stander et al., which are on totally different datasets, we believe our results should be robust to hyperparameter changes. We believe the fact that individual neurons learn coset circuits on three totally different datasets, being modular addition, permutation group multiplication, and dihedral multiplication is remarkable.
> >
> > **Across architectures with different training objectives**. We established universality between two totally different architectures. One is MLPs trained to directly classify the output, and the other is autoregressive transformers trained to predict the next token. The transformers also include residual connections, which the MLPs don’t.
> >
> > **Across problem complexity.** We established universality over different datasets, being many orders of magnitude of dihedral group size. This is our plot that found that consistently, O(log(n)) neural representations (Cayley graphs) were learned. This implies our main result, being the divide and conquer algorithm that requires learning O(log(n)) Cayley graphs, is learned consistently across scales of problem size.
> >
> > **Across training training and architectural hyperparameters.** We now report universality across hyperparameters, being widths, learning rates, weight decays, quadratic or ReLU activations, and depths (1 and 2).
> >
> > We again thank the reviewer for this suggestion as we believe it greatly strengthens the paper.
> >
> > > Weakness 2. Theoretical Justification: There is little discussion as to why deep neural networks learn this particular algorithm compared to other algorithms for performing dihedral multiplication.
> > This is out of the scope of our study. We seek to definitively answer what is learned by networks trained on dihedral group multiplication at multiple levels of abstraction.
> >
> > We note that out of the many papers studying modular addition, no theoretical explanation exists yet for why neurons in deep neural networks learn sinusoidal functions (Morwani et al. claimed it was to maximize the margin by learning sine functions of every sinusoidal frequency (n-1)/2 total frequencies learned, but McCracken et al. showed DNNs use O(log(n)), and don’t maximize the margin.
> >
> > We speculate that our methodologies’ ability to show interpretations at three scales of abstraction may help researchers devise a theoretical explanation for why these solutions are learned since it gives interpretations at more scales of understanding than just neuron-by-neuron.
> >
> > That said, do you have something specific in mind with respect to your statement “compared to other algorithms for performing dihedral multiplication”? If you mean the group composition by representations (GCR) algorithm that Chughtai et al. (https://openreview.net/pdf?id=jCOrkuUpss) conjectured DNNs would universally learn on group tasks, it’s the case that our work actually stands to falsify their claim. We find coset circuits, not matrix multiplication of irreducible group representations (GCR) to be learned universally. Note that the transformers and MLPs Chughtai et al. train are the same ones we train.

---

> > > ### Author Response · Authors · 2025-11-27
> > >
> > > > Presentation: A significant portion of the paper is devoted to introducing the problem, reviewing related work, and presenting figures. It would perhaps be more appropriate to provide the results of various ablation studies to support the claim of universality.
> > >
> > > See 1. For now we have added the hyperparameter tuning and ablation plots to the appendix, but we agree and promise to add the hyperparameter tuning and ablations plot (over depths) to the main paper in a revision we will soon post. We will also reference the appendix section dedicated to these experiments for more experiments supporting universality. We will also do our best to streamline the introduction, but should the paper be accepted, it will be easy to add our new ablations over hyperparameters plots that you suggested.
> > >
> > > Questions.
> > >
> > > > 1. Does the emergence of this algorithm appear at a particular point in training, say as a grokking effect?
> > >
> > > Similarly to our explanation in weakness 2, we believe that our work should focus on explaining what is learned. As grokking involves the study of training dynamics, it’s more the study of *how and why* things are learned.
> > >
> > > Our goal was to provide the highest resolution understanding of neural representations that’s ever been achieved on a toy dataset, and to hopefully show that our understanding is universal at multiple scales of abstraction across very different architectures (and now also very different experimental settings). We fear added discussion dedicated to training dynamics would dilute the strength of our explanation about what is learned.
> > >
> > > > 2. How do other architectural factors – such as activation function, depth and width – influence the emergence of the divide and conquer algorithm?
> > >
> > > In weakness 1 we added plots showing our claims are robust and universally learned across a very large volume of training hyperparameters, and repeated this experiment across depths, architectures, widths, and even with quadratic (very different) activations. In all experiments, we found the majority of the volume where models can learn to generalize to the test set agrees with our claims. The only volume where our claims sometimes failed our strict requirement of maintaining 100% test set accuracy after ablation was near the edge of where networks stop learning. We believe this strongly supports our claims, and greatly strengthens the paper.
> > >
> > > **Concluding remarks.** We’re glad that you thought we provided a novel perspective on Neural Representations. This is exactly what we hoped readers would take away from the paper. We hope that we’ve adequately addressed your questions and concerns. If we haven’t, please help us understand what would further strengthen our claims. We think we’ve made a significant contribution to the literature on interpreting group multiplications, and believe our framework will generalize to helping with the interpretation of other, more complicated groups.
> > >
> > > We invite you to read our response to reviewer RG3S because they were also instrumental in suggesting changes that strengthened our paper. We believe the experiments the two of you requested have resulted in a much stronger paper. Again, we sincerely thank you for your time and effort in reviewing our work.

---

### Official Review · Reviewer_RG3S · 2025-10-29

**Soundness:** 3
**Presentation:** 3
**Contribution:** 2
**Rating:** 2
**Confidence:** 4

**Summary:**

The paper investigates how deep neural networks (MLPs and transformers) learn to perform multiplication in the dihedral group, which is
a non-commutative finite group representing the symmetries of an n-gon.

The authors claim that neural networks learn a divide-and-conquer algorithm for this operation.
Their internal representations correspond to Cayley graphs and coset structures of the group. This supports a “universality hypothesis”: networks trained on algebraic tasks consistently discover similar algorithmic structures,
regardless of architecture or random seed.

**Strengths:**

- Very interesting approach to gain insight into learning
- The “divide-and-conquer” hypothesis is quite reasonable and insightful
- Comprehensive experimentation: many thousands of training runs were studied
- High reproducibility: many experiments over random seeds
- The manuscript is well written

**Weaknesses:**

The paper is essentially an exploratory analysis rather than a hypothesis-driven study.
No explicit, testable hypotheses are formulated or quantitatively evaluated.
Most conclusions rest on visual inspection of PCA- and Group Fourier Transform -derived figures, whose interpretations are ambiguous.
It remains unclear whether these structures reflect genuine inductive biases or are artifacts of the analytical lens itself - especially given that the Group Fourier Transform already encodes the group structure being “discovered.”
This raises a risk of circular reasoning.
To be more convincing, the authors should (i) define explicit hypotheses about what structures should appear under what conditions, (ii) quantify the strength or prevalence of these patterns statistically, and (iii) include negative or counterexample experiments where no such structure is expected.
Such controls would clarify whether the observed patterns genuinely reflect learned algorithmic structure rather than the setup or analysis.

In this sense, the approach is a visually appealing first step, but scientifically, it is wholly unclear what the results even mean.

**Questions:**

1. Could the authors formulate explicit, testable hypotheses about what structures (e.g., Cayley graphs, cosets, scaling laws) should appear under specific conditions?
2. How would those hypotheses be falsified -- what results would not support the proposed divide-and-conquer or universality interpretation?
3. Have the authors considered applying the same training and analysis pipeline to a problem where such structured behavior is not expected (e.g., random multiplication tables, corrupted group laws, or non-algebraic mappings)?
4. Would the same Cayley-graph patterns and O(log n) scaling appear in those cases?
5. If so, how would that affect the interpretation of the current findings?
6. Given that the Group Fourier Transform is defined directly in terms of the dihedral group’s representation theory, how can the authors ensure that the observed structure is not a byproduct of analyzing activations in a basis already aligned with that structure?
7. Can the authors provide quantitative metrics (e.g., distances between embeddings and Cayley-graph adjacency matrices, reproducibility statistics across seeds) rather than relying primarily on visual inspection?

---

> ### Author Response · Authors · 2025-11-22
>
> We’re glad that you liked our quantitative experiments over thousands of random seeds, and thank you for your review. It seems that you have three concerns and we’re confident we can address them adequately. We really want to thank you for your review, because it has been very useful and made it clear to us how, with minor revisions, clarifications and additions, we can greatly improve the paper. We’re very happy that you see the strength of our novelty and said we have a “Very interesting approach to gain insight into learning”.
>
> > Concern 1. “In this sense, the approach is a visually appealing first step, but scientifically, it is wholly unclear what the results even mean.”
>
> We realized we cut out remarks that were important to contextualize how our work fits into the literature. We’ve added them back in the revision, but contextualize our work here: the paper that inspired our study was McCracken et al., which surprised us by rigorously presenting that deep neural networks (DNNs) learn a divide-and-conquer algorithm, being the Chinese remainder theorem, to solve modular addition. McCracken et al, proposed a conjecture: that DNNs would divide and conquer all group tasks by utilizing cosets. They proposed this could be falsified by studying related, but different group tasks like the dihedral group multiplication which isn’t commutative, thus is very different to modular addition.
>
> McCracken et al. made no mention of what the structures corresponding to neural representations would be. Our work finds the neural representations are Cayley graphs. Therefore, we can update McCracken et al.’s conjecture: DNNs will divide and conquer group multiplications by intersecting differently generated Cayley graphs, which are the neural representations composed of neurons that individually learned cosets.
>
> **We give three independent levels of abstraction where universality may be falsified** by studying DNNs learning group multiplications.
> - Globally: what are the solutions learned by DNNs? Can work find DNNs not using divide-and-conquer algorithmic structure?
> - Structurally: what are the neural representations? Can work find they’re not always Cayley graphs?
> - Neuron-wise: what are the activations? Can work find they’re not always cosets?
>
> **Why is this important and why are we excited about it?** There are three major deep learning hypotheses influencing thoughts about what DNNs learn. (i) The first is the universality hypothesis, that DNNs trained on similar data will make use of similar principles (Olah et al.; Li et al.). McCracken et al.’s conjecture only relates to this hypothesis. Our work sheds light into universality and the: (ii) manifold hypothesis, that DNNs efficiently learn (despite the curse of dimensionality) by finding low dimensional manifolds that the dataset lives on (Goodfellow et al.) and (iii) platonic representation hypothesis, which posits that architectures trained on different learning objectives at large scales, learn a platonic representation, meaning that distances between datapoints in different classes of data “converge” (Huh et al.).
>
> Our work offers the ability to inspect and falsifiably investigate these three major hypotheses simultaneously. Should our conjecture (and McCracken et al.’s) be verified across a diverse breadth of group tasks, it has the potential to show that universality exists at multiple scales of abstraction, and would suggest there are universality classes of particular manifolds that are learned by DNNs. Furthermore, it would suggest that neural representations universally order points on group tasks. Therefore, we claim our work is of significant interest as it provides many roads toward understanding the nature of solutions learned by DNNs.
>
> To clarify how our work relates to the platonic representation hypothesis, recall that it posits different architectures (even with different training objectives) learn neural representations that embed points “similarly”. Our novel methodology explicitly shows that autoregressive transformers trained with a next-token prediction objective (sequence generation) learn the same neural representations (Cayley graphs) as feed-forward MLPs trained with a direct classification objective. Our new quantitative experiment below confirms that the points on these graphs are distributed with similar distances even between architectures. Additionally, if MLPs learn the same Cayley graph structure as a transformer, then both models have learned the same generators, and thus the vertices of these graphs will have the same structure/order. This follows from the adjacency matrix of a Cayley graph being fully determined by the generators of the graph.

---

> > ### Author Response · Authors · 2025-11-22
> >
> > > Concern 2. How do we know our methodology, which requires the group Fourier transform in order to identify neurons collaborating to form a neural representation, doesn’t guarantee that we will see Cayley graph structure?
> >
> > It follows mathematically from the fact that the irreducible representations of a group form a basis for functions on a group. Thus, if a DNN were to learn to solve the task by learning structure that wasn’t group structure, the GFT wouldn’t concentrate cleanly (sparsely) on the irreducible representations of the group.
> >
> > We decided to demonstrate this via your suggestion. We added an experiment on random multiplication tables to appendix B.5. We perform our same methodology, finding the structures DNNs learn on random group multiplications aren’t well-formed when collected via the dihedral GFT. We put fig. 9 to show a neuron trained on a random multiplication table has a diffuse (non-sparse) Fourier spectra, which doesn’t concentrate on any representations of the dihedral group. This means whatever was learned by the neuron doesn’t “play nice” with the basis functions of the group. Thus, it requires many linear combinations of different irreducible representations (basis functions) to be “composed”.
> >
> > We now quantify coset geometry using:
> > - **Coset clustering score** $s_{\mathrm{cluster}}$: we compute the mean distance between points within the same coset and the mean distance between points in different cosets. We define the score as
> >    $$
> >     s_{\mathrm{cluster}} = 1 - \frac{\mathbb{E}[\text{distance between points within the same coset}]}{\mathbb{E}[\text{distance between points across different cosets}]},
> >   $$
> >
> >     which gives that larger values indicate tighter and better-separated coset clusters.
> > - **Coset disjointness**: for each coset, we compute the minimum enclosing ball that contains all points in the coset. The coset disjointness is the percentage of cosets whose enclosing balls are strictly disjoint; we report the mean fraction across cosets.
> >
> > The statistics below give that the distance of datapoints in cosets in MLPs are very similar to the distances in transformers trained on dihedral groups. This contrasts to the random multiplication, which using the same methodology, doesn’t result in clean dihedral group Cayley graph structure being found, the distances between points are not cleanly separated into coset structure.
> >
> > (mean $\pm$ std. over 1000 seeds) for $n=18$:
> >
> > | arch        | law        | n  | s_cluster         | disjointness       |
> > |------------|------------|----|-------------------|--------------------|
> > | MLP        | dihedral   | 18 | 0.993 ± 0.001     | 1.000 ± 0.001      |
> > | MLP        | random     | 18 | 0.240 ± 0.055     | 0.130 ± 0.107      |
> > | Transformer| dihedral   | 18 | 0.974 ± 0.071     | 0.977 ± 0.082      |
> > | Transformer| random     | 18 | 0.255 ± 0.054     | 0.140 ± 0.112      |
> >
> > As illustrated quantitatively above and qualitatively in Figs. 9-10 of the appendix, using a random multiplication table, while applying the same methodology, causes the coset-level geometry (tight clusters, high disjointness) to almost entirely disappear. This supports the interpretation that the structures we observe are tied to the learned dihedral multiplication law and are not generic artifacts of PCA or the GFT analysis pipeline.
> >
> > Additionally, this experiment serves to quantify our qualitative observations, and substantially strengthens our claims that Cayley graphs are learned because we find that all the points in the same coset are indeed together on each vertex of the Cayley graph (since the score and disjointness are high)..
> >
> > > Concern 3. Can the authors provide quantitative metrics (e.g., distances between embeddings and Cayley-graph adjacency matrices, reproducibility statistics across seeds) rather than relying primarily on visual inspection?
> >
> > It is possible to reconstruct a Cayley-graph adjacency matrix just by knowing the generators of the Cayley graph. The generators are the most compressed way of communicating this information.
> >
> > However, we realized from your remark “the approach is a visually appealing first step.” that it’s likely you’re requesting a quantitative verification of our visually appealing qualitative results. Originally, we believed that our quantitative analysis where we recover the generators was sufficient, but it wasn't without the experiment above. Our algorithm to retrieve the generator of the Cayley graph assumes that all points in cosets in Cayley graphs are close to each other. Thus, we added a method to quantitatively prove this over 1000 random seeds (see table in Concern 2). With this empirically verified, the qualitative plots of each neural representation now serve only to show the reader how the Cayley graphs look visually.
> >
> > We believe that our main argument, that totally different architectures universally learn the same neural representations (Cayley graphs) has now been concretely and robustly demonstrated.

---

> ### Author Response · Authors · 2025-11-22
>
> Questions:
>
> > 1. Could the authors formulate explicit, testable hypotheses about what structures (e.g., Cayley graphs, cosets, scaling laws) should appear under specific conditions?
>
> See concern 1. We now make an explicit conjecture, and explain how our work adds a new level of insight into McCracken et al.’s conjecture. We explain how our work gives opportunities to learn more about general hypotheses beyond universality.
>
> Note: as far as the log(n) scaling result goes, it’s possible for divide and conquer algorithms to utilize e.g. O(n^2) like selection sort. We only present this because we think it’s beautiful that O(log(n)) neural representations are required as it’s an interesting quantification for how efficient the solutions of DNNs learning this task are.
>
> > 2. How would those hypotheses be falsified -- what results would not support the proposed divide-and-conquer or universality interpretation?
>
> See concern 1.
>
> > 3. Have the authors considered applying the same training and analysis pipeline to a problem where such structured behavior is not expected (e.g., random multiplication tables, corrupted group laws, or non-algebraic mappings)?
>
> See concern 2.
>
> > 4. Would the same Cayley-graph patterns and O(log n) scaling appear in those cases?
>
> No, they would not (as demonstrated by concern 2’s models trained on random multiplications).
>
> > 5. If so, how would that affect the interpretation of the current findings?
>
> N/A.
>
> > 6. Given that the Group Fourier Transform is defined directly in terms of the dihedral group’s representation theory, how can the authors ensure that the observed structure is not a byproduct of analyzing activations in a basis already aligned with that structure?
>
> See concern 2.
>
> > 7. Can the authors provide quantitative metrics (e.g., distances between embeddings and Cayley-graph adjacency matrices, reproducibility statistics across seeds) rather than relying primarily on visual inspection?
>
> See concern 3.
>
> We hope that our addition of a formal conjecture and better context to how we build on prior work alleviates your concerns and shows how we envision future work can use the scientific method to study DNNs trained on group multiplications. We thank you for your suggestion to use a random multiplication table, as we believe this should make the paper clearer to readers less familiar with Fourier analysis. We also hope that our clarification of how we quantify the neural representations, and our added experiment quantifying that the Cayley graphs are indeed consistently structured convinces you that our claims our strongly supported by our experiments. We sincerely thank you for pushing us on this because we now have both quantitatively and qualitatively proven the neural representations are as beautifully structured as we claimed. We hope that our clarifications are sufficient for you to increase your score. If not, please suggest what else we can do to present this story with more strength, as we believe it should be interesting to all communities studying learned representations. We look forward to our continued discussion.
>
> McCracken, G., Moisescu-Pareja, G., Letourneau, V., Precup, D., & Love, J. (2025). Uncovering a Universal Abstract Algorithm for Modular Addition in Neural Networks. arXiv:2505.18266.
>
> Olah, C., Mordvintsev, A., & Schubert, L. (2017). Feature Visualization. Distill, 2(11), e7. https://doi.org/10.23915/distill.00007
>
> Olah, C., Cammarata, N., Schubert, L., Goh, G., Petrov, M., and Carter, S. Zoom In: An Introduction to Circuits. Distill, 5(3):e00024.001, March 2020. ISSN 2476-0757. doi: 10.23915/distill.00024.001.
>
> Li, Y., Yosinski, J., Clune, J., Lipson, H., & Hopcroft, J. (2016). Convergent Learning: Do different neural networks learn the same representations? arXiv:1511.07543.
>
> Goodfellow, I., Bengio, Y., & Courville, A. (2016). Deep Learning. MIT Press.
>  (Chapter 5.11.3)
>
> Minyoung Huh, Brian Cheung, Tongzhou Wang, and Phillip Isola. The platonic representation hypothesis, 2024. URL https://arxiv.org/abs/2405.07987.

---

> > ### Comment · Reviewer_RG3S · 2025-11-25
> >
> > Many thanks to the authors for the detailed rebuttal and the changes. I have to admit that I am very impressed by the new results and how they address the criticism from my initial report. In fact, I now think that the results are much more scientifically grounded. I will happily increase my score to 6.

---

> > > ### Author Response · Authors · 2025-11-27
> > >
> > > Thank you again for your thoughtful review and for reassessing our paper. Your feedback directly led to a much stronger version of this work, and we've since built on the ideas from our initial discussion together to answer reviewer EEmm.
> > >
> > > Since our last response, we have added new experiments requested by reviewer EEmm to strengthen the evidence for universality.  Our new experiments vary architecture depth and width, training hyperparameters, and test quadratic activations (as they're very different from ReLUs). We use the coset score and coset disjointness that emerged from our earlier discussion to measure whether the neural representations take the form our paper claims: Cayley graphs generated by group elements. We also use the R^2 of the fits to test whether individual neurons implement what we claim. We find that across the entire region of hyperparameters where networks reliably learn the task (away from the failure-to-generalize boundary), they consistently converge to the solution we claim. We feel this now provides a very robust empirical foundation for our interpretation across architectures and training settings.
> > >
> > > These results are included in the revised manuscript (Appendix B.4) and visualized at: https://anonymous.4open.science/r/plots-D7C4/
> > >
> > > We believe that fully understanding DNN solutions on toy tasks, at multiple levels of abstraction, can help answer the question “What is the space of solutions DNNs can learn?”. With that goal in mind, we would be very grateful for any further suggestions you might have for strengthening or completing the story we’re presenting, including any thoughts on the overall strength and significance of the current version.

---

### Official Review · Reviewer_oD9U · 2025-10-31

**Soundness:** 2
**Presentation:** 1
**Contribution:** 1
**Rating:** 2
**Confidence:** 2

**Summary:**

This paper studies how deep neural networks (DNNs) perform the *dihedral* group operation (the group of symmetries of a regular polygon, which includes rotations and reflections). By visualizing the neural activation using a frequency-based remapping, the authors argue that DNNs learn neural representations that correspond precisely to Caley graphs.

**Strengths:**

- First comprehensive mechanistic analysis of neural networks trained on non-commutative group tasks.
- The experiments include hundreds to thousands of random seeds, showing robustness across architectures (MLP vs transformer).
- Results scale across orders of magnitude (2-512), demonstrating the claimed logarithmic feature efficiency.

**Weaknesses:**

- The experimental setup is incompletely described. For example, it is not stated how many layers are used for the MLPs and Transformers, how the sequences in the language are sampled for transformers, etc.
- The novelty of this work is not properly discussed in the context of existing work. Specifically, the algorithms learned by DNNs to perform group operations have been already studied by [1]. The current work focuses specifically on the non-commutative dihedral group, but this appears to be just a subcase of [1] since every group is isomorphic to a subgroup of a symmetric group (Cayley’s Theorem). It is not clear if the current work has any additional insights compared to [1].
- The algorithm learned by DNNs to perform the dihedral group operation is not explained with sufficient detail and clarity. The discussion in section 5 relies heavily on cosets and the Chinese Remainder Theorem (CRT), but these concepts are not adequately introduced.
- The "algorithm" recovered by the authors is quite abstract. The authors do not say how the networks weights are able to compute the Caley graph representations.

1. Chughtai, Bilal, Lawrence Chan, and Neel Nanda. "A toy model of universality: Reverse engineering how networks learn group operations." International Conference on Machine Learning. PMLR, 2023. https://arxiv.org/pdf/2302.03025

**Questions:**

- Is the Group Fourier Transform applied layer-wise or neuron-wise?
- How are clusters defined quantitatively (e.g., thresholding on frequency similarity, correlation distance)?
- Could the same procedure be applied to arbitrary networks without access to group labels?

---

> ### Author Response · Authors · 2025-11-23
>
> We thank reviewer oD9U for recognizing that our work (i) offers a first comprehensive mechanistic analysis of networks trained on non-commutative group tasks, (ii) is supported by large-scale experiments over many random seeds and architectures (MLPs and transformers), and (iii) demonstrates universal behavior across increasing dataset size (group size) with deep neural networks (DNNs) learning a logarithmic number of neural representations. We’d like to add however, that to our knowledge, **our work is the first to fully characterize and determine all the neural representations that are universally learned by different architectures trained on a class of datasets**. The reason for this is that neural representations are distributed across many neurons and there’s no known method for identifying which neuron’s correspond to which neural representations in general.
>
> Weaknesses:
> >1. Experimental setup and missing details
> “The experimental setup is incompletely described. For example, it is not stated how many layers are used for the MLPs and Transformers, how the sequences in the language are sampled for transformers, etc.”
>
> In the revision, we have expanded the “Experimental setup” paragraph to state:
> the number of layers for MLPs and transformers (one- and two-layer),
> the hidden width (512 ReLU units per layer for MLPs) and
> that input sequences are sampled uniformly from $D_{n}$,
>
> Concretely, the relevant part now reads:
> Experimental setup. We train one- and two-layer perceptrons (MLPs) to 100\% test accuracy with 512 ReLU neurons per layer,
>  [...]
> We also train one- and two-layer transformers
> [...]
> We sample pairs of group elements uniformly from $D_{n}$. More experimental details could be found in Appendix B EXPERIMENTAL SETTINGS.
>
> We also added a short “Two-layer behavior” paragraph that explains how the middle-layer representations in two-layer networks relate to the one-layer case, and point to additional quantitative results in the appendix.
>
> Two-layer behavior. For the two-layer networks, we empirically observe that the neural representations in the middle layer are geometrically highly similar to the cluster contributions to the logits in the corresponding one-layer networks. To keep the presentation concise, the main text therefore focuses on figures based on layer 1 activations, while additional quantitative results for the two-layer models are reported in Appendix B.6.1.
>
> We hope this adequately resolves your concern.
>
> >2. Relation to prior work [1] (novelty)
> “The novelty of this work is not properly discussed in the context of existing work. Specifically, the algorithms learned by DNNs to perform group operations have been already studied by [1]. The current work focuses specifically on the non-commutative dihedral group, but this appears to be just a subcase of [1] since every group is isomorphic to a subgroup of a symmetric group (Cayley’s Theorem). It is not clear if the current work has any additional insights compared to [1].”
>
> We respectfully disagree that we didn’t properly discuss [1]. Our related work section states that [1] claimed that neural networks and transformers trained on all finite groups will learn their proposed universal circuit: group composition by representations (GCR). [1] claimed their circuit, which utilized the matrix multiplication of the irreducible representation matrices of the group, was universally learned across architectures, but recent studies by [2] and [3] both found this algorithm wasn’t learned, and that instead, coset circuits were learned. This was stated on lines 148-159 of our original submission, but is now on Lines 138-149 in the revised version.
>
> **Our novelty.** We also don’t find the GCR circuit [1] claimed. Instead, we find individual neurons learn coset circuits, like [2] and [3] found on other groups. Indeed, we highlight the fact we find neurons learn cosets as a contribution, since it establishes universality with groups that are very different in structure to the dihedral group.
>
> We go much farther than [1], [2] and [3] however: our novel methodology allows us to identify every neuron that participates in a neural representation, and thus, we’re first to fully classify and characterize the neural representations learned in DNNs on a task. No prior work has proposed that the learned neural representations in DNNs trained on group tasks could be Cayley graphs, yet our work qualitatively and quantitatively shows the neural representations are Cayley graphs. Furthermore, we go even farther than determining *just* the neural representations. We identify how the representations are used by a global divide-and-conquer strategy implemented universally by different architectures. We verify these results universally, by inspecting different architectures on different training objectives: comparing MLPs performing direct classification with sequence generating autoregressive transformers trained to predict the next token in sequences.

---

> > ### Author Response · Authors · 2025-11-23
> >
> > Thus, our work achieves the highest resolution understanding of a group task ever. We present three levels of complete interpretation:
> > - Neurons: We find that individual neurons learn coset circuits (agreeing with [2] and [3]).
> > - Neural representations: composed of many different neurons, they correspond to different Cayley graphs of the group. We give an algorithm to determine the generators of these Cayley graphs and quantitatively show they’re learned over thousands of seeds.
> > - The global algorithm: by studying all the neural representations simultaneously, we see that the network uses them to perform a superposition operation akin to the classic Chinese remainder theorem's set intersection, and thus we give the global algorithm that networks of different architectures universally implement.
> >
> > We’d like to point your attention to a recent position paper that makes our work quite timely. “Position: We Need An Algorithmic Understanding of Generative AI” (https://arxiv.org/abs/2507.07544) argues that we need an algorithmic understanding of how transformers learn things.
> >
> > Our work provides this at a surprisingly high resolution, showing that the Cayley graphs are used as independent computational objects, with each being a different branch of a dynamic program. We find all Cayley graphs are highly ordered structures, and each is independent of computations in the other Cayley graph neural representations until after the last non-linear layer, when they are all superimposed as shown in Fig. 7 onto the logits. This superposition is effectively a global intersection of the different computations to uniquely determine the correct answer to the group multiplication. We also find this algorithm to be universally learned across many orders of magnitude of dihedral complexity (group size), consistently requiring a logarithmic number of neural representations to be learned.
> >
> > To finish, yes “every group is isomorphic to a subgroup of a symmetric group (Cayley’s Theorem)”, but this doesn’t imply that DNNs will universally learn to activate on cosets, or universally form neural representations corresponding to Cayley graphs to divide-and-conquer the answer to the group multiplication. There are many ways to encode functions on a group, and [2] found neurons learn cosets on permutation groups, [3] found the same on cyclic groups, and now we find the same on dihedral groups (and go significantly beyond these works by classifying the neural representations and global algorithmic computation as well as individual neuron behaviour).
> >
> > We hope our clarifications about the significance of our novelty encourage you to reconsider your score, and contribution rating.
> >
> > 3. Clarity of the algorithm, cosets, and CRT background
> > “The algorithm learned by DNNs to perform the dihedral group operation is not explained with sufficient detail and clarity.”
> > While we did inspect the weights, our work is dedicated to studying the activation geometry of the networks we train (we are doing activation-based interpretability, not weight-based).
> >
> > For each neuron, we fit first-order sinusoids corresponding to the relevant Fourier modes to its preactivations over the full $D_n \times D_n$ grid and report the resulting coefficient of determination $R^2$. In Layer 1 we fit only first-order sinusoids, whereas in
> > Layer 2 we fit a combination of first- and second-order sinusoids. We can confidently declare the matrix multiplication GCR algorithm of [1] is not learned due to the fact the phase shifts of these sine functions play a critical role in how neurons activate on cosets. The GCR algorithm does not require these phase shifts. [3] talked about this in modular addition, and already explained that the representation matrices aren’t learned, and that instead, projections of them are learned.
> >
> > Furthermore, we replace neurons with our fits and see no change to the networks test accuracy. As summarized in Table~3, the mean average $R^2$ of our fits per Fourier basis is extremely high across both MLPs and transformers (typically $\geq 0.99$ for Layer 1 and only slightly lower for Layer 2). Moreover, for both architectures, replacing every neuron's preactivation by its fitted sinusoid(s) leaves test accuracy at $100\%$ in $100\%$ of runs. This shows that the network's effective algorithm can be captured almost entirely by these low-order sinusoidal activations. Again, this is not the algorithm described by [1] due to each neuron taking support on cosets determined by the phase of the neuron. This behaviour can not be explained by matrix multiplication of irreducible representations.

---

> > > ### Author Response · Authors · 2025-11-23
> > >
> > > In the revision, we improved the visualization demonstrating how the algorithm intersects the different learned neural representations (Cayley graphs) to get the correct answer (Fig. 7). This Figure abstractly demonstrates that since every Cayley graph (neural representation) in the network has learned a different generator (and the vertices of the graph are different cosets), the intersection of activations of different Cayley graphs uniquely determines the answer. Imagine this as ReLU cutting off vertices on these graphs leaving only one coset (vertex on the graph) that’s activated on. Thus, the neural representation pushes mass onto each logit that’s in the coset equivalence class of c, the correct answer.
> > >
> > > >“The discussion in section 5 relies heavily on cosets and the Chinese Remainder Theorem (CRT), but these concepts are not adequately introduced.”
> > >
> > > 1. Moved and simplified the coset discussion from appendix into main background.
> > >
> > > The paragraph gives a concrete, group-theoretic but less technical description of what cosets mean for the patterns we see in neuron activations, and explains approximate cosets:
> > >
> > > “Given a subgroup $H \le G$, the left cosets $gH$ all have the same cardinality and form a partition of $G$. In our setting, this corresponds to the case where a learned “frequency” is well aligned with the group structure, so that the neuron’s response is (approximately) constant on each coset block. By contrast, when the frequency does not divide the group size (for example, attempting to ``divide'' the dihedral group $D_{18}$ by $5$), no such exact partition exists: the neuron’s activations vary almost uniquely across individual group elements. We refer to this pattern of responses as an \emph{approximate coset}.
> > >
> > > 2. Added a description of the CRT.
> > >
> > > “The Chinese remainder theorem states that when the moduli $m$ and $n$ are coprime, arithmetic modulo $mn$ is equivalent to performing arithmetic modulo $m$ and modulo $n$ in parallel: every integer $x$ modulo $mn$ is uniquely determined by the pair of its remainders $(x \bmod m, x \bmod n)$. In our setting, this provides intuition for when a ``frequency'' or pattern factors through a product of smaller cyclic components of a group.”
> > >
> > > Questions.
> > >
> > > > Q: Is the Group Fourier Transform applied layer-wise or neuron-wise?
> > >
> > >  It is applied neuron-wise as stated on lines 208-211 in the original submission.
> > >
> > > > Q: How are clusters defined quantitatively (e.g., thresholding on frequency similarity, correlation distance)?
> > >
> > > We mentioned the way of clustering briefly (revision lines 306-307) (line 208 original). Specifically:
> > >
> > > For each neuron $h$, we treat its pre-activation as a function $f_h : D_{n} \times D_{n} \to \mathbb{R}$ and compute its neuron-wise group Fourier transform $\widehat{f}_h$ over the input grid. We then identify the dominant frequency by finding the Fourier basis element(s) with the largest-magnitude coefficient. Neurons are then grouped into their neural representations based on their dominant Fourier components.
> > >
> > > > Q: Could the same procedure be applied to arbitrary networks without access to group labels?
> > >
> > > No, as our limitations state, the same procedure can not be applied to networks trained on arbitrary datasets, but we see this as a necessary need for future work on other group tasks. One of the best reasons to study group multiplications is that it makes progress toward understanding how to automate circuit detection. At this point in time, we don’t even know if it’s possible to automate circuit detection in arbitrary groups, and our step toward this is a major contribution.
> > >
> > > We believe it’s the case that future work on group multiplications will reveal more about the nature of universality. For now, the universality does seem to be that neural networks are learning to divide-and-conquer [3]. Should this be true elsewhere, it will narrow the “space” of possible solutions that networks learn.
> > >
> > > We hope our explanation of novelty and clarifications above allow you to reconsider your scoring. We’re confident that our results will be of interest to the AI safety community, if only to serve as the first proof of concept that it’s possible to identify the neuron’s composing specific neural representations and fully characterize the neural representations and how they're used algorithmically.
> > >
> > > [1] Chughtai, Bilal, Lawrence Chan, and Neel Nanda. "A toy model of universality: Reverse engineering how networks learn group operations." International Conference on Machine Learning. PMLR, 2023. https://arxiv.org/pdf/2302.03025
> > >
> > > [2] Dashiell Stander, Qinan Yu, Honglu Fan, and Stella Biderman. Grokking group multiplication with cosets. In Forty-first International Conference on Machine Learning, 2024.
> > >
> > > [3] McCracken, G., Moisescu-Pareja, G., Letourneau, V., Precup, D., & Love, J. (2025). Uncovering a Universal Abstract Algorithm for Modular Addition in Neural Networks. arXiv:2505.18266.

---

> > > > ### Comment · Reviewer_oD9U · 2025-11-23
> > > >
> > > > I thank the authors for their comprehensive response and for integrating my suggestions. I believe that the revised manuscript is addressing some of my original concerns, hence I have increased my score accordingly. However, I still lean towards reject because I believe this work has a significant limitation of being entirely focused on activations while not explaining how those activations emerge as a consequence of the weights.

---

> > > > > ### Author Response · Authors · 2025-11-24
> > > > >
> > > > > We respectfully disagree and hope that, by providing additional context about activation-based interpretability, you may reconsider this concern, especially given that your stated confidence is 2.
> > > > >
> > > > > **Our paper provides a proof of concept that to our knowledge, has never been done before: our methodology explicitly and directly identifies every learned neural representation and every neuron composing each one.** Our approach is in the activation-based interpretability paradigm, which treats the learned representations (hidden activations) as the primary object of study.
> > > > >
> > > > > **Criticising our work on the grounds that it “only” studies activations would apply equally to an enormous body of influential interpretability research.** Many high-impact works form their conclusions solely through activation-based analyses. For example, sparse autoencoders, which are **the current state of the art for discovering monosemantic, interpretable features, operate entirely on hidden activations rather than raw weight matrices** (Cunningham et al., Sparse Autoencoders Find Highly Interpretable Features in Language Models”  https://arxiv.org/abs/2309.08600).
> > > > > Furthermore, the seminal work of Nanda et al., “Progress measures for grokking via mechanistic interpretability", was also activation based, with weights only being inspected in general ways like L2 norms and Fourier transforms. This paper is a direct ancestor that has led to our work.
> > > > >
> > > > > **Most importantly: “The goal of interpretability is to produce abstract explanations of internal mechanisms of a model.”** (Section 3.4.1., Anwar et al. Foundational Challenges in Assuring Alignment and Safety of Large Language Models, https://arxiv.org/pdf/2404.09932). It’s well known that at the parameter level, many transformations (e.g., neuron permutations, basis rotations) leave the network’s overall function unchanged, but drastically alter weight matrices and weight based interpretations. Due to the infinitely many possible linear transformations that can mutate weights, different weight parametrizations can implement very similar activation patterns. Also, due to the fact neurons can be permuted (DNN neuron placement is permutation invariant), there exists only a known O(n!) way to find the correct ordering on neurons (and their weights) between hidden layers in networks with depths > 1. Activation-based approaches instead look for stable representational structure that is **invariant to such reparameterizations.**
> > > > >
> > > > > That said, we’re the first work to directly explain the learned activation geometry at three different levels of abstraction on a dataset. We explain the activations of every neuron in trained DNNs (and these activations are a direct result of the weights). We explain the activation geometry of every possible neural representation by directly finding the neural representations (i.e. we do it directly, not via a sparse autoencoder). Putting the neural representations together, we find a universal algorithm is learned, even across architectures with very different training objectives (direct classifying MLPs vs sequence generating transformers outputting the next token).

---

### Official Review · Reviewer_NHvf · 2025-10-31

**Soundness:** 3
**Presentation:** 3
**Contribution:** 3
**Rating:** 6
**Confidence:** 2

**Summary:**

The work is centered around *how* feed forward and transformer neural networks learn dihedral multiplication. The authors present an argument for how the cyclic nature of dihedral multiplication presents an interesting testbed. They follow this up studying the activations and their principal components. This study reveals that the manifold of learned neural representations correspond to Cayley graphs.

**Strengths:**

The PCA approach to neural representation for their test bed of learning group multiplication in the dihedral group $D_n$ is novel and very interesting. I found the exposition instructive, given some background. Their results align perfectly with the hypothesis in toy setup. They position their work in light of previous work very well. They compare to literature on grokking and interpretability using analytically generated datasets. I found the result in section 5: neural networks learn $O(\log n)$ algorithm for group operation to be very interesting.

**Weaknesses:**

One challenge I faced while reading the work was the background. I believe the authors can present the following topics in the main body of the paper:
1. Group Fourier Transform: the authors use and re-use this idea throughout but the background is relegated to the Appendix. I believe a brief intro would benefit all readers with varying levels of familiarity with Fourier analysis.

2. An example of a coset (e.g. all with + sign or - sign) would help a reader like me.

3. I am not sure how the learning dynamics fit into the picture? Since the problem is motivated from a Grokking perspective, which has to do with learning dynamics, I would be interested in at what stage in their training do NNs Grok these divide and conquer algorithms.

------------------------------------------

Minor issues:

Line 101-103: I would use citep here to put all the references in ()

Line 432-433: broken reference "??"

**Questions:**

See above.

---

> ### Author Response · Authors · 2025-11-22
>
> We thank Reviewer NHvf for the thoughtful feedback. We agree that moving background from the appendix to the main background serves to improve reader accessibility.
>
> > 1. Group Fourier Transform (GFT) “Group Fourier Transform: the authors use and re-use this idea throughout but the background is relegated to the Appendix. I believe a brief intro would benefit all readers with varying levels of familiarity with Fourier analysis. (introduce GFT in the background)”
>
> We agree and have integrated this suggestion. We moved an intuitive introduction of the GFT from the appendix to the main text. “Analogously to the classical Fourier transform, which decomposes a time-domain signal into sinusoidal components, the group Fourier transform (GFT) decomposes a complex-valued function on a group $G$ into spectral components that reflect the symmetry structure of $G$. These components are indexed by the (unitary) irreducible representations of $G$ and capture how the function varies along different symmetry modes, such as rotational versus reflective modes in the case of the dihedral group $D_n$; see Appendix A.1.2.”
> We keep the more formal, representation-theoretic introduction in Appendix A.1.2, so that readers who want the rigorous definition can consult it without interrupting the main narrative.
>
> > 2. “An example of a coset (e.g. all with + sign or - sign) would help a reader like me.”
>
> The neuron in Fig. 2's right most column has learned the + sign coset for a, and the + sign coset for b, which is the + sign coset for c because a rotation times a rotation is a rotation (sign +). This neuron does not activate for reflections. This is stated in the Figure caption on lines 212-214 in the revised version (135-137 original submission), but we will add a pointer toward it from the main text.
>
> >3.  “I am not sure how the learning dynamics fit into the picture? Since the problem is motivated from a Grokking perspective, which has to do with learning dynamics, I would be interested in at what stage in their training do NNs Grok these divide and conquer algorithms.”
>
> Our work is not motivated by grokking, and doesn't aim to study learning dynamics. We only intended to give trivia, that grokking was first observed on a group multiplication dataset, being modular addition.
>
> The answer however, is once the test accuracy is 100%. This is because the divide and conquer algorithm requires that the neural representations are already learned, which can only be well formed after the neuron’s composing them finish learning their respective coset structures.
>
> Please consider that this paper is scoped to what is learned by deep neural networks DNNs trained on dihedral multiplication and gives the highest resolution interpretation ever given on a toy dataset. To our knowledge, we are the first work to fully characterize and determine all neural representations that are universally learned across different DNN architectures (direct classification MLPs vs sequence generating autoregressive transformers predicting next tokens). The reason we're first is that in general, neural representations are distributed across many neurons and there’s no known method for consistently identifying which neuron’s correspond to which neural representation(s).
>
> We believe our proof of concept that this is possible will influence future work. Indeed, our revised paper refines this statement and makes it clear how we build on the existing literature and clearly formulates a conjecture that future work on group multiplications can investigate.
>
> We give three independent levels of abstraction where universality may be falsified by studying DNNs learning group multiplications.
> - Globally: what are the solutions learned by DNNs? Are they always divide-and-conquer algorithms?
> - Structurally: what are the neural representations? Are they always Cayley graphs embedded on manifolds?
> - Neuron-wise: what are the activations? Are they always cosets?
>
> **Why is this important and why are we excited about it?** There are three major deep learning hypotheses influencing thoughts about what DNNs learn. (i) The first is the universality hypothesis, that DNNs trained on similar data will make use of similar principles (Olah et al.; Li et al.). Our work sheds light into universality and the: (ii) manifold hypothesis, that DNNs efficiently learn by finding low dimensional manifolds that the dataset lives on (Goodfellow et al.) and (iii) platonic representation hypothesis, which posits that architectures trained on different learning objectives at large scales, learn a platonic representation, meaning that distances between datapoints in different classes of data “converge” (Huh et al.).

---

> ### Author Response · Authors · 2025-11-22
>
> Our work offers the ability to inspect and falsifiably investigate these three major hypotheses simultaneously. Should our conjecture (and McCracken et al.’s) be verified across a diverse breadth of group tasks, it has the potential to show that universality exists at multiple scales of abstraction, and would suggest there are universality classes of particular manifolds that are learned by DNNs. Furthermore, it would suggest that neural representations universally order points on group tasks. Therefore, we claim our work is of significant interest as it provides many roads toward understanding the nature of solutions learned by DNNs.
>
> We hope that after reviewing our rebuttal and revisions to our paper you’ll consider increasing your score. We believe that our detailed interpretation of MLPs and transformers at three different levels of abstraction will be appreciated by many communities in machine learning. Recently, position papers have called for the type of analysis our novel methodology has provided, making our work quite timely. For example, see “Position: We Need An Algorithmic Understanding of Generative AI” (https://arxiv.org/abs/2507.07544), which states “What algorithms do LLMs actually learn and use to solve problems? Studies addressing this question are sparse, as research priorities are focused on improving performance through scale, leaving a theoretical and empirical gap in understanding emergent algorithms.”
> Indeed, our work serves to close this gap. Not only do we uncover the divide-and-conquer strategy used universally between different architectures learning dihedral multiplication, but we identify and fully characterize the different neural representations that can be learned across random seeds. We show these neural representations correspond to simpler computations (subproblems), being Cayley graphs that DNNs require only a logarithmic number of these simpler subproblems to be learned in order to solve the task. We find this particularly satisfying, as it helps explain the efficiency of the solutions deep neural networks learn.
>
> Again, we thank you for your time and review and hope to discuss our work further with you.
>
> McCracken, G., Moisescu-Pareja, G., Letourneau, V., Precup, D., & Love, J. (2025). Uncovering a Universal Abstract Algorithm for Modular Addition in Neural Networks. arXiv:2505.18266.
>
> Olah, C., Mordvintsev, A., & Schubert, L. (2017). Feature Visualization. Distill, 2(11), e7. https://doi.org/10.23915/distill.00007
>
> Olah, C., Cammarata, N., Schubert, L., Goh, G., Petrov, M., and Carter, S. Zoom In: An Introduction to Circuits. Distill, 5(3):e00024.001, March 2020. ISSN 2476-0757. doi: 10.23915/distill.00024.001.
>
> Li, Y., Yosinski, J., Clune, J., Lipson, H., & Hopcroft, J. (2016). Convergent Learning: Do different neural networks learn the same representations? arXiv:1511.07543.
>
> Goodfellow, I., Bengio, Y., & Courville, A. (2016). Deep Learning. MIT Press. (Chapter 5.11.3)
>
> Minyoung Huh, Brian Cheung, Tongzhou Wang, and Phillip Isola. The platonic representation hypothesis, 2024. URL https://arxiv.org/abs/2405.07987.

---

### Author Response · Authors · 2025-12-03
**Global rebuttal**

Dear AC (and reviewers),

Thank you for handling our paper, given recent complications. We want to briefly summarize where the paper stands post-rebuttal/revision cycle.

In short: reviewer concerns directly led to concrete additions that completed the story. At this point, we do not see remaining gaps that would change the paper; further work would be natural extensions rather than required fixes. Before public awareness of the "leak", both reviewers that participated in discussion increased their scores.

In response to the reviews, we:

- Added more quantitative tests to direct addressing RG3S’s request for stronger empirical validation. These metrics show that Cayley graphs are consistently learned across random seeds and resulted in definitive verification of our claims.

- Per EEmm’s comments on universality, we performed an exhaustive sweep over training hyperparameters (different activation functions, widths, depths, learning rates, etc.) using R^2 and our new quantitative tests. The resulting plots show that our claimed solution is learned everywhere in the region where networks robustly generalize on the task. On the boundary of learnability, deviations are explained by convergence to noisier local minima rather than learning a different mechanism than we proposed.

Together, these quantitative additions make our claims robust and empirically undoubtable. Thus, we succeed in precisely characterizing what MLPs and autoregressive transformers learn on this family of group tasks. The reviewers consistently highlighted the novelty of **directly** inspecting the activation geometry and the interest of the methodology, which is designed to transfer to *every* group-structured task.

Most importantly: **to our knowledge this is the first work to ever use a methodology capable of directly inspecting neural representations on any task and go beyond inspecting them, giving closed-form mathematical understanding of them**.

All reviewers noted this, stating about our methodology:

- NHvf: "novel and very interesting" (didn't participate pre-leak)

- oD9U: "First comprehensive mechanistic analysis of neural networks trained on non-commutative group tasks" (increased score to 4 (confidence 2), with discussion prematurely ended due to leak)

- RG3S: "Very interesting approach to gain insight into learning" (increased score to 6 (confidence 4), with discussion prematurely ended due to leak)

- EEmm: "Novel Perspective on Neural Representations" (didn't participate pre-leak)

For these reasons, we believe the rebuttal process has substantially strengthened the work, and that it is now in a mature and complete state. Because groups underlie many algorithmic settings (e.g., sorting lists is a group task), the work now offers a clear, well-supported framework that can be applied and tested more broadly to **directly inspect the neural representations on many algorithmic tasks of interest**. The next steps are follow-on projects applying our method to reverse engineering deep transformers and MLPs trained on tasks of higher complexity (like sorting lists).

To conclude, we believe the reviewers, particularly RG3S and EEmm, played critical roles bringing this paper to completion. As a final send-off: we thank the reviewers one last time for their comments and requests.

---

### Meta-Review · Area_Chair_ArQy · 2026-01-07

**Summary:**

The article investigates how neural networks implement dihedral multiplication.

Initial reviews expressed concerns with the structure of the study, lack of hypotheses, lack of theoretical justification. The initial rebuttal addressed some of the concerns and prompted some of reviewers to revise initial scores (to 6,4,6,4). This is a positive trend, but still borderline.

I believe the extent of the revisions merits further review. Therefore I am recommending reject with encouragement to revise and resubmit.

**Reviewer Concerns:**

Reviewer RG3S:

exploratory analysis rather than a hypothesis-driven study
-> Response in rebuttal

risk of circular reasoning
-> Response in rebuttal

Reviewer oD9U:

experimental setup is incompletely described
-> Response in rebuttal

**Reviewer Scores:**

For each review, specify how you think the reviewer would have changed their score if they had been able to participate fully in the discussion.

Reviewer NHvf: 6 -> 6
Reviewer oD9U: 2 -> 4
Reviewer RG3S: 2 -> 6
Reviewer EEmm: 4 -> 4

---

### Decision · Program_Chairs · 2026-01-26

Reject